

SciPost Phys. Lect. Notes 8 (2019)

# The Tensor Networks Anthology: Simulation techniques for many-body quantum lattice systems

Pietro Silvi[1,2⋆], Ferdinand Tschirsich[1], Matthias Gerster[1], Johannes Jünemann[4,5], Daniel Jaschke[1,3], Matteo Rizzi[4] and Simone Montangero[1,6,7]

**1** Institute for Complex Quantum Systems and Center for Integrated Quantum Science and Technologies, Universität Ulm, D-89069 Ulm, Germany.
**2** Institute for Theoretical Physics, Innsbruck University, A-6020 Innsbruck, Austria.
**3** Department of Physics, Colorado School of Mines, Golden, Colorado 80401, USA.
**4** Institut für Physik, Johannes Gutenberg-Universität, D-55128 Mainz, Germany.
**5** Graduate School Materials Science in Mainz, D-55128 Mainz, Germany.
**6** Theoretische Physik, Universität des Saarlandes, D-66123 Saarbrücken, Germany.
**7** Dipartimento di Fisica e Astronomia, Università degli Studi di Padova, I-35131 Italy.

⋆ pietro.silvi@uibk.ac.at

## Abstract

We present a compendium of numerical simulation techniques, based on tensor network methods, aiming to address problems of many-body quantum mechanics on a classical computer. The core setting of this anthology are lattice problems in low spatial dimension at finite size, a physical scenario where tensor network methods, both Density Matrix Renormalization Group and beyond, have long proven to be winning strategies. Here we explore in detail the numerical frameworks and methods employed to deal with low-dimensional physical setups, from a computational physics perspective. We focus on symmetries and closed-system simulations in arbitrary boundary conditions, while discussing the numerical data structures and linear algebra manipulation routines involved, which form the core libraries of any tensor network code. At a higher level, we put the spotlight on loop-free network geometries, discussing their advantages, and presenting in detail algorithms to simulate low-energy equilibrium states. Accompanied by discussions of data structures, numerical techniques and performance, this anthology serves as a programmer's companion, as well as a self-contained introduction and review of the basic and selected advanced concepts in tensor networks, including examples of their applications.



# 1  Introduction

The understanding of quantum many-body (QMB) mechanics [1] is undoubtedly one of the main scientific targets pursued by modern physics. Discovering, characterizing, and finally engineering novel, exotic phases of quantum matter [2] will be the key for the development of quantum technologies [3], quantum computation and information platforms [4], and even lead to an enhanced, non-perturbative understanding of the building blocks of reality, including high energy settings [5–7]. The research development is hindered by the inherent difficulty of such problems, by the extremely small amount of exactly solvable models [8] and the limitations posed by the various approximate approaches that have been proposed so far [9].

From the computational point of view, the obstacle for addressing quantum many-body problems is the exponential growth in the dimension of the configurations space with the number of elementary constituents of the system (number of quantum particles, lattice sites, and so forth). Exact diagonalization techniques [10] are thus restricted to small system sizes, and inapt to capture the thermodynamical limit of QMB systems, which characterizes the macroscopic phases of matter.

Approaches relying on a semi-classical treatment of the quantum degrees of freedom [11], such as mean field techniques e.g. in the form of the Hartree–Fock method [12], have been applied with various degrees of success. It is well known, however, that while they are an accurate treatment in high spatial dimensions, they suffer in low dimensions [13], especially in 1D where entanglement and quantum fluctuations are so important that it is not even possible to establish true long-range order through continuous symmetry breaking [14].

On the other hand, stochastic methods for simulating QMB systems at equilibrium, such as Monte Carlo techniques, have been extremely successful [15, 16]. They have the advantage of capturing both quantum and statistical content of the system by reconstructing the partition functions of a model, and have been adopted on a wide scale. Difficulties arise when the stochastic quasi-probability distribution to be reconstructed is non-positive, or even complex (sign problem) [17, 18].

An alternative approach to address these intrinsic quantum many-body problems is the construction of a computationally manageable variational ansatz, which is capable of capturing the many-body properties of a system while being as unbiased as possible. Tensor Network (TN) states fulfill this criterion [19–24]: By definition, their computational complexity is directly controlled by (or controls) their entanglement. In turn, TN states can span the whole quantum state manifold constrained by such entanglement bounds [25–27]. Consequently, TNs are an efficient representation in those cases where the target QMB state exhibits sufficiently low entanglement. Fortunately, tight constraints on the entanglement content of the low-energy states, the so-called *area laws* of entanglement have been proven for many physical systems of interest, making them addressable with this ansatz [28–37]. TN states have the advantage of being extremely flexible and numerically efficient. Several classes of tensor networks, displaying various geometries and topologies, have been introduced over the years, and despite their differences, they share a vast common ground. Above all, the fact that any type of QMB simulation relies heavily on linear algebra operations and manipulation at the level of tensors [38].

Accordingly, in this anthology we will provide a detailed insight into the numerical algebraic operations shared by most tensor network methods, highlighting useful data structures as well as common techniques for computational speed-up and information storage, manipulation and compression. As the focus of the manuscript is on numerical encoding and computational aspects of TN structures, it does not contain an extended review of physical results or a pedagogical introduction to the field of TNs in general. For those two aspects, we refer the reader to the excellent reviews in Refs. [19, 39].

The anthology is thus intended both as a review, as it collects known methods and strategies for many-body simulations with TNs, and as a programmer's manual, since it discusses the necessary constructs and routines in a programming language-independent fashion. The anthology can be also used as a basis for teaching a technical introductory course on TNs. We focus mostly on those tensor network techniques which have been previously published by some of the authors. However, we also introduce some novel strategies for improved simulation: As a highlight, we discuss ground state search featuring an innovative *single-tensor update with subspace expansion*, which exhibits high performance and is fully compatible with symmetric TN architectures.

## 1.1 Structure of the Anthology

The anthology is organized as follows: In Sec. 1 we review shortly the historical background and some general theoretical concepts related to tensor network states. In Sec. 2 we characterize the data structures which build the TN state, and we list the standard linear algebra tools employed for common tasks. A brief review on embedding symmetries into the TN design is provided in Sec. 3, alongside numerical techniques to exploit the (Abelian) symmetry content for computational speed-up and canonical targeting. In Sec. 4 we introduce tensor network structures without loops (closed cycles in the network geometry) and explain how we can explore possible gauges to give these states favorable features. A generalization of the Density Matrix Renormalization Group (DMRG) applying to tensor networks of this type is detailed in Sec. 5, as well as instructions for practical realizations of the algorithm. We draw our conclusions and sketch the outlook for further research in Sec. 6. A diagrammatic representation of the arrangement of the anthology contents is shown in Fig. 1, where the logical dependencies among the topics are highlighted, and depicted as embedded boxes. This diagram, in turn, reflects the hierarchy among the modules of an ideal TN programming library: Using this hierarchy, the outmost component (ground state search) can be easily replaced with several other physically relevant tasks, such as real-time dynamics (see also Sec. 1.5).

## 1.2 Short History of Tensor Networks for Quantum Many-Body Problems

The Density Matrix Renormalization Group (DMRG), introduced in the early '90s [40, 41], was developed as a reformulation of the numerical renormalization group [42, 43], where now the renormalized, effective, degrees of freedom were identified in the density matrix eigenspaces instead of in the lowest energy manifold of real space subsystems [44]. Eventually it was realized that DMRG can be recast as a minimization algorithm over the class of finitely-correlated states, which admit a formulation as Matrix Product States (MPS) [25, 45–47]. In time it became clear that the reason for the success of DMRG is the limited entanglement present in ground states of short-range interacting Hamiltonians, as it obeys the area laws of entanglement [28–33, 37], and such scaling is well captured by MPS for one-dimensional systems. On the one hand, such reformulation into MPS language [27, 39] opened a clear path for substantial improvements and expansions of the DMRG technique. Prominent examples are: concepts for out-of-equilibrium evolution [26, 48–51], direct addressing of the thermodynamic limit [52–56], exploration of finite temperatures [57, 58], and direct access to the density

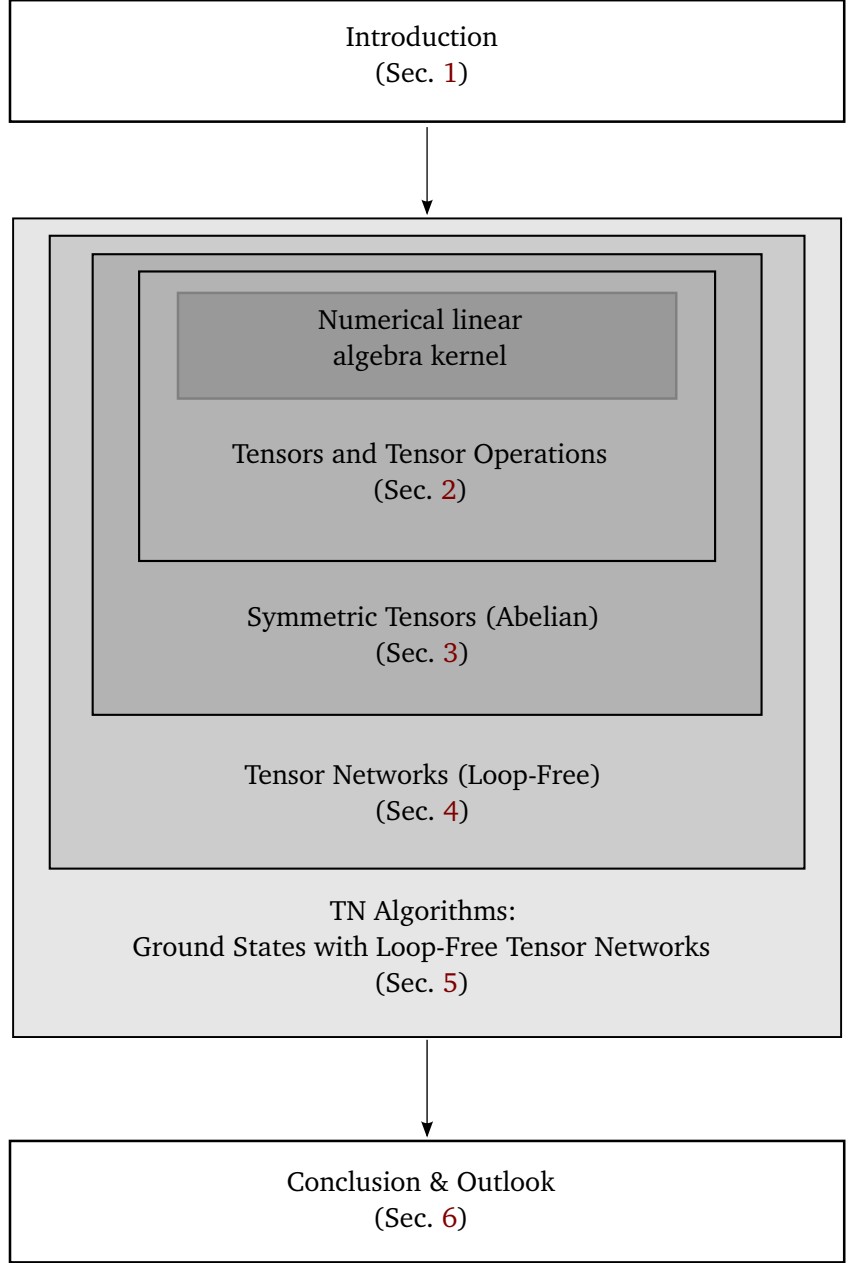

Figure 1: Organization of the anthology. In an embedded-boxes structure, the introduction is followed by a self-consistent treatment of elementary tensors and tensor-operations plus their extension in presence of symmetries. On this foundation we build up and investigate (loop-free) tensor networks and algorithms. The parts about symmetries are optional and can be skipped. As this manuscript is intended as a practical anthology, the structure is chosen such that it can be directly mapped into numerical libraries, organized in the same hierarchy, starting from a kernel of numerical linear-algebra routines which are widely available and thus not covered in this anthology.

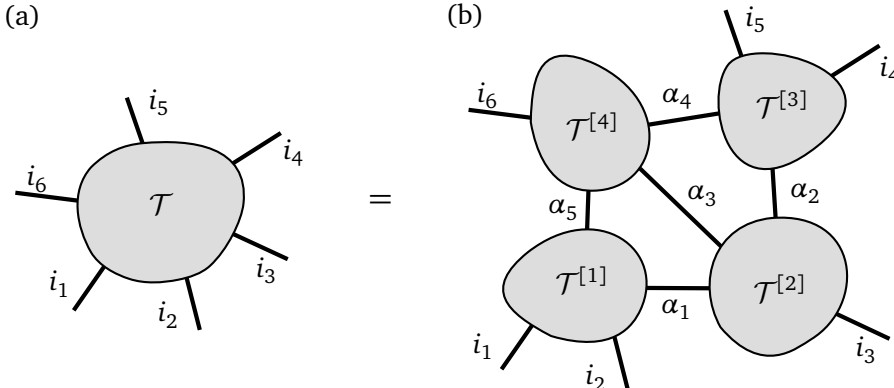

Figure 2: (a) The coefficients of a QMB wave function (here with $N = 6$ constituents) can be understood as the elements of a tensor $\mathcal{T}$ with $N$ links. Graphically, we represent a tensor by a circle-like object (node) with an open link for every physical site $s$. Fixing the indices $i_s$ for all links $s$ then identifies the element $\mathcal{T}_{i_1 \ldots i_N}$. (b) A possible TN decomposition of the system: The state is now represented by $Q = 4$ smaller tensors $\mathcal{T}^{[q]}$, which are connected through $L = 5$ auxiliary links. The original tensor object is obtained once again by contracting tensors over shared indices, which are represented by the links connecting them. For a detailed introduction to the graphical notation, see Sec. 2.

of states and spectral properties [59, 60]. On the other hand, the discovery encouraged the engineering of new classes of tailored variational wave functions with the common desirable feature of storing the QMB information in compact tensors, i.e. easy to handle numerical objects, and controlling the entanglement through a network of virtual connections [19]. Noteworthy classes of tensor networks that have been proposed towards the quantum many-body problem are: Projected Entangled Pair States (PEPS) [61, 62], weighted graph states [63], the Multiscale Entanglement Renormalization Ansatz (MERA) [64–67], branching MERA [68, 69], Tree Tensor Networks (TTN) [70–75], entangled-plaquette states [76], string-bond states [77], tensor networks with graph enhancement [78, 79] and continuous MPS [80]. Additionally, various proposals have been put forward to embed known stochastic variational techniques, such as Monte Carlo, into the tensor network framework [76, 81–86], to connect tensor networks to quantum error correction techniques [87], and to construct direct relations between few-body and many-body quantum systems [88]. The numerical tools we will introduce in the next two sections indeed apply to most of these TN classes.

## 1.3  Tensor Network States in a Nutshell

A tensor network state is a tailored quantum many-body wave function ansatz. For a discrete system with $N$ components, the QMB state can be written

$$|\Psi_{\mathrm{QMB}}\rangle = \sum_{i_1,\ldots,i_N} \mathcal{T}_{i_1,\ldots,i_N} |i_1,\ldots,i_N\rangle, \tag{1}$$

where $\{|i_s\rangle\}_s$ is a canonical basis of the subsystem $s$. The complex coefficients of this Hilbert space wave function, $\mathcal{T}_{i_1 \ldots i_N}$, determine the state uniquely.

The TN prescription expresses the amplitudes tensor as the contraction of a set of smaller tensors $\mathcal{T}^{[q]}$ over auxiliary indices (sometimes also called virtual indices), namely

$$\mathcal{T}_{i_1 \ldots i_N} = \sum_{\alpha_1 \ldots \alpha_L} \mathcal{T}^{[1]}_{\{i\}_1,\{\alpha\}_1} \mathcal{T}^{[2]}_{\{i\}_2,\{\alpha\}_2} \cdots \mathcal{T}^{[Q]}_{\{i\}_Q,\{\alpha\}_Q}, \tag{2}$$

for a network with $Q$ nodes and $L$ links. Each link connects and is shared by two tensors only, see Fig. 2(b). Here a tensor $\mathcal{T}^{[q]}$ can possess any number of physical indices $\{i\}_q = \{i_s : s \text{ is physical link at node } q\}$ as well as any number of auxiliary indices $\{\alpha\}_q = \{\alpha_\eta : \eta \text{ is auxiliary link at node } q\}$. However, to make the ansatz computationally efficient for QMB states, the number of links (indices) connected to it *should not scale* with the system size $N$. Similarly, the auxiliary dimensions $D_\eta = \#\{\alpha_\eta\}$ (where the symbol $\#$ denotes the cardinality of a set) should be non-scaling quantities. Conversely, the number of tensors $Q$ will scale with $N$.

Throughout this anthology, we use a graphical, compact notation to express tensors and tensor networks, and the typical operations performed on them. These diagrams allow us to recast most of the equations in a form which is less verbose, often clearer, and mnemonically convenient, such as in Fig. 2 which represents Eq. (2). More details about the diagrammatic representation of TNs will be provided in Sec. 2 and Sec. 3.

It is easy to show [89] that the entanglement entropy $\mathcal{E}$ of the TN state $|\Psi_{QMB}\rangle$ under system bipartitions satisfies rigorous bounds, which depend on the TN geometry. In fact, the TN state is a wave function ansatz whose only constraint is a direct control of its entanglement properties, and is thus considered an only slightly biased ansatz. For the same reason, it is typical to express TN states in real space, where the $N$ components are the lattice sites: it is within this setup that one can exploit the area laws and successfully work with TN states having low entanglement content [21, 23, 30, 31, 37, 90]. Tensor network state descriptions in momentum space representation have also been proposed and implemented in simulations [91–95].

## 1.4 Many-Body Statistics

One of the major features of the tensor network ansatz is its flexibility: In fact, it can be adapted for several types of local degrees of freedom and quantum statistics. The typical scenarios dealt with are the following:

(i) **Spin systems** - Spin lattices are the natural setting for tensor networks. The quantum spins sit on a discrete lattice and their local Hilbert space $\{|i_s\rangle\}$ has a finite dimension $d$ fixed by the problem under study; e.g. for lattice sites with spin $l$: $d = 2l + 1$.

(ii) **Bosonic lattice** - Boson commutation rules are analogous to the ones for spins, but normally, bosons do not have a compact representation space for their local degree of freedom. Unless some inherent mechanism of the model makes the bosons $d$-hardcore, an artificial cutoff of the local dimension $d$ must be introduced. This is, however, usually not dramatic, since typically the repulsive interactions make highly occupied states very improbable and the consistency of results can be checked by enlarging $d$ slightly.

(iii) **Fermionic lattice** - The standard way of dealing with fermionic models is mapping them into a spin system via a Jordan–Wigner (JW) transformation [96, 97]. Doing so is very convenient for short-range interacting (parity-preserving) 1D models, as the JW-mapping preserves the short-range nature. Other techniques to encode the fermionic exchange statistics directly in the tensor network contraction rules are known [98, 99], but it is argued that these methods are more useful in 2D, where the JW-mapping does not preserve interaction range.

(iv) **Hybrid lattices** - Tensor networks can also handle hybrid theories which include both bosons and fermions. In this scenario, the two species usually live in two distinct sublattices. Lattice gauge systems such as the lattice Yang–Mills theories are a textbook example of this setting [5]: In these models, matter can be fermionic, while the gauge field is typically bosonic [100–107].

(v) **Distinguishable particles** - When quantum particles are distinguishable and discrete, they can be considered as the lattice sites, and both their motional and internal degrees of freedom may be described in a first quantization fashion. This scenario appears, for instance, when studying phononic non-linearities in trapped ion systems [108]. In this representation, the motional modes of each particle are often truncated to a compact dimension by selecting $d$ orbitals $\{|i\rangle\}$ to act as local canonical basis, e.g. those with lowest local energy, or the solutions of a mean field approach [109, 110].

It is important to mention that the TN ansatz can be extended to encompass variational states for quantum field theories, for instance via the continuous-MPS formalism [80, 111]. However, this scenario lies beyond the purpose of this paper and will not be discussed here.

## 1.5 Typically Addressed Problems

Tensor network states are routinely employed to tackle the following physical problems on 1D lattices:

 (i) **Ground states** of interacting Hamiltonians, usually (but not necessarily) built on two-body interactions. For the model to be manifestly 1D, the interaction has to be either finite-range or decrease with range sufficiently fast [112]. Ground state search algorithms based on TN technology can usually be extended to target other low energy eigenstates [113].

 (ii) **Non-equilibrium dynamics** of arbitrary QMB states under unitary Hamiltonian evolution. The Hamiltonian itself can be time-dependent, thus encompassing quenches [114, 115], quasi-adiabatic regimes [116, 117], and even optimal control [118]. The most common strategies to perform such dynamics on the TN states rely either on a Suzuki–Trotter decomposition of the unitary evolution (TEBD, tDMRG) [48, 49] or on a Time-Dependent Variational Principle (TDVP) on the TN class [51], however, alternative routes are known [119–122]. Clearly, these techniques perform best when describing processes with small entanglement growth: Either short time scale quenches [114, 115], quasi-adiabatic quenches [116, 117, 123], or controlled dynamics [124]. Long time scale quenches are also accessible for small system sizes [125].

 (iii) **Annealing and Gibbs states**. Strategies to achieve imaginary-time evolution are analogous to real-time unitary evolution, but in order to reconstruct the Gibbs ensemble at finite temperatures, one must either perform several pure state simulations or extend tensor networks to mixed states [57, 58, 126].

 (iv) **Open system dynamics**, usually undergoing a Lindblad master equation evolution (Markovian scenario), with local dissipative quantum channels. Besides the need to reconstruct mixed states, this setup has the additional requirement of designing a scheme for implementing a master equation dynamics into the TN ansatz, either by direct integration [57, 58, 126] or by stochastic unravelling [127].

Depending on the type of problem one aims to address and on the boundary condition under consideration (open or periodic), some TN designs are preferable to others. In Sec. 5 we will describe in detail an algorithm for ground state search based on loop-free TNs.

## 1.6 Accessible Information in Tensor Networks

Being able to effectively probe and characterize a relevant quantum many-body state is just as important as determining the state itself to understand the underlying physics and the emergent properties of the system. The TN architectures that we discuss here are meant to describe

finite systems, while the thermodynamical limit properties are obtained via extrapolation. The thermodynamical limit can also be directly addressed, e.g. by means of infinite homogeneous tensor networks (such as i-MPS/i-PEPS [52, 53, 128] or i-TTN/i-MERA [75, 129, 130]), but these methods will not be discussed here.

Typical quantities of interest for tensor network ansatz states are:

(i) **State norm** $\langle \Psi_{\mathrm{QMB}} | \Psi_{\mathrm{QMB}} \rangle$ — The capability of extracting the norm of a tensor network in *polynomial computational time* as a function of the number of tensors in the network itself is often simply referred to as *efficient contractibility*.

Loop-free tensor networks, such as MPS and TTN, have this property built-in (assuming that the link dimensions are bounded from above), as we will remark in Sec. 4. Some tensor networks with loops, such as MERA, also have this property as a result of their specific geometry and algebraic constraints. In general, however, this is not true — the PEPS being the textbook example of a non-efficiently contractible tensor network (although approximate, stochastic methods for PEPS contraction are known [81]). In loop-free TNs and holographic TNs (MERA, branching MERA) the calculation of the state norm can even be made computationally cheaper by imposing specific TN gauge rules, as we will see in Sec. 4.

(ii) **State overlap** $\langle \Psi_{\mathrm{QMB}} | \Psi'_{\mathrm{QMB}} \rangle$ — Provided that the two states $| \Psi_{\mathrm{QMB}} \rangle$ and $| \Psi'_{\mathrm{QMB}} \rangle$ are given in the same TN geometry and that the resulting network is efficiently contractible (which is always the case for loop-free TNs), this quantity can be easily computed and the numerical cost is the same as for the state norm. Here, however, the TN gauge invariance cannot be exploited to further reduce the costs.

(iii) **Expectation values of tensor product observables** $\langle \Psi_{\mathrm{QMB}} | \bigotimes_s O^{[s]} | \Psi_{\mathrm{QMB}} \rangle$ — Efficient calculation of these quantities relies on the fact that every single-site operator can be effectively absorbed into the TN state without altering the network geometry, and thus performing local operations preserves the efficient contractibility property. This statement is true whether the single-site operator $O^{[s]}$ is homogeneous or site-dependent. Useful subclasses of this observable type contain:

- *Local observables* $\langle O^{[s]} \rangle$, useful for reconstructing reduced density matrices and identifying local order.
- *Correlation functions* $\langle O^{[s]} O^{[s+\ell]} \rangle$, important for detecting long-range order. This can, of course, be extended to $n$-points correlators.
- *String observables* $\langle O^{[s]} O^{[s+1]} \dots O^{[s+\ell-1]} O^{[s+\ell]} \rangle$, useful for pinpointing topological properties (e.g. the Haldane order [131]) and boundary order, as well as to implement long-range correlations in Jordan–Wigner frameworks.

(iv) **Expectation values of operators in a tensor network form** — When the observable itself can be expressed in a TN formalism, it is natural to extend the notion of efficient TN contractibility to the $\langle$ state $|$ operator $|$ state $\rangle$ expectation value. This is, for instance, the case when measuring the expectation value of a Matrix Product Operator (MPO) over a matrix product state, which is indeed an efficiently contractible operation [132].

(v) **Entanglement properties** — Any bipartition of the network tensors corresponds to a bipartition of the physical system. Via contraction of either of the two sub-networks it is then possible to reconstruct the spectrum of the reduced density matrix of either subsystem, and thus extract the entanglement entropy, i.e. the von Neumann entropy of the reduced density matrix, and all the Rényi entropies, ultimately enabling the study of critical phenomena by means of bipartite [73, 133], and even multipartite entanglement [134].

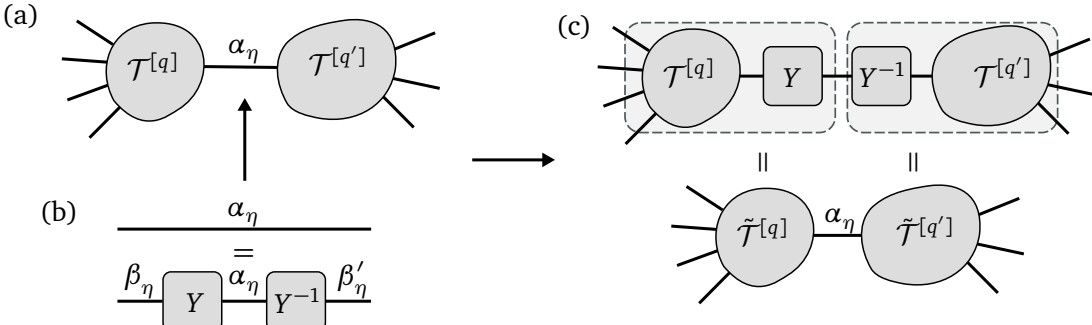

Figure 3: Gauge transformation: For a given tensor network (a), the link $\eta$ between tensors $\mathcal{T}^{[q]}$ and $\mathcal{T}^{[q']}$ is replaced by a matrix $Y$ and its inverse (b). Contracting $Y$ and $Y^{-1}$ into their respective neighboring tensors (c) gives an alternative description of the tensor network without changing the physical content of the stored information.

## 1.7 The Tensor Network Gauge

Tensor network states contain some form of information redundancy with respect to the quantum state they describe. This redundancy translates conceptually into a set of linear transformations of the tensors in the network which leave the quantum state, and thus all the physically relevant quantities, unchanged [26, 27, 39, 70]. Since these operations have no impact at the physical level, they are often referred to as *gauge transformations*, in analogy to field theories. The *gauge invariance* relies on the fact that performing local invertible manipulations on the auxiliary TN spaces does not influence the physical degrees of freedom. We will therefore introduce the concept of gauge transformation here and use it extensively in the rest of the anthology.

Specifically, let $\eta$ be a given virtual link of the tensor network state from Eqs. (1) and (2), connecting node $q$ to $q'$, with auxiliary space dimension $D_\eta$ (see Fig. 3(a)). Then, given a $D'_\eta \times D_\eta$ (left-) invertible matrix $Y_{\alpha_\eta, \alpha'_\eta}$ (i.e. $\sum_{\alpha'} Y^{-1}_{\alpha, \alpha'} Y_{\alpha', \alpha''} = \delta_{\alpha, \alpha''}$), the tensor network state is unchanged if $\mathcal{T}^{[q]}$ and $\mathcal{T}^{[q']}$ transform as

$$\mathcal{T}^{[q]}_{\{i\}, \{\alpha\}_{\backslash\eta}, \alpha_\eta} \rightarrow \tilde{\mathcal{T}}^{[q]}_{\{i\}, \{\alpha\}_{\backslash\eta}, \alpha_\eta} = \sum_{\beta_\eta} Y_{\alpha_\eta, \beta_\eta} \mathcal{T}^{[q]}_{\{i\}, \{\alpha\}_{\backslash\eta}, \beta_\eta}, \tag{3a}$$

$$\mathcal{T}^{[q']}_{\{i'\}, \{\alpha'\}_{\backslash\eta}, \alpha_\eta} \rightarrow \tilde{\mathcal{T}}^{[q']}_{\{i'\}, \{\alpha'\}_{\backslash\eta}, \alpha_\eta} = \sum_{\beta_\eta} Y^{-1}_{\beta_\eta, \alpha_\eta} \mathcal{T}^{[q']}_{\{i'\}, \{\alpha'\}_{\backslash\eta}, \beta_\eta}, \tag{3b}$$

where $\{\alpha\}_{\backslash\eta}$ collects the indices of all the virtual links of the tensor, except $\eta$. This operation is simply the contraction of $Y$ into $\mathcal{T}^{[q]}$ through link $\eta$ and simultaneous contraction of its inverse matrix $Y^{-1}$ into $\mathcal{T}^{[q']}$ through link $\eta$: it leaves the composite tensor $\mathcal{T}^{[q+q']}$, obtained by the contraction of $\mathcal{T}^{[q]}$ and $\mathcal{T}^{[q']}$ over link $\eta$, invariant. Since the QMB state $|\Psi\rangle$ depends on $\mathcal{T}^{[q]}$ and $\mathcal{T}^{[q']}$ only via $\mathcal{T}^{[q+q']}$, it is insensitive to this transformation. Similarly, the network geometry is unchanged (as long as $Y$ acts on a single auxiliary link). Such an operation is sketched using the diagrammatic language in Fig. 3.

Combining multiple (single-link) gauge transformations as in Eq. (3), even on different links, leads to a larger class of transformations, which still leave the quantum many-body state invariant, and can be performed efficiently numerically. In fact, these algebraic manipulations turn out to be very useful tools when designing computational algorithms which operate on TN states. In many architectures (most notably in DMRG/MPS) adapting the gauge during runtime is a fundamental step for achieving speed-up and enhanced precision.

We will review linear algebra methods which implement the gauge transformations in

Sec. 2, and their symmetric tensor counterparts in Sec. 3. We will explore the advantages of employing gauge conditions in Sec. 4 and Sec. 5.

# 2 Tensors and Tensor Operations

Tensor networks organize numerical data in the form of interconnected tensors. In practice, these tensors are represented by floating-point numerical data in the computer memory (i.e. RAM or hard drive); they should be stored and organized in a way as to efficiently perform linear algebra operations on them. This section formalizes the notion of a tensor and introduces a set of basic operations serving as the toolbox for designing various tensor network geometries and respective algorithms.

In the following, we first introduce tensors formally and graphically (Sec. 2.1). We then present operations acting on these tensors with technical details and extend the graphical notation (Sec. 2.2).

## 2.1 Tensors

We define a tensor $\mathcal{T}$ with $n$ links ($n \in \mathbb{N}_0$) as an $n$-dimensional array $\mathcal{T}_{i_1 i_2 \dots i_n}$ of complex valued elements. It can be seen as a mapping

$$\mathcal{T} : \mathbb{I}_1 \times \mathbb{I}_2 \times \cdots \times \mathbb{I}_n \to \mathbb{C} : (i_1, i_2, \dots, i_n) \mapsto \mathcal{T}_{i_1 i_2 \dots i_n}, \tag{4}$$

where each complex scalar element $\mathcal{T}_{i_1 i_2 \dots i_n}$ is accessed by a tuple (an ordered integer list) of $n$ indices $i_1, i_2, \dots, i_n$. The index $i_r$ at position $r$ takes values in the index set $\mathbb{I}_r = \{1, \dots, d_r\} \subset \mathbb{N}$ and is referred to as the $r$-th *link* of the tensor. For numerical purposes, the *link dimension* $d_r := \#\mathbb{I}_r$ is always finite. Tensors and links allow intuitive graphical representations as network nodes and network edges respectively (see Fig. 4). We do not need to formally distinguish virtual and physical links at the level of basic operations for links and tensors. The distinction between virtual and physical links is instead treated at the level of the network, as we will see later in Sec. 4.

A zero-link tensor is a complex scalar. A one-link tensor of dimension $d_1$ is a $d_1$-dimensional complex vector. A two-link tensor of dimensions $d_1 \times d_2$ is a (rectangular) complex matrix with $d_1$ rows and $d_2$ columns, and so forth. A detailed example of a three-link tensor is given in Fig. 5. In general, the $n$-link tensor is an element of the complex space $\mathbb{C}^{d_1 \times \cdots \times d_n}$ with $\#\mathcal{T} := \dim(\mathcal{T}) = \prod_r d_r$ independent components. We notice that in certain cases, e.g. for problems with time-reversal symmetric Hamiltonians, the tensor elements can be restricted to real values (see also Appendix A).

For our following discussion, we assume, in accordance with standard programming, that tensor elements are stored in a contiguous section of linearly addressed computer memory (array). Specifically, each element $\mathcal{T}_{i_1 i_2 \dots i_n}$ can be stored and addressed in such a linear array by assigning a unique integer index (address offset) to it (see an example in Fig. 5). A simple way to implement this is by the assignment

$$\text{offset}(i_1, i_2, \dots, i_n) := \sum_{r=1}^{n} (i_r - 1) \prod_{k=1}^{r-1} d_k, \tag{5}$$

which is known as "column-major" ordering. There are, indeed, other models for storing elements. For efficient coding, one should consider the respective advantages of the ordering method (many programming languages and compilers use an intrinsic row major ordering, after all). In Sec. 3, we extend the definition of the tensor object: We will also consider tensors with additional inner structure (given by symmetry), generalizing diagonal or block-diagonal matrices.

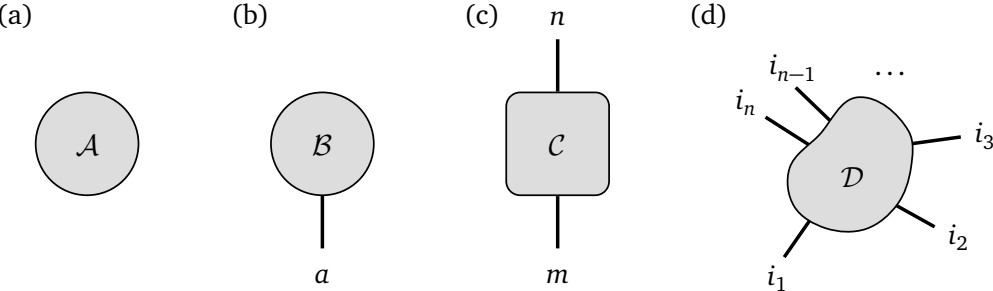

Figure 4: Graphical representation of (a) a zero-link tensor (i.e. a scalar), (b) a one-link tensor (a vector), (c) a two-link tensor (a matrix) and (d) a tensor with an arbitrary number $n$ of links. Indices of a tensor (i.e. its links) can be denoted with arbitrary symbols; the ordering of the open link-ends reflects the order of the indices in the tensor.

| offset | $\mathcal{T}_{i_1 i_2 i_3}$ | $i_1$ | $i_2$ | $i_3$ | $a$ | $b$ | $j$ |
|---|---|---|---|---|---|---|---|
| 0 | $1/\sqrt{2}$ | 1 | 1 | 1 | 1 | 1 | 1 |
| 1 | 0 | 2 | 1 | 1 | 2 | 1 | 2 |
| 2 | 0 | 3 | 1 | 1 | 3 | 1 | 3 |
| 3 | 0 | 1 | 2 | 1 | 1 | 2 | 4 |
| 4 | 0 | 2 | 2 | 1 | 2 | 2 | 5 |
| 5 | $\sqrt{2}/\sqrt{3}$ | 3 | 2 | 1 | 3 | 2 | 6 |
| 6 | 0 | 1 | 1 | 2 | 1 | 3 | 7 |
| 7 | 0 | 2 | 1 | 2 | 2 | 3 | 8 |
| 8 | $1/\sqrt{3}$ | 3 | 1 | 2 | 3 | 3 | 9 |
| 9 | $-1/\sqrt{2}$ | 1 | 2 | 2 | 1 | 4 | 10 |
| 10 | 1 | 2 | 2 | 2 | 2 | 4 | 11 |
| 11 | 0 | 3 | 2 | 2 | 3 | 4 | 12 |

(a)

$$\mathcal{V}_j = \begin{pmatrix} 1/\sqrt{2} \\ 0 \\ 0 \\ 0 \\ 0 \\ \sqrt{2}/\sqrt{3} \\ 0 \\ 0 \\ 1/\sqrt{3} \\ -1/\sqrt{2} \\ 1 \\ 0 \end{pmatrix}$$

(c)

$$\mathcal{M}_{ab} = \begin{pmatrix} 1/\sqrt{2} & 0 & 0 & -1/\sqrt{2} \\ 0 & 0 & 0 & 1 \\ 0 & \sqrt{2}/\sqrt{3} & 1/\sqrt{3} & 0 \end{pmatrix}$$

(b)

(d)

Figure 5: Example of a real-valued three-link tensor $\mathcal{T}_{i_1 i_2 i_3} \in \mathbb{R}$ of dimensions $d_1 = 3$, $d_2 = d_3 = 2$. (a) In the table we list all tensor elements and their indices (in column-major order) using the ordering (offset) from Eq. (5). In the additional columns to the right we also added the respective indices after the tensor is reshaped into a matrix and a vector (see *fusing* in Sec. 2.2.1). (b) The matrix $\mathcal{M}_{ab} = \mathcal{T}_{i_1 (i_2 i_3)}$, has row- and column-indices $a = i_1$ and $b = \text{fuse}(i_2, i_3)$ of dimensions $d_a = d_1 = 3$ and $d_b = d_2 d_3 = 4$. (c) The vector $\mathcal{V}_j = \mathcal{T}_{(i_1 i_2 i_3)}$ has a combined index $j = \text{fuse}(i_1, i_2, i_3)$, according to Eq. (8), of total dimension $d_j = \#\mathcal{T} = 12$. (d) A graphical representation of the three-link tensor. The tensor $\mathcal{T}$ could e.g. encode the three normalized states $|\Psi_{i_1}\rangle = \sum_{i_2, i_3} \mathcal{T}_{i_1 i_2 i_3} |i_2 i_3\rangle$, i.e. $|\Psi_1\rangle = 1/\sqrt{2}(|11\rangle - |22\rangle)$, $|\Psi_2\rangle = |22\rangle$ and $|\Psi_3\rangle = 1/\sqrt{3}(|21\rangle + \sqrt{2}|12\rangle)$, spanning a subspace in the two-qubit Hilbert space $\{|i_2, i_3\rangle\}$. Their expansions can be read from the rows of $\mathcal{M}$, with columns corresponding to $(i_2, i_3) = (1,1), (2,1), (1,2), (2,2)$ from left to right.

## 2.2   Basic Tensor Operations

All tensor network algorithms involve certain basic operations on the tensors, ranging from initialization routines, element-wise operations such as multiplication by a scalar, complex conjugation $\mathcal{T}^*$, or tensor addition and subtraction, to very general unary or binary operations [135]. This section presents selected examples from two important categories of tensor operations, namely basic *manipulations*, and more involved *linear algebra operations* (see e.g. Fig. 6). Tensor manipulations usually operate on the elements or the shape of a single tensor, including position, number and dimensions of the links. They often reallocate and reshuffle data within the computational memory, but generally require no FLoating point OPerations (FLOPs). On the other hand, linear algebra operations such as contractions and decompositions require FLOPs, such as complex addition and multiplication.

Tensor network algorithms, operating extensively on the level of tensors, will spend the majority of their runtime with basic tensor operations. In numerical implementations, it is therefore important that these operations are highly optimized and/or rely on robust and efficient linear algebra libraries such as the *Linear Algebra PACKage* (LAPACK) [136], and the *Basic Linear Algebra Subprograms* (BLAS) [137–139]. Here we will not enter such level of detail and instead outline the algorithms and give estimates on overall computational cost and scaling behavior. Nevertheless, our advice, for readers who want to program tensor network algorithms, is to dedicate specific functions and subroutines to each of the following operations. It is worth mentioning that there are various open-source numerical libraries which handle tensor manipulations with similar interfaces, such as the iTensor [140], Uni10 [141] or the TNT library [142].

### 2.2.1   Tensor Manipulations

**Link Permutation.**   Reordering the link positions $i_1, \ldots, i_n$ of a tensor $\mathcal{T}$ by moving a link at position $l$ in tensor $\mathcal{T}$ to a position $k = \sigma(l)$ in the new (permuted) tensor $\mathcal{T}^p$:

$$\mathcal{T}^p_{i_1 \ldots i_n} = \mathcal{T}_{i_{\sigma(1)}, \ldots, i_{\sigma(n)}}. \tag{6}$$

As an example, imagine a four-link tensor and a permutation $\sigma(1) = 2$, $\sigma(2) = 3$, $\sigma(3) = 1$, $\sigma(4) = 4$. The tensor after permutation then has elements $\mathcal{T}^p_{c,a,b,d} = \mathcal{T}_{a,b,c,d}$ (see Fig. 6(a)). Note that our definition describes an *active* permutation, while other conventions prefer *passive* permutations where the roles of input- and output tensors are reversed. For tensors with two links, permutations correspond to matrix transpositions, $M^T_{ab} = M_{ba}$. Carrying out a permutation has a computational complexity bounded by $\mathcal{O}(\#\mathcal{T})$, i.e. linear in the number of elements that must be rearranged in memory.

**Link Fusion.**   Fusing links is an operation that combines $m - k + 1$ adjacent links at positions $k, \ldots, m$ into a single new *fused link* $j \sim (i_k, i_{k+1}, \ldots, i_m)$ which replaces the original links in the *fused tensor* (see Fig. 6(b)):

$$\mathcal{T}'_{i_1 \ldots i_{k-1} j \, i_{m+1} \ldots i_n} = \mathcal{T}_{i_1 \ldots i_{k-1} (i_k \ldots i_m) i_{m+1} \ldots i_n}. \tag{7}$$

The combined index $j$ takes values in the Cartesian product of all fused links $\mathbb{J} = \mathbb{I}_k \times \cdots \times \mathbb{I}_m$. Typically we use an integer enumeration

$$j = \mathrm{fuse}(i_k, i_{k+1}, \ldots, i_m) := 1 + \mathrm{offset}(i_k, i_{k+1}, \ldots, i_m) \tag{8}$$

within all possible combinations of the fused link indices. Fusing can be realized in negligible computational cost $\mathcal{O}(1)$ as with the proper enumeration no elements have to be moved in memory (cf. Eq. 5). As an important example, a tensor can be converted into a matrix by

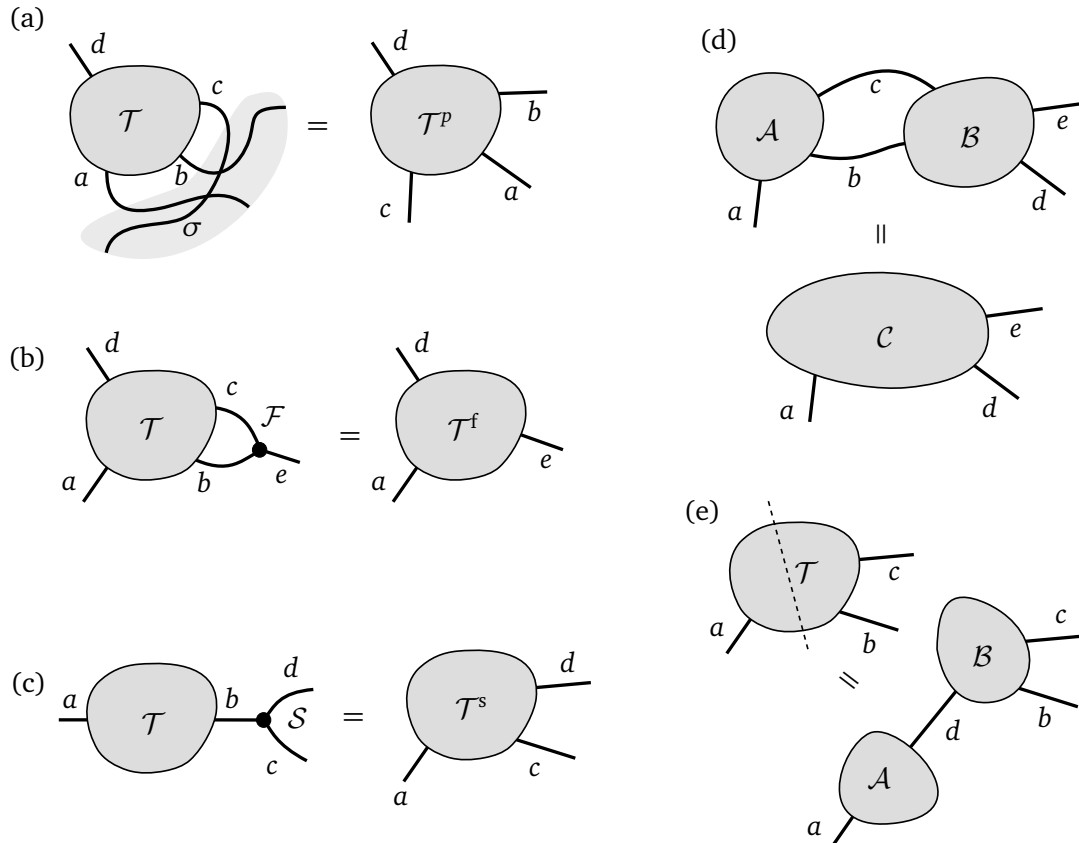

Figure 6: Basic tensor operations. Common TN operations can be separated into elemental steps including: (a) Link permutation. Graphically, the permutation of links of a tensor $\mathcal{T}$ is shown by intertwining the links ($\sigma$, shaded area) to obtain the new order in $\mathcal{T}^p$. (b) Link fusion: Two or more links of a tensor $\mathcal{T}$ are joined to form a single one in the new tensor $\mathcal{T}^f$. This can be seen as a contraction with a fuse-node $\mathcal{F}$ (see Sec. 2.2.2). (c) Link splitting: a single link of a tensor $\mathcal{T}$ is split into two or more resulting in a tensor $\mathcal{T}^s$. The splitting is equivalent to the contraction with a split-node $\mathcal{S}$. (d) Tensor contractions and (e) tensor decompositions.

simultaneously fusing both sides of the link-bipartition $1, \ldots, k-1$ and $k, \ldots, n$ into row and column indices $a$ and $b$ respectively (see also Fig. 5):

$$\mathcal{M}_{ab} = \mathcal{T}_{(i_1 \ldots i_{k-1})(i_k \ldots i_n)}. \tag{9}$$

Through link fusion, matrix operations become directly applicable to tensors with more than two links. Important examples include multiplication and factorization, which are discussed in Sec. 2.2.2. Fusing arbitrary combinations of non-adjacent tensor links can be handled by permuting the selected links into the required positions $k, \ldots, m$ first.

**Link Splitting.**     Splitting of a link at position $k$ of the original tensor $\mathcal{T}$ is the inverse operation to fusing. It produces a tensor $\mathcal{T}'$ with one original link $j_k$ replaced (via inversion of Eq. (8)) by $m - k + 1$ new links $i_k, \ldots, i_m$ of dimensions $d'_k, \ldots, d'_m$, their product being equal to the dimension $d_k$ of the original link,

$$\mathcal{T}'_{i_1 \ldots i_{k-1} (i_k \ldots i_m) i_{m+1} \ldots i_n} = \mathcal{T}_{i_1 \ldots i_{k-1} j_k i_{m+1} \ldots i_n}, \tag{10}$$

so that $j_k = \text{fuse}(i_k,\ldots,i_m)$ (Fig. 6(c)). This operation is uniquely defined once the split dimensions $d'_k,\ldots,d'_m$ are set. Note that similar to fusing, splitting operations can also be performed in non-scaling time $\mathcal{O}(1)$, but may require a link permutation after execution. A special case of splitting is the insertion of an extra one-dimensional virtual link (*dummy* link) at any position, which may be useful when a certain tensor shape must be matched. Imagine, for example, that some TN-algorithm interface formally asks for a matrix-product operator (see also Sec. 5.1), but all we have is a simple product of non-interacting local operators. Joining those by dummy links allows one to match the generic interface in an elegant manner. A similar situation arises when computing a direct product of two tensors, for which we could simply employ the contraction operation, given we join both tensors by a dummy link, as proposed in Sec. 2.2.2.

**Link Index Remapping: Compression, Inflation, and Reorder.** A typical class of shape operations on a tensor are those that rearrange the index values from one (or more) of the connected links. These link remapping manipulations either reorder the index values, or increase/decrease the link dimension itself. Compressing one or more links connected to a tensor $\mathcal{T}_{i_1\ldots i_n}$ means selecting a subset of their allowed indices and discarding the rest, thus actually reducing the link dimensions. Link compression is carried out extracting a subset of the linked tensor elements by means of *subtensor readout*

$$\mathcal{S}_{j_1\ldots j_n} := \mathcal{T}_{i_1\ldots i_n} \quad \text{for} \quad j_r \in \mathbb{J}_r \subseteq \mathbb{I}_r, \quad r \in \{1,\ldots,n\} \tag{11}$$

and we call $\mathcal{S}$ a *subtensor* of $\mathcal{T}$. The subtensor still has $n$ links, and each may have smaller dimensions, depending on the number $\#\mathbb{J}_r$ of indices kept per link. Subtensors arise naturally as a generalization of block-diagonal matrices for instance due to the consideration of symmetry constraints, as we will see in Sec. 3.5.1. More specifically, let a link index $i_r$ undergo a mapping

$$\mathfrak{M}_r : \mathbb{J}_r \rightarrow \mathbb{I}_r : j_r \mapsto i_r. \tag{12}$$

If $\mathfrak{M}_r$ is injective, the result is a subtensor of reduced link dimension(s). Furthermore, if $\mathfrak{M}_r$ is invertible, it is a permutation, which allows one to access elements in a different order. The latter is useful for sorting or aligning elements (e.g. after the fuse-operation of 2.2.1) into contiguous blocks for fast memory access. Finally, the inflation (padding) of a link corresponds to an increase of its dimension, and compression is its (left-) inverse operation. Inflation is performed by overwriting a subtensor $\mathcal{S}$ in a larger tensor holding the padded elements (see Sec. 5.1.3), by means of *subtensor assignment*

$$\mathcal{T}_{i_1\ldots i_n} := \mathcal{S}_{j_1\ldots j_n} \quad \text{for} \quad j_r \in \mathbb{J}_r \subseteq \mathbb{I}_r, \quad r \in \{1,\ldots,n\}. \tag{13}$$

The remaining elements in the larger tensor $\mathcal{T}$ must then be initialized by the user, e.g. set to zeros.

Manipulating, overwriting or extracting subtensors from or within larger tensors is an integral part of variational subspace expansion techniques [143, 144]. Further use cases for subtensor assignment and readout arise in the construction of MPOs [132] and other operations listed at the end of Sec. 2.2.2.

### 2.2.2 Tensor Linear Algebra Operations

**Tensor Contraction.** Tensor contractions are a generalization of matrix multiplications to arbitrary numbers of indices: Whenever two tensors $\mathcal{A}$ and $\mathcal{B}$ with links $\{a_{1\ldots n}\}$ and $\{b_{1\ldots m}\}$ of dimensions $\{d^a_{1\ldots n}\}$ and $\{d^b_{1\ldots n}\}$ have a subset of $k$ links in common — say $d^a_{n-k+j} = d^b_j =: d^s_j$ for $j = 1\ldots k$ — we can contract over these *shared links* $s_j$. The result is a *contracted tensor*

Figure 7: Intermediate steps in the contraction of two tensors, realized through matrix-multiplication. First, the input tensors $\mathcal{A}$ and $\mathcal{B}$ are brought into correct matrix-shape, which in general involves (a) permuting and (b) fusing their links. After their matrix-multiplication (c) is carried out, the remaining original links are restored into the expected order by a final (d) splitting and (e) permutation operation. Those permutation, fuse- and split-steps are very common among tensor-operations that rely on matrix-operations internally.

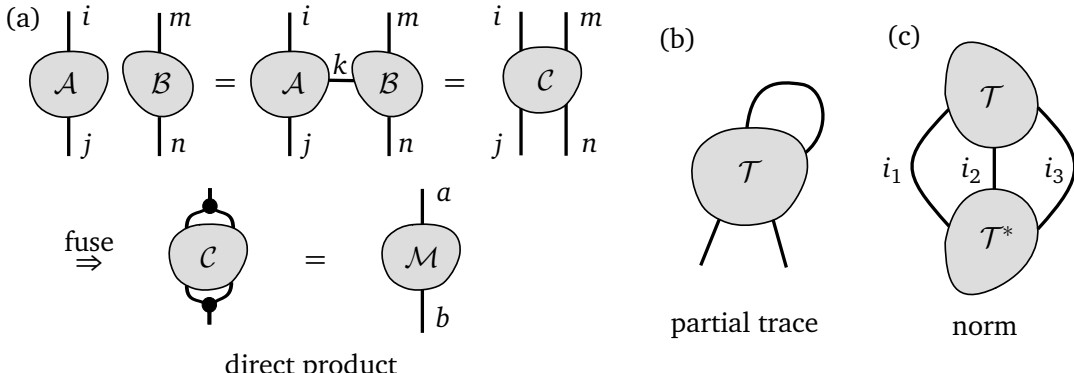

Figure 8: Variants of contractions. (a) Direct product of two matrices, or two-link tensors $\mathcal{A}_{ij}$ and $\mathcal{B}_{mn}$. By attaching a one-dimensional "dummy" link to both tensors, we can indicate the contraction $\mathcal{C}_{ijmn} = \mathcal{A}_{ij} \cdot \mathcal{B}_{mn}$. Formally, a subsequent (column-major, Eq. (8)) fusion over the permuted link-bipartition $\mathcal{M}_{ab} = \mathcal{C}_{(mi)(nj)}$ gives us the Kronecker product $\mathcal{M} = \mathcal{A} \otimes \mathcal{B}$. (b) The partial trace. (c) The Hilbert–Schmidt norm obtained by full contraction with the complex conjugate $\mathcal{T}^*$.

$\mathcal{C}$ spanned by the remaining $n + m - 2k$ links $\{a_{1...n-k}, b_{k+1...m}\}$. Graphically we represent a contraction by simply connecting the links to be contracted over (Fig. 6(d)). The elements of $\mathcal{C}$ are obtained by multiplying the respective elements of $\mathcal{A}$ and $\mathcal{B}$ and summing over the shared links

$$\mathcal{C}_{a_1...a_{n-k}\, b_{k+1}...b_m} = \sum_{s_1...s_k} \mathcal{A}_{a_1...a_{n-k}\, s_1...s_k} \mathcal{B}_{s_1...s_k\, b_{k+1}...b_m}. \tag{14}$$

Equation (14) can be cast into a matrix multiplication after fusing the shared links into a single link $s = \mathrm{fuse}(s_1, \ldots, s_k)$ of dimension $d_s = d_1^s \cdots d_k^s$ and the remaining links of tensors $\mathcal{A}$ and $\mathcal{B}$ into row- and column indices $a$ and $b$ of dimensions $d_a = d_1^a \cdots d_{n-k}^a$ and $d_b = d_{k+1}^b \cdots d_m^b$, respectively:

$$\mathcal{C}_{ab} = \sum_s \mathcal{A}_{as} \mathcal{B}_{sb}. \tag{15}$$

In the most common scenario for a tensor contraction, the links are not ordered as in Eq. (14). We then proceed as follows (see also Fig. 7): We first permute the links, so that tensors $\mathcal{A}$ and $\mathcal{B}$ appear as in Eq. (14). Then the link fusion is carried out, followed by the matrix multiplication as in Eq. (15). Finally we split the row and column links again and possibly permute a second time to obtain the desired link positions at $\mathcal{C}$.

The contraction can be a costly operation in terms of computational complexity. Carried out straightforwardly as written in Eq. (14), it scales with the product of the dimensions of all links involved (considering shared links only once) as $\mathcal{O}(d_a d_s d_b)$.

Implementing the tensor contraction via the matrix multiplication of Eq. (15) has a twofold advantage: It enables efficient parallelization [138, 139, 145] and it allows us to exploit optimized matrix multiplication methods [136, 137, 146]. As an example for the latter, the complexity of a contraction of square matrices can be improved from $\mathcal{O}(d^3)$ to $\mathcal{O}(d^{2.376})$ when using optimized linear algebra methods [147]. However, an overhead for the required permutations must be considered. A careful initial analysis of the arguments can often help to avoid some of the permutations, or to replace a permutation on a large input or output tensor at the expense of one or two additional permutations on smaller ones. Such optimization is possible by exploiting the full freedom we have in rearranging the internal order in which links

are fused into column- and row indices and also by utilizing implicitly transposed arguments available with many matrix-multiplication implementations.

**Special Cases of Contractions.** Some operations which are very common to tensor network architectures and algorithms can be regarded as special instances of tensor contractions. We list the most relevant ones in this paragraph.

(i) Fusing $k$ links of a tensor $\mathcal{T}$ can be achieved by contracting a sparse $k+1$ link tensor, the *fuse-node* $\mathcal{F} \equiv \mathcal{F}_{i_1 \ldots i_k, j}$ defined by

$$
\mathcal{F}_{i_1 \ldots i_k, j} = \begin{cases} 1 & \text{if } j = \text{fuse}(i_1, \ldots, i_k) \\ 0 & \text{otherwise} \end{cases}
\tag{16}
$$

over those links $i_1 \ldots i_k$ of $\mathcal{T}$ that shall be fused into $j$. Analogously, the split-operation can be carried out by contracting a corresponding *split-node* $\mathcal{S} := \mathcal{F}^\dagger$ over $j$, with $\mathcal{F}^\dagger_{j,(i_1 \ldots i_k)} = \mathcal{F}_{(i_1 \ldots i_k),j}$ in a matrix notation. In this formalism, it is clear that fusing and splitting are mutually inverse operations, i.e. $\mathcal{F}^\dagger \mathcal{F} \equiv \delta_{j,j'}$ and $\mathcal{F}\mathcal{F}^\dagger \equiv \prod_r \delta_{i_r,i'_r}$, since $\mathcal{F}$ is unitary by construction.

While possible, fusing or splitting by such a contraction is less efficient than the enumeration procedure of Sec. 2.2.1. Nevertheless, fuse-nodes play an important role in the context of symmetric tensors where fusing and splitting is carried out according to group representation theory (see Sec. 3). We also employ them to encode the fusion graphically, as in Fig. 6(b).

(ii) The Kronecker-product or matrix-direct product

$$
\mathcal{C}_{a_1 \ldots a_{n-k} \, b_{k+1} \ldots b_m} = \mathcal{A}_{a_1 \ldots a_{n-k}} \mathcal{B}_{b_{k+1} \ldots b_m}
\tag{17}
$$

corresponds to the case when no links are shared in the contraction of Eq. (14). This operation can be achieved (up to a permutation) by attaching a shared dummy link on both tensors and contracting over it, as visualized in Fig. 8(a) with a practical application in Fig. 11(b).

(iii) In analogy to the multiplication with a diagonal matrix, we can contract a "diagonal" two-link tensor $\mathcal{D}_{ij} = \delta_{i,j} \lambda_i$ over a single link into some other tensor $\mathcal{T}$. This operation can be carried out in time $\mathcal{O}(\#\mathcal{T})$ scaling linearly with the number of tensor elements. We represent those (often real) diagonal matrices or *weights* $\lambda_i$ over a link graphically by a thin bar as depicted in Fig. 10, emphasizing their sparsity.

(iv) The *(partial) trace* is a variant of the contraction, involving a single tensor only. It is depicted in Fig. 8(b). This operation sums up all tensor elements that share similar indices on $k$ specified link-pairs at the *same tensor*

$$
\text{tr}[\mathcal{T}]_{i_{2k+1} \ldots i_n} = \sum_{s_1 \ldots s_k} \mathcal{T}_{(s_1 \ldots s_k)(s_1 \ldots s_k) i_{2k+1} \ldots i_n},
\tag{18}
$$

where we contracted the first $k$ links at positions $1 \ldots k$ with the following $k$ links at positions $2k+1 \ldots 2k$. If the tensor $\mathcal{T}$ has $n$ links, the resulting tensor $\text{tr}[\mathcal{T}]$ has $n-2k$ links. If $k = n/2$, the tensor trace is equivalent to a matrix trace over the given bipartition of links. The complexity of the operation equals the product of dimensions of all paired links and all remaining links (if present), thus $\mathcal{O}(d_{s_1} \cdots d_{s_k} d_{i_{2k+1}} \cdots d_{i_n})$.

(v) A *tensor norm*, defined as a generalization of the Hilbert–Schmidt- or Frobenius-norm by

$$\|\mathcal{T}\| = \sqrt{\sum_{i_1 \dots i_n} \mathcal{T}_{i_1 \dots i_n} \mathcal{T}^*_{i_1 \dots i_n}} \tag{19}$$

can be computed in $\mathcal{O}(\#\mathcal{T})$. As shown in Fig. 8(c), this is equivalent to the contraction of $\mathcal{T}$ with its element-wise complex conjugate $\mathcal{T}^*$ over all links.

(vi) In principle, we can also regard *permutation* as a contraction

$$\mathcal{T}^p_{j_1 \dots j_n} = \mathcal{T}_{i_1 \dots i_n} \cdot \sigma_{i_1 \dots i_n, j_1 \dots j_n}, \tag{20}$$

where $\sigma_{i_1 \dots i_n, j_1 \dots j_n} = \delta_{i_1 j_{\sigma(1)}} \cdot \dots \cdot \delta_{i_n j_{\sigma(n)}}$ (see Fig. 6(a)). However, this is not the most efficient numerical approach.

As a general remark, we stress that finding the optimal strategy to contract multiple tensors efficiently is usually a difficult problem [148]; nevertheless it is a fundamental issue, as contractions are often the bottleneck of the computational costs. This question ultimately encourages the usage of specific network geometries whose optimal contraction strategies are known (see e.g. Ref. [149]). Such geometries will be further discussed in Sec. 5.

**Tensor Decompositions.**    A tensor decomposition takes a single tensor as input, and replaces it with two or more output tensors (see Fig. 6(e)). For every link of the input tensor, there will be a similar link attached to one of the output tensors. Additional "internal" links may connect the output tensors among each other.

For a tensor decomposition to be meaningful, we demand that the contraction of those output tensors over their shared internal links restores the input tensor within a certain precision. The simplest example for a tensor decomposition is the inversion of a contraction: we take the contracted tensor as input and replace it by the tensors that we originally contracted. In practice, however, we are more interested in special decompositions that equip the output tensors with favorable properties, such as an advantageously small size or certain isometries that come in handy when working with larger tensor networks (see Sec. 4).

The most notable decompositions are the QR- and the singular value decomposition (SVD), both instances of so-called *matrix factorizations* [150]. As a recipe for such an (exact) tensor decomposition, first choose a bipartition of the tensor links $i_1, \dots, i_r$ and $i_{r+1}, \dots, i_n$ of an $n$-link tensor and fuse them into single row and column indices $a$ and $b$ respectively. If a different combination of fused links is required, we apply a permutation beforehand. We then decompose the matrix

$$\mathcal{T}_{(i_1 \dots i_r)(i_{r+1} \dots i_n)} = \mathcal{T}_{ab} \tag{21}$$

into factor matrices $\mathcal{A}_{ak}$, $\mathcal{B}_{kb}$, and possibly a diagonal matrix $\lambda_k$, such that the matrix multiplication yields

$$\sum_{k=1}^{d_k} \mathcal{A}_{ak} \lambda_k \mathcal{B}_{kb} = \mathcal{T}_{ab}. \tag{22}$$

Finally, we split the links we initially fused to restore the outer network geometry. Thus we obtained tensors that contract back to $\mathcal{T}$.

**QR Decomposition.**    Following the procedure of Eqs. (21) and (22), the QR decomposition splits the tensor $\mathcal{T}$ into two tensors $\mathcal{Q}$ and $\mathcal{R}$ such that

$$\mathcal{T}_{(i_1 \dots i_r)(i_{r+1} \dots i_n)} \overset{\text{fuse}}{=} \mathcal{T}_{ab} \overset{\text{QR}}{=} \sum_{k=1}^{d_k} \mathcal{Q}_{ak} \mathcal{R}_{kb} \overset{\text{split}}{=} \sum_{k=1}^{d_k} \mathcal{Q}_{(i_1 \dots i_r)k} \mathcal{R}_{k(i_{r+1} \dots i_n)}, \tag{23}$$

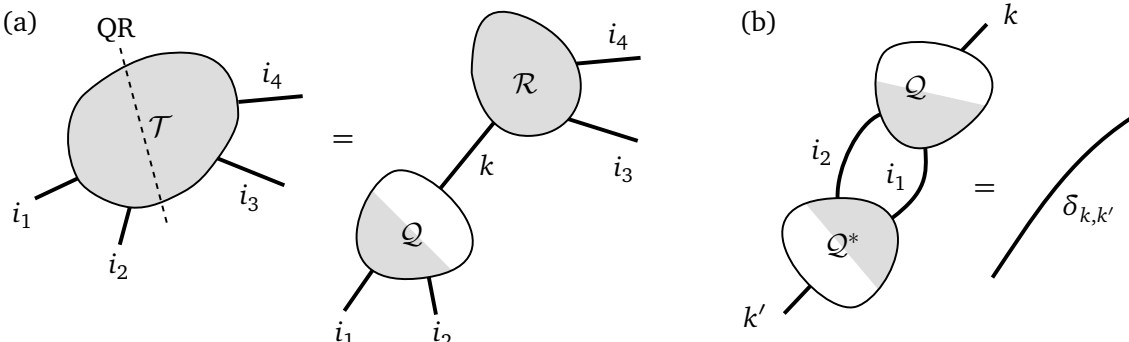

Figure 9: QR decomposition and isometry. (a) We split $\mathcal{T}$ at the dashed line into $\mathcal{R}$ and an isometry $\mathcal{Q}$ via QR-decomposition. (b) The isometry is depicted by a gray-white filling: the gray partition indicates the links over which we have unitarity. In other words, the link index $k$ at the opposite partition enumerates an orthonormal set of $d_k$ vectors $\left(q^k\right)_{(i_1\dots i_r)} = \mathcal{Q}_{k(i_1\dots i_r)}$.

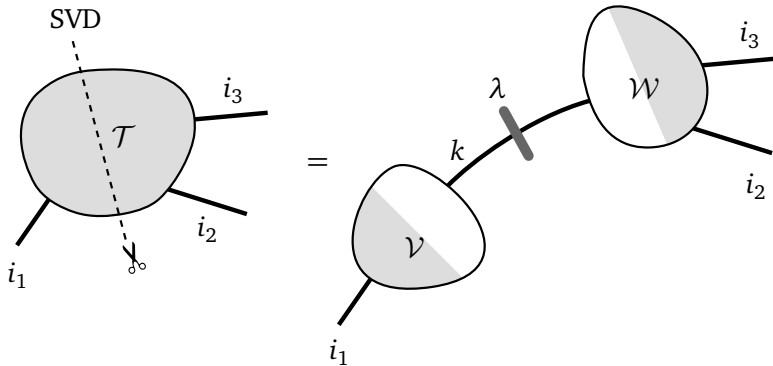

Figure 10: The singular value decomposition of a tensor produces an intermediate link $k$ and two semi-unitary tensors $\mathcal{W}$ and $\mathcal{V}$ (depicted by shade over the orthonormal link bipartition, cf. Eq. (25)). The singular values appear on the intermediate link as a non-negative diagonal matrix or weights $\lambda_k$, represented by a bar to highlight the sparsity. Removing singular values from the link (thereby *compressing* it into a smaller dimension $d_k$) does not disturb the tensor isometries.

where $\mathcal{R}_{kb}$ is an upper-trapezoidal matrix and $\mathcal{Q}_{ak}$ is a (semi-)unitary matrix with respect to $k$ (or *left isometry*), meaning that it satisfies $\sum_a \mathcal{Q}_{ak}\mathcal{Q}^*_{ak'} = \delta_{k,k'}$ (as shown in Fig. 9). Finally, $\mathcal{T}$ is equal to the contraction of $\mathcal{Q}$ and $\mathcal{R}$ over the intermediate link $k$, which has the minimal dimension

$$d_k = \min\{d_a, d_b\} = \min\{d_1 \cdot \dots \cdot d_r, d_{r+1} \cdot \dots \cdot d_n\}. \tag{24}$$

**Singular Value Decomposition.** The SVD is defined as

$$\mathcal{T}_{(i_1\dots i_r)(i_{r+1}\dots i_n)} \overset{\text{fuse}}{=} \mathcal{T}_{ab} \overset{\text{SVD}}{=} \sum_{k=1}^{d_k} \mathcal{V}_{ak}\lambda_k\mathcal{W}_{kb} \overset{\text{split}}{=} \sum_{k=1}^{d_k} \mathcal{V}_{(i_1\dots i_r)k}\lambda_k\mathcal{W}_{k(i_{r+1}\dots i_n)}, \tag{25}$$

see also Fig. 10. It yields a set of non-negative, real valued singular values $\{\lambda_k\}$ and (contrary to the QR) *two* semi-unitary (or isometric) matrices $\mathcal{V}$ and $\mathcal{W}$. Semi-unitarity means that the tensors fulfill $\mathcal{V}^\dagger\mathcal{V} = \mathbb{1}$ and $\mathcal{W}\mathcal{W}^\dagger = \mathbb{1}$, while this does not necessarily hold for $\mathcal{V}\mathcal{V}^\dagger$ and $\mathcal{W}^\dagger\mathcal{W}$. The singular values can be organized in a diagonal matrix (as we saw in Sec. 2.2.2) and are

(a)

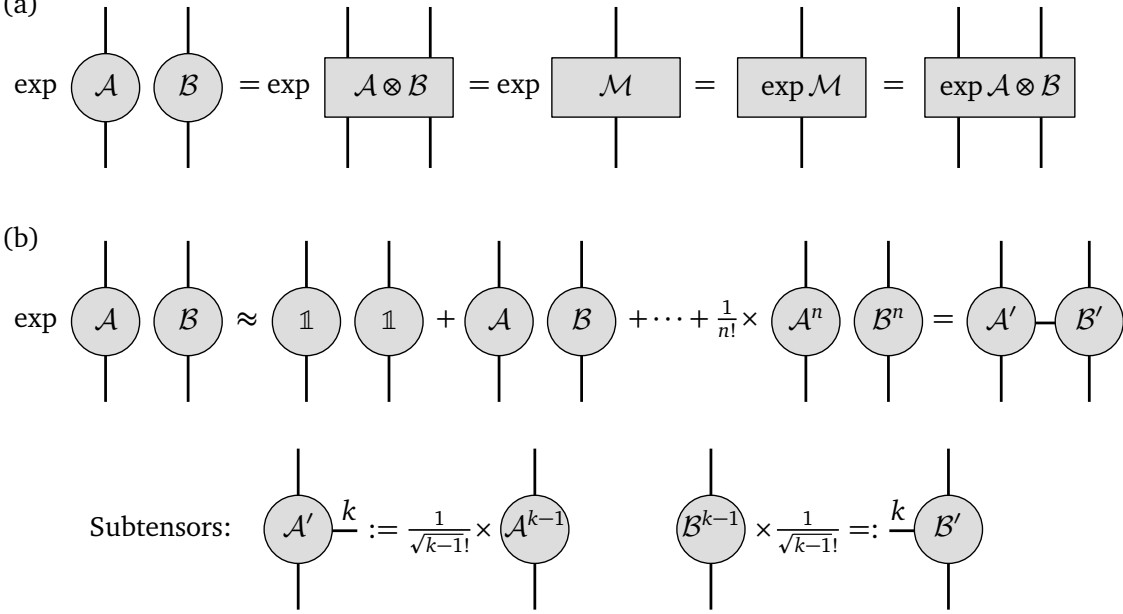

(b)

Figure 11: Two strategies for exponentiating a Kronecker product of two tensors $\mathcal{A} \otimes \mathcal{B}$ (i.e. hosting the matrix elements of two local operations on a lattice). Both demonstrate the power of combining several basic tensor operations with a unified graphical notation. (a) The exact exponential $\exp\{\mathcal{A} \otimes \mathcal{B}\}$ is formed from first expanding the Kronecker product to a full four-link tensor (by contracting over an inserted dummy link, cf. Fig. 8 (a)). Fused into a square matrix $\mathcal{M}$, this tensor can be exponentiated with matrix numerical routines. Finally, we split the fused links into the original configuration. (b) The Taylor-series expansion up to (small) order $n$ can be rewritten as a contraction of smaller tensors $\mathcal{A}'$ and $\mathcal{B}'$ over a third, shared link $k$ of dimension $d_k = n + 1$. To this end, scaled powers of $\mathcal{A}$, $\mathcal{B}$ are inscribed as subtensors into $\mathcal{A}'$, $\mathcal{B}'$ at the respective index values of $k$, as depicted. Note that in practice, conditions on numerical stability apply.

basically pure weight factors on the intermediate link. Conventional algorithms on QR- and singular value decomposition display an algorithm complexity of $\mathcal{O}\left(d_{\max} d_k^2\right)$ [151], with

$$d_{\max} = \max\{d_1 \cdot \ldots \cdot d_r, d_{r+1} \cdot \ldots \cdot d_n\} \tag{26}$$

being the maximum dimension of the fused links.

Even though the SVD carries a computational cost overhead with respect to the QR decomposition (a direct comparison is shown in Fig. 48), it is often useful to have both available, since the SVD has several advantages:

(i) In common tensor network algorithms, the singular values reveal entanglement properties. This is true, in particular, for loop-free TNs (see Sec. 4).

(ii) Singular values equal to numerical zeros can be removed from the intermediate link. The dimension $d_k$ of the latter is then reduced to the actual matrix rank of $\mathcal{T}_{ab}$, and the factors $\mathcal{V}_{ak}$, $\mathcal{W}_{kb}$ become full rank matrices which can be extracted via subtensor readout (c.f. Eq. (11)) from the retained intermediate link indices.

(iii) Small singular values can be approximated with zeros, effectively further reducing the link dimension $d_k$. This truncation, basically equivalent to a link compression as we discussed in Sec. 2.2.1, usually generates an error on the "compressed" state, but in most

tensor network geometries the discarded values can be used to estimate an upper error bound (see e.g. Ref. [39]).

(iv) Both output tensors are semi-unitary (isometries) with respect to $k$, regardless of the amount of singular values kept. This may prove advantageous in establishing specific tensor network gauges, as we will see in Sec. 4.

By discarding negligible singular values (compressing the tensor network) we relax Eq. (25) to an approximated equality, but it can be motivated physically and has a significant impact on performance of TN algorithms (see Sec. 5). If the number of singular values hereby removed from the intermediate link becomes large compared to $d_k$, one can in addition take advantage of efficient approximate algorithms for matrix-factorizations [152–156], some of which offer good control over the errors arising both from the stochastic method itself and from the discarded singular values.

**Other Operations.**   Other relevant linear algebra operations are those restricted to square matrices, such as *diagonalization*, which is typically carried out either exactly [136] or via approximate methods such as Lanczos [157], Jacobi–Davidson [158], or Arnoldi [159], as well as *exponentiation*, usually performed via Taylor or Padé expansion [160] or using the EXPOKIT package [161]. Note that tensor operations, especially when implemented with unified and compatible interfaces, are a powerful and flexible numerical toolbox. For example, in Fig. 11 we combine several basic operations to exponentiate local interaction operators, a central task in time-evolution algorithms like TEBD [48].

# 3   Abelian Symmetric Tensors and Operations

Symmetries are ubiquitous in nature, and both in classical and quantum mechanics they provide an essential advantage by reducing the full problem into a set of decoupled, simpler sub-problems: Symmetries are generally expressed in terms of conserved quantities (Noether's theorem); in quantum systems, such conserved quantities are identified by operators commuting with the Hamiltonian. In forming a multiplicative group, all these operators can then be block-diagonalized simultaneously, along with the Hamiltonian.

In numerical contexts, the role of symmetries is twofold: they provide a substantial computational speed-up, and they allow for precise targeting of symmetry sectors. More precisely, one can study any variational problem by restricting only to those states having a well-defined quantum number. This is a valid, robust, more direct approach than using a "grand canonical" framework (e.g. selecting the average number of particles by tuning the chemical potential), which is computationally more expensive as it requires to consider multiple symmetry sectors at once.

Here, we first review the quantum framework for Abelian symmetries and provide examples of the Abelian groups appearing in typical problems (Sec. 3.1). We then discuss symmetries that can be encoded in tensor networks (Sec. 3.2). In the rest of this section, we upgrade the tensor data structures and operations introduced in the previous section to include Abelian symmetry content. We discuss, in particular, how to organize the symmetry information in the tensors (Sec. 3.3) and how networks built from such tensors display the same symmetry properties (Sec. 3.4). Finally, we review how tensor operations benefit from the inclusion of symmetry information (Sec. 3.5).

## 3.1 Quantum Symmetries in a Nutshell: Abelian Groups

In quantum mechanics, a symmetry is a typically unitary representation $U(g)$ of a group $\mathcal{G}$ of transformations protected by the dynamics, i.e. commuting with the Hamiltonian $\mathcal{H}$: $[U(g),\mathcal{H}] = 0 \,\forall\, g \in \mathcal{G}$ (we briefly discuss anti-unitary representations in Appendix A). As a consequence, there exists an eigenbasis of the Hamiltonian such that each energy eigenstate belongs to a single irreducible representation (irrep) subspace.

Specifically, symmetric eigenstates will be of the form $|\Psi_{\ell,m,\partial}\rangle$ and satisfy the relation $\mathcal{H}|\Psi_{\ell,m,\partial}\rangle = E_{\ell,\partial}|\Psi_{\ell,m,\partial}\rangle$ where $\ell$ labels the irrep subspace, and is often referred to as *sector*, or *charge*, or (primary) quantum number. The index $m$ distinguishes states within the same irrep subspace (secondary quantum number), and the energy level $E_{\ell,\partial}$ is independent of $m$ as a consequence of Schur's lemma. Finally, $\partial$ contains all the residual information not classified by the symmetry (symmetry degeneracy label). In this sense, the symmetry group acts as $U(g)|\Psi_{\ell,m,\partial}\rangle = \sum_{m'}|\Psi_{\ell,m',\partial}\rangle W^{[\ell]}_{m,m'}(g)$ where $W^{[\ell]}(g)$ is the $\ell$-sector irrep matrix associated to the group element $g$. Symmetries in quantum mechanics can be either Abelian (when $[U(g),U(g')] = 0 \,\forall\, g,g'$) or non-Abelian (otherwise). From a computational perspective, the former are typically easier to handle as the related Clebsch–Gordan coefficients (*fusion rules* of the group representation theory) become simple Kronecker deltas [162]. In tensor network contexts, handling fusion rules is fundamental (see Sec. 3.5), thus addressing non-Abelian symmetries is somehow cumbersome (although definitely possible [163–166]), requiring the user to keep track of Clebsch–Gordan coefficients for arbitrary groups. For this reason, in this anthology we focus on addressing Abelian symmetry groups: Even when simulating a non-Abelian symmetric model one can always choose to address an Abelian subgroup [167], gaining partial speed-up (in this case, unpredicted degeneracies in the energy levels are expected to occur). Some examples of Abelian reduction strategies are mentioned later in this section.

Abelian symmetries have one-dimensional unitary (complex) irreps. Such irreps take the form of phase factors $W^{[\ell]}(g) = e^{i\varphi_\ell(g)}$, where the phase $\varphi$ depends on the quantum number $\ell$ and group element $g$, while no secondary quantum numbers $m$ arise. Then, the symmetric Hamiltonian is simply block-diagonal in the sectors $\ell$ and has no additional constraints.

In the following we present the Abelian symmetry groups commonly encountered in quantum mechanical problems, and we list irreps, group operations and fusion rules (see Eq. (32)) for each of them. Additionally, for later use, we introduce the irrep inversion $\ell \to \ell^\dagger$ as the mapping to the inverse (or adjoint) irrep, defined by $\varphi_{\ell^\dagger}(g) = -\varphi_\ell(g) \,\forall\, g \in \mathcal{G}$. It will prove useful that inversion is distributive under fusion, $(\ell \oplus \ell')^\dagger = \ell^\dagger \oplus \ell'^\dagger$. We will also generally write for the identical irrep $\ell_{\text{ident}} = 0$. This irrep exists for every group, it fulfills $\varphi_0(g) = 0 \,\forall\, g \in \mathcal{G}$ and is self-adjoint.

(i) $\mathbb{Z}_n$ **discrete planar rotation group**, also known as group of the oriented regular polygon with $n$ edges. This is the cyclic Abelian group with $n$ elements, and satisfies $g^n = 0$ (the identity group element) for any element $g \in \mathbb{Z}_n$. The parity group $\mathbb{Z}_2$ is a particularly relevant case.
*Group elements:* $g \in \{0 \ldots n-1\}$
*Group operation:* $g \circ g' = (g + g') \bmod n$
*Irrep labels:* $\ell \in \{0 \ldots n-1\}$
*Irrep phases:* $\varphi_\ell(g) = 2\pi g\ell/n$
*Inverse irrep:* $\ell^\dagger = (-\ell) \bmod n := (n-\ell) \bmod n$
*Fusion rule:* $\ell \oplus \ell' = (\ell + \ell') \bmod n$
*Examples:* Ising model ($\mathbb{Z}_2$); $q$-states chiral Potts model ($\mathbb{Z}_q$) [168].

(ii) U(1) **continuous planar rotation group**, or group of the oriented circle. This is the symmetry group related to the conservation laws of integer quantities, such as total

particle number ($\ell$ is this number), an extremely common scenario in many-body physics.
*Group element parameter:* $g \in [0, 2\pi)$
*Group operation:* $g \circ g' = (g + g') \bmod 2\pi$
*Irrep labels:* $\ell \in \mathbb{Z}$
*Irrep phases:* $\varphi_\ell(g) = g\ell$
*Inverse irrep:* $\ell^\dagger = -\ell$
*Fusion rule:* $\ell \oplus \ell' = \ell + \ell'$
*Examples:* XXZ model with field along z-axis (symmetry under any z-axis rotation, applied to every site); spinless Bose–Hubbard model (total boson number conservation).

(iii) $\mathbb{Z}$ **infinite cyclic group**, or the group of integers under addition. This group is connected to the conservation law of a quantity within a real interval, usually $[0, 2\pi)$ (such as the Bloch wave-vector conservation under translation of a lattice constant in a periodic potential problem).
*Group element parameter:* $g \in \mathbb{Z}$
*Group operation:* $g \circ g' = g + g'$
*Irrep labels:* $\ell \in [0, 2\pi)$
*Irrep phases:* $\varphi_\ell(g) = g\ell$
*Inverse irrep:* $\ell^\dagger = (2\pi - \ell) \bmod 2\pi$
*Fusion rule:* $\ell \oplus \ell' = (\ell + \ell') \bmod 2\pi$
*Examples:* Discrete translations on an infinite 1D lattice.

(iv) **Multiple independent symmetry groups.** For a model exhibiting multiple individual Abelian symmetries, which are independent (i.e. mutually commuting and having only the identity element of the composite symmetry in common), the composite symmetry group is given by the direct product of the individual symmetry groups, and is Abelian. Typical composite groups are combinations of the aforementioned basic groups ($\mathbb{Z}_n, \mathrm{U}(1), \mathbb{Z}$). Here we provide a symmetry combination example built from two individual symmetries, generalization to more symmetries being straightforward. In this scenario, one can select the irrep label $\ell_k$ for each elementary subgroup $k$ (or subsymmetry), making the composite irrep label actually an ordered tuple of scalar labels, which fuse independently.
*Group elements parameter:* $g_1 g_2 = g_2 g_1$, with $g_1 \in \mathcal{G}_1$ and $g_2 \in \mathcal{G}_2$.
*Irrep labels:* $\ell = \left(\ell^{[1]}, \ell^{[2]}\right)$, every element in this ordered tuple is an allowed scalar irrep label of the corresponding elementary symmetry group, with $\ell_{\text{ident}} = (0, 0)$.
*Irrep phases:* $\varphi_\ell(g) = \varphi_{\ell^{[1]}}(g) + \varphi_{\ell^{[2]}}(g)$
*Inverse irrep:* $\ell^\dagger = \left(\ell^{[1]\dagger}, \ell^{[2]\dagger}\right)$
*Fusion rule:* $\ell \oplus \ell' = \left(\ell^{[1]} \oplus_1 \ell^{[1]\prime}, \ell^{[2]} \oplus_2 \ell^{[2]\prime}\right)$, where $\oplus_k$ is the fusion rule of subgroup $k$.
*Examples:* XYZ model for an even number of sites: $\mathbb{Z}_2 \times \mathbb{Z}_2 = \mathrm{D}_2$ ($\pi$-rotation along any of the three Cartesian axes, applied at every site). $\mathrm{D}_2$ is the simplest noncyclic Abelian group, also known as the group of the rectangle, or Klein four-group.

Regarding the reduction of non-Abelian symmetry groups into their largest Abelian component, typical examples follow. The full regular polygon group $\mathrm{D}_n$ (present in the standard Potts model) is often reduced to $\mathbb{Z}_n$: the rotations are kept, the reflections are discarded. The unitary group of degree $N$, $\mathrm{U}(N)$, is often reduced to $\mathrm{U}(1)^{\times N}$ by considering only the diagonal unitary matrices, which are defined by $N$ independent angles. Similarly, $\mathrm{SU}(N)$ is usually reduced to $\mathrm{U}(1)^{\times(N-1)}$ ($N$ angles with 1 constraint). Abelian reduction can be often performed naively, even for more complex symmetry groups: For instance, the symmetry group of the spin-1/2 Fermi–Hubbard model, which is $\mathrm{SO}(4)$ ($\cong \mathrm{SU}(2) \times \mathrm{SU}(2)/\mathbb{Z}_2$) [169], is often treated just as a $\mathrm{U}(1) \times \mathrm{U}(1)$ symmetry (particle conservation for each spin species).

## 3.2   Encodable Symmetries in Tensor Networks

Not every symmetry (regardless of its group) can be efficiently encoded in the tensor network formalism: Depending on how a symmetry transformation acts on the QMB system it may be suitable or not for TN embedding. Symmetries that are known to be particularly apt for implementation in tensor networks [135,165,170,171] are those which *transform independently each degree of freedom without making them interact*, namely those satisfying $U(g) = \bigotimes_j V_j(g)$, where $V_j(g)$ is the local representation of group element $g$ at site $j$. $V_j(g)$ acts only on the degrees of freedom at site $j$, and it may depend explicitly on $j$. We refer to this symmetry class as "pointwise" symmetries, because they act separately on every point (site) of the real space. There are two main sub-types of this symmetry class which have a major impact on physical modeling:

(i) **Global pointwise** symmetries: These symmetries have support on the full system, i.e. the $V_j(g)$ are non-trivial (different from the identity) for every site $j$. Usually these symmetries also have a homogeneous pointwise representation, that is, $V_j(g)$ does not explicitly depend on $j$. Typical examples are conservation of the total particle number or total spin magnetization.

(ii) **Lattice gauge** symmetries [104,172–174]: These pointwise symmetries have restricted supports, which usually extend just to a pair or plaquette of neighboring sites ($V_j(g)$ is the identity elsewhere). In turn, lattice gauge models often exhibit an extensive number of such symmetries (e.g. one for every pair of adjacent sites), which makes them again a powerful computational tool.

Most condensed matter systems also include symmetric content which is not of the pointwise form. For instance, this is the case for lattice geometry symmetries, such as translational invariance, lattice rotation/reflection and chiral symmetry. These symmetries play often an important role, and characterize some non-local order properties of various lattice systems (e.g. topological order) [175,176]. Although there are indeed proposals on how to address explicitly these other symmetry classes in specific tensor network architectures [93,177,178], they cannot be properly treated on a general ground, and will not be discussed here. Symmetric tensor network structures have also been proposed as analytical tools in the characterization of topological phases of matter; we refer the reader to [179] and references therein.

Having reviewed which classes of Hamiltonian symmetries we can efficiently use, we now provide in-detail documentation on how to exploit *pointwise* symmetries to achieve a computational advantage on TNs. Our presentation primarily draws upon Ref. [135], which specializes the underlying construction introduced in Ref. [170] to the case of an Abelian group. The construction relies on precise targeting of quantum many-body states having a well-defined quantum number: In this sense we will talk about invariant and covariant TN states, as we will see soon in Sec. 3.4. Since these TNs are built of symmetry-invariant tensor objects and symmetric link objects, we first introduce those in the next section.

## 3.3   Constructing Symmetric Tensors

In the construction of symmetric tensors, links (both physical and auxiliary) become associated with group representations and tensors become invariant objects with respect to the linked representations. This allows us to upgrade the data structures of links and tensors and their manipulation procedures to accommodate the symmetry content. Ultimately, symmetric links and tensors are the building blocks to construct symmetry-invariant and symmetry-covariant many-body TN states, as we will see in Sec. 3.4.

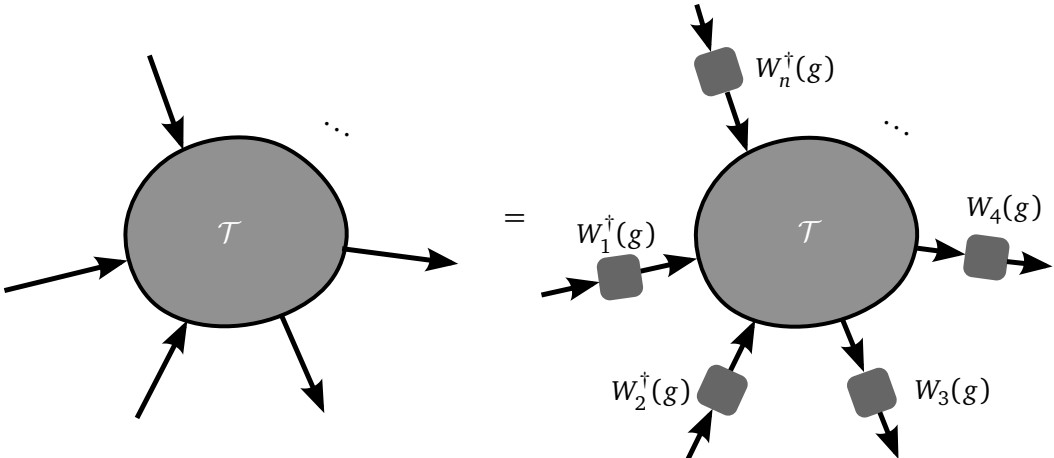

Figure 12: Graphical representation of an Abelian symmetric tensor (left) with $n$ directed links. Incoming links are drawn as arrows directed towards the tensor. By definition, the tensor is invariant under application of the representations $W_{r_{out}}(g)$ of $\mathcal{G}$ associated with all outgoing links $r_{out}$ (e.g. 3 and 4) and inverse representations $W_{r_{in}}^{\dagger}(g)$ associated with the incoming links $r_{in}$ (e.g. 1 and 2).

### 3.3.1 Abelian Symmetric Links

A tensor link is upgraded to be *symmetric* by associating a quantum number $\ell$ to each link index $i \in \mathbb{I}$. The number of different quantum numbers $\bar{\ell}$ can be smaller than the link dimension $d$. If we then distinguish indices having the same quantum number by a degeneracy count $\partial \in \{1, \ldots, \bar{\partial}_\ell\}$, we identify

$$i \equiv (\ell, \partial), \tag{27}$$

and call the tuple $(\ell, \partial)$ the (Abelian) symmetric link index. It is useful to organize the symmetric link indices with the same quantum number $\ell$ in a $\bar{\partial}_\ell$-dimensional degeneracy space or sector $\mathbb{D}_\ell$ such that the link index set $\mathbb{I}$ decomposes into their disjoint union

$$\mathbb{I} \simeq \biguplus_\ell \mathbb{D}_\ell. \tag{28}$$

From a physical perspective, we associate a specific unitary representation of the group $\mathcal{G}$ with the link. Namely, each tensor-link index $i$ identifies a specific irrep of group $\mathcal{G}$, labeled by the quantum number (or charge) $\ell$ and acting as a phase multiplication $e^{i\varphi_\ell(g)}$. In this sense, one can also interpret a symmetric link as a link equipped with a diagonal, unitary matrices group $W(g) \in \mathbb{C}^{d \times d}$ (or representation of $\mathcal{G}$) in which the diagonal elements are irreps $e^{i\varphi_\ell(g)}$ of $\mathcal{G}$, each appearing $\bar{\partial}_\ell$ times in accordance with the index decomposition of Eq. (28):

$$[W(g)]_{j,j'} \equiv [W(g)]_{(\ell,\partial),(\ell',\partial')} = e^{i\varphi_\ell(g)} \delta_{\ell,\ell'} \delta_{\partial,\partial'} \quad \forall g \in \mathcal{G}, j, j' \in \mathbb{I}. \tag{29}$$

Thanks to the group representation structure embedded in the links, tensors can share information related to the symmetry.

### 3.3.2 Abelian Symmetric Tensors

A tensor with all $n$ links symmetric is called *symmetric (invariant) tensor* if the tensor is left unchanged by the simultaneous application of the $n$ representations $W_r(g)$ of $\mathcal{G}$ associated with each tensor link $r$:

$$\mathcal{T}_{i_1 \ldots i_n} \overset{!}{=} \prod_{r=1}^{n} \sum_{j_r} \left[ W_r^{\dagger_r}(g) \right]_{i_r j_r} \mathcal{T}_{j_1 \ldots j_n} \quad \forall g \in \mathcal{G}, i_r, j_r \in \mathbb{I}_r. \tag{30}$$

With the superscripts $\dagger_r \in \{1, \dagger\}$ we specify on which links the acting representations are meant to be inverted. That is, we distinguish between *incoming* links, where the representations are inverted, and *outgoing* links, where they are not inverted. This formalism allows for a convenient graphical representation of links as arrows directed away from the tensor or into the tensor (see Fig. 12). In a network of symmetric tensors, every internal link must be going out of one tensor and into another: it is then guaranteed that the group acts as a gauge transformation on internal links of a tensor network, $W(g)W^\dagger(g) = \mathbb{1}$.

The product $\bigotimes_{r=1}^n W_r^{\dagger_r}(g)$ in Eq. (30) represents a symmetry of the tensor and is a proto-typical global pointwise symmetry, because it acts on *all* the links of the tensor (global), but as a tensor product of *local* operations (pointwise).

### 3.3.3 Inner Structure of Abelian Symmetric Tensors

With the link indices $i_r$ replaced by the associated quantum numbers $\ell_r$ and degeneracy indices $\partial_r$, the invariance Eq. (30) reads

$$\mathcal{T}_{\partial_1 \dots \partial_n}^{\ell_1 \dots \ell_n} \overset{!}{=} \mathcal{T}_{\partial_1 \dots \partial_n}^{\ell_1 \dots \ell_n} \prod_{r=1}^n \big[ e^{i\varphi_{\ell_r}(g)} \big]^{\dagger_r} \quad \forall g \in \mathcal{G}, \tag{31}$$

and we conclude that a tensor $\mathcal{T}$ is symmetric if it has nonzero elements $\mathcal{T}_{\ell_1 \dots \ell_n}^{\partial_1 \dots \partial_n}$ only where $\prod_{r=1}^n \big[ e^{i\varphi_{\ell_r}(g)} \big]^{\dagger_r} = 1$. For our convenience, we introduce the Abelian irrep- or quantum number fusion rule

$$\ell_1 \oplus \ell_2 = \ell' :\Longleftrightarrow \varphi_{\ell_1}(g) + \varphi_{\ell_2}(g) = \varphi_{\ell'}(g) \quad (\mathrm{mod}\, 2\pi) \quad \forall g \in \mathcal{G}, \tag{32}$$

and require that the quantum numbers from all the links fuse to the identical irrep $\ell_{\mathrm{ident}} = 0$:

$$\ell_1^{\dagger_1} \oplus \ell_2^{\dagger_2} \oplus \cdots \oplus \ell_n^{\dagger_n} \overset{!}{=} \ell_{\mathrm{ident}} \tag{33a}$$

$$\Longleftrightarrow \quad \pm\varphi_{\ell_1}(g) \pm \varphi_{\ell_2}(g) \pm \cdots \pm \varphi_{\ell_n}(g) = 0 \quad (\mathrm{mod}\, 2\pi) \quad \forall g \in \mathcal{G}, \tag{33b}$$

where the sign of the individual phases is to be taken opposite for incoming and outgoing links (due to the inversion $\varphi_{\ell^\dagger} = -\varphi_\ell$).

On this basis we define the *structural tensor*

$$\tilde{\delta}_{\ell_1 \dots \ell_n} = \begin{cases} 1 & \text{if } \ell_1^{\dagger_1} \oplus \ell_2^{\dagger_2} \oplus \cdots \oplus \ell_n^{\dagger_n} = \ell_{\mathrm{ident}} \\ 0 & \text{otherwise.} \end{cases} \tag{34}$$

We refer to any set of quantum numbers $\ell_1, \dots, \ell_n$ that fuse to the identical one as a matching set of quantum numbers, or *match* for short. In addition to the structural tensor, according to Eq. (31), a symmetric tensor is identified by *degeneracy* (sub-)*tensors* $\big[ \mathcal{R}^{\ell_1 \dots \ell_n} \big]_{\partial_1 \dots \partial_n}$. The latter are defined over the degeneracy subspaces $\mathbb{D}_{\ell_1} \otimes \mathbb{D}_{\ell_2} \otimes \cdots \otimes \mathbb{D}_{\ell_n}$ of all possible matches $\ell_1, \dots, \ell_n$:

$$\mathcal{T}_{\partial_1 \dots \partial_n}^{\ell_1 \dots \ell_n} = \big[ \mathcal{R}^{\ell_1 \dots \ell_n} \big]_{\partial_1 \dots \partial_n} \times \tilde{\delta}_{\ell_1 \dots \ell_n}. \tag{35}$$

The maximal number of (nonzero) elements of a symmetric tensor $\mathcal{T}$ hence reduces to the number of elements inside the degeneracy tensors

$$\#T = \sum_{\ell_1 \dots \ell_n} \#\{\mathcal{R}^{\ell_1 \dots \ell_n}\} \times \tilde{\delta}_{\ell_1 \dots \ell_n} = \sum_{\ell_1 \dots \ell_n} \tilde{\delta}_{\ell_1 \dots \ell_n} \prod_r \bar{\partial}_{\ell_r} \tag{36}$$

which is often much smaller than the number of elements of its non-symmetric counterpart (see later in this section for a more detailed discussion).

In Fig. 13 we give a graphical representation of a symmetry-invariant tensor. Attaching a dummy selector link (holding a single, non-degenerate irrep $\ell_{\mathrm{select}}$) to a symmetry-invariant tensor is a practical and yet rigorous way to build a covariant tensor (Fig. 14).

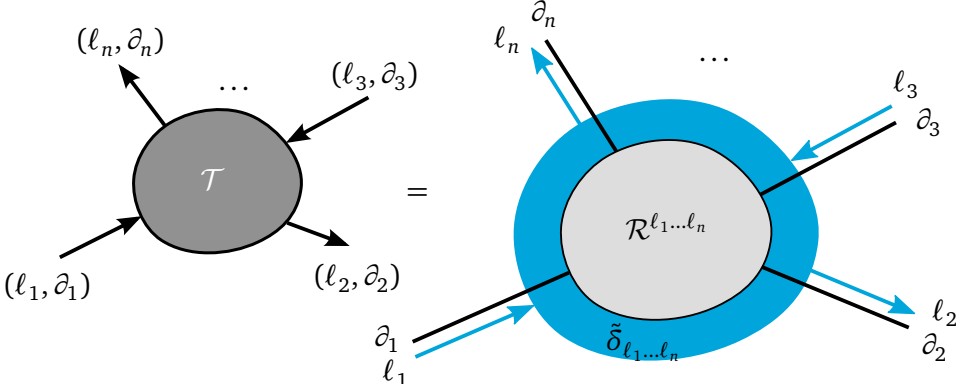

Figure 13: Inner structure of an Abelian symmetric tensor: The underlying cyan shape on the right represents the structural tensor $\tilde{\delta}$. Here it reads $\tilde{\delta}_{\ell_1 \dots \ell_n} = \delta_{\ell_1^\dagger \oplus \ell_2 \oplus \ell_3^\dagger \oplus \cdots \oplus \ell_n, \ell_{\text{ident}}}$ with inversion (†) of the quantum numbers on incoming links (such as links 1 and 3). Only sets $\ell_1 \dots \ell_n$ which are matches contribute a degeneracy tensor $\mathcal{R}_{\ell_1 \dots \ell_n}$, depicted as non-symmetric tensor (gray shape) defined over degeneracy indices $\partial_1 \dots \partial_n$ (black lines).

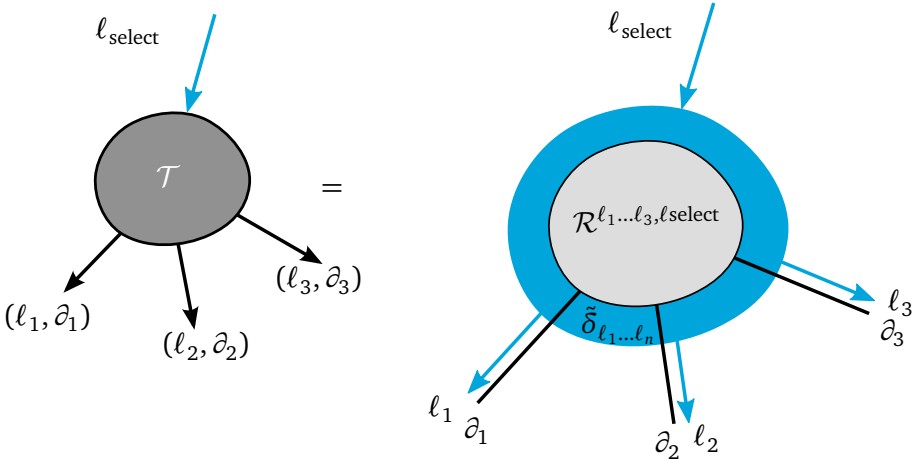

Figure 14: A covariant symmetric tensor is a symmetric (invariant) tensor with one link being a quantum-number selector link. This special link holds only a single, non-degenerate quantum number $\ell_{\text{select}}$. With this artifice, the structural tensor takes the usual form $\tilde{\delta}_{\ell_1 \ell_2 \ell_3 \ell_{\text{select}}} = \delta_{\ell_1 \oplus \ell_2 \oplus \ell_3 \oplus \ell_{\text{select}}^\dagger, \ell_{\text{ident}}}$.

### 3.3.4 Symmetric Links and Tensor Data Handling

We have shown in Eq. (35) that symmetric tensors are characterized by an inner block structure. This block structure emerged from the links being organized in quantum numbers and corresponding degeneracy spaces of Eq. (28), together with an assumed tensor invariance under symmetry transformations associated with the quantum numbers (Eq. (30)). In this section, we define all the necessary symmetric tensor operations to run symmetric TN algorithms on a computer. But first of all, we define the following three numerical objects, suited for object-oriented programming.

**Abelian Symmetry Group Object.** This object is a composition of a set of allowed quantum numbers $\left\{ \ell : e^{i\varphi_\ell(g)} \text{ is irrep of } \mathcal{G} \right\}$, the quantum numbers fusion rule $\oplus$ and the quantum number inversion operation $\ell \mapsto \ell^\dagger$.

| $\mathfrak{L}$ $(\ell_1^\dagger, \ell_2, \ell_3)$ | $\mathcal{R}^{\ell_1\ell_2\ell_3}$ | | | | |
| --- | --- | --- | --- | --- | --- |
| | $\bar\partial_{\ell_1} \times \bar\partial_{\ell_2} \times \bar\partial_{\ell_3}$ | $\partial_1$ | $\partial_2$ | $\partial_3$ | $[\mathcal{R}^{\ell_1\ell_2\ell_3}]_{\partial_1\partial_2\partial_3}$ |
| $(0^\dagger, 0, 0)$ | $2 \times 1 \times 1$ | 1 | 1 | 1 | $1/\sqrt{2}$ |
| | | 2 | 1 | 1 | 0 |
| $(1^\dagger, 1, 0)$ | $1 \times 1 \times 1$ | 1 | 1 | 1 | $\sqrt{2}/\sqrt{3}$ |
| $(1^\dagger, 0, 1)$ | $1 \times 1 \times 1$ | 1 | 1 | 1 | $1/\sqrt{3}$ |
| $(0^\dagger, 1, 1)$ | $2 \times 1 \times 1$ | 1 | 1 | 1 | $-1/\sqrt{2}$ |
| | | 2 | 1 | 1 | 1 |

(b)

| $i_1$ | $\ell_1$ | $\bar\partial_{\ell_1}$ | $\partial_1$ |
| --- | --- | --- | --- |
| 1 | 0 | 2 | 1 |
| 2 | | | 2 |
| 3 | 1 | 1 | 1 |

| $i_2$ | $\ell_2$ | $\bar\partial_{\ell_2}$ | $\partial_2$ |
| --- | --- | --- | --- |
| 1 | 0 | 1 | 1 |
| 2 | 1 | 1 | 1 |

| $i_3$ | $\ell_3$ | $\bar\partial_{\ell_3}$ | $\partial_3$ |
| --- | --- | --- | --- |
| 1 | 0 | 1 | 1 |
| 2 | 1 | 1 | 1 |

(a)

$i_1 \equiv (\ell_1, \partial_1)$

$\mathcal{T}$

$i_3 \equiv (\ell_3, \partial_3)$

$i_2 \equiv (\ell_2, \partial_2)$

(c)

Figure 15: Symmetric tensor version of the same tensor $\mathcal{T}_{i_1 i_2 i_3}$ as introduced in Fig. 5. To better see the symmetry of this tensor, imagine that the three-link tensor with link dimensions $d_1 = 3$ and $d_2 = d_3 = 2$ encodes two-qubit superpositions $\Psi_{i_1} = \sum_{i_2, i_3} \mathcal{T}_{i_1 i_2 i_3} |i_2 i_3\rangle$ of well-defined parity $i_2 + i_3 \,(\mathrm{mod})\, 2$. It is therefore invariant under a $\mathcal{G} = \mathbb{Z}_2$ group of parity symmetry operations: Take $W_r\,(g{=}0) = \mathbb{1}$ and $W_r\,(g{=}1) = \sigma^z$, e.g. $[W_r\,(g{=}1)]_{j,j} = e^{i(j-1)\pi}$, on the single-qubit links $r = 2, 3$. Then the tensor transforms as $\sum_{j_2, j_3} [W_2\,(g)]_{i_2 j_2} [W_3\,(g)]_{i_3 j_3} \mathcal{T}_{i_1 j_2 j_3} = \sum_{j_1} [W_1\,(g)]_{i_1 j_1} \mathcal{T}_{j_1 i_2 i_3}$, hence it is invariant under the global pointwise representation $W_1^\dagger \otimes W_2 \otimes W_3$ with $[W_1\,(g)]_{j,j} = e^{i\varphi_{\ell_1}(g)}$. Since the tensor is symmetry invariant, we can recast it as a symmetric tensor using the following data structures: (a) The three symmetric tensor links $r = 1, 2, 3$ as $i_r = (\ell_r, \partial_r)$. (b) The symmetric tensor itself, defined by its set of matches $\mathfrak{L}$ (in this example all four possible matches are present), plus a degeneracy subtensor $[\mathcal{R}^{\ell_1\ell_2\ell_3}]_{\partial_1\partial_2\partial_3} \times \tilde\delta_{\ell_1\ell_2\ell_3}$ for each match in $\mathfrak{L}$. (c) The graphical representation, where we consider the two qubit links outgoing and the first link incoming in accordance with the inversion on $W_1^\dagger$ in the defining invariance. Although the $\mathbb{Z}_2$ symmetry is self-adjoint and link directions could be neglected in this example, this is not true in general.

For practical purposes, it is often sufficient to represent (a subset of) the group's quantum numbers as integers. In the presence of multiple symmetries, acting as a direct-product group $\mathcal{G} = \mathcal{G}_1 \times \cdots \times \mathcal{G}_K$ of $K$ independent symmetry groups $\mathcal{G}_k$, $k = 1 \ldots K$, it is convenient to operate with ordered tuples of individual quantum numbers instead: $\ell = \left( \ell^{[1]}, \ldots, \ell^{[K]} \right)$. These tuples obey element-wise fusion rules, as described in item (iv) of Sec. 3.1.

**Symmetric Link Object.** The symmetric link object encompasses a subset of $\bar{\ell}$ different quantum numbers $\ell$ of the group $\mathcal{G}$, and their corresponding degeneracies $\bar{\partial}_\ell \geq 1$.

**Symmetric Tensor Object.** A symmetric tensor object $\mathcal{T}$ consists of:

(i) $n$ symmetric link objects.

(ii) $n$ directions $\dagger_r \in \{1, \dagger\}$ (boolean variables), specifying for each link whether it is incoming (inverted representation) or outgoing (non-inverted representation).

(iii) A set of *matching quantum numbers*

$$\mathfrak{L} = \left\{ (\ell_1, \ldots, \ell_n) : \ell_1^{\dagger_1} \oplus \ell_2^{\dagger_2} \oplus \cdots \oplus \ell_n^{\dagger_n} = 0 \right\}, \tag{37}$$

also simply called *matches*, where each quantum number $\ell_r$ is taken from its respective link object.

(iv) For each match $(\ell_1, \ldots, \ell_n)$ within $\mathfrak{L}$, a degeneracy tensor $\mathcal{R}^{\ell_1 \cdots \ell_n}$. This is defined as an ordinary, non-symmetric tensor (defined in Sec. 2.1) over the corresponding degeneracy spaces with index dimensions $\bar{\partial}_{\ell_1}, \ldots, \bar{\partial}_{\ell_n}$, which can be queried from the links.

Fig. 15 shows a practical example of this data structure. An important optimization is that only matches that come with a non-vanishing degeneracy tensor $\mathcal{R}^{\ell_1 \cdots \ell_n} \neq 0$ have to be listed in $\mathfrak{L}$. We say that these matches are "present" in $\mathcal{T}$.

In order to unify the handling of multiple symmetries in direct product groups with tuples $\ell = \left( \ell^{[1]}, \ldots, \ell^{[K]} \right)$ of $K$ individual quantum numbers, one can define simple integer *sectors $s$* and use them to index quantum numbers $\ell \equiv \ell(s)$ within a given link. In this way, the elements of $\mathfrak{L}$ can be replaced by *matching sectors* $(s_1, \ldots, s_n)$.

## 3.4 Symmetric Tensor Networks

For a symmetric Hamiltonian $\mathcal{H}$, different symmetry sectors $\ell \neq \ell'$ are dynamically decoupled. Therefore, we can always restrict each TN simulation to span a single sector $\ell_{\text{select}}$ of the symmetry, without loss of generality. In particular, we will construct, via a symmetric tensor network ansatz, quantum many-body states which are either invariant or covariant under the action of the (pointwise) symmetry. This construction will rely, of course, on symmetric tensor objects.

(i) **Invariant states** are those which satisfy $U(g)|\Psi\rangle = |\Psi\rangle$, i.e. they belong by definition to the sector $\ell = 0$, the identical irrep. Let now the QMB state $|\Psi\rangle$ be a tensor network state, given by Eqs. (1) and (2). Let us also associate a representation $W_\eta(g)$ of the symmetry group $\mathcal{G}$ to every auxiliary space (or nonphysical link) $\eta$. Then it is easy to see that if every tensor is invariant under the action of the group, i.e. it fulfills Eq. (30), the state $|\Psi\rangle$ is symmetry invariant (see Fig. 16) [170]: We name this construct a symmetric (invariant) tensor network. The reverse is also true: it is possible to decompose an arbitrary invariant state into the symmetric tensor network formulation, but link dimensions might grow if the TN is not loop-free [180]. We will investigate this construction in detail in Sec. 4.2.4.

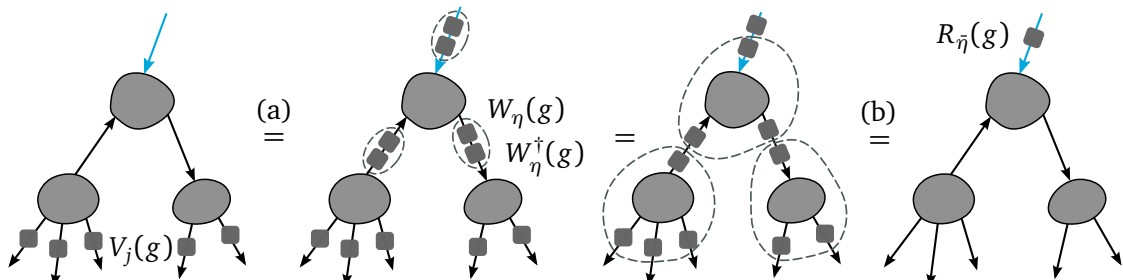

Figure 16: Symmetric tensor network, composed of symmetric tensors. A global pointwise operation on the physical links (left) is absorbed by the network as follows: (a) We apply a gauge transformation with the representations on the virtual links. (b) Now we use the invariance property of each tensor to absorb all surrounding transformations at each tensor. We see that the application of the global pointwise symmetry $\bigotimes_j V_j(g)$ is equivalent to the application of the phase shift $R_{\bar{\eta}}(g) = e^{i\varphi_{\ell_{\text{select}}}(g)}$. The representation at the selector link thus determines the sector $\ell_{\text{select}}$ of the physical state.

(ii) **Covariant states** are states $|\Psi_\ell\rangle$ that belong to a single, selected irrep $\ell_{\text{select}}$ of the symmetry (thus actually extending the concept of invariance to other irreps than the identical one, $\ell_{\text{ident}} = 0$). For Abelian symmetries they are the eigenstates of the whole symmetry group $U(\mathcal{G})$, and transform as $U(g)|\Psi_\ell\rangle = e^{i\varphi_\ell(g)}|\Psi_\ell\rangle$. They are necessary to address a variational problem for a given (conserved) number of particles $\ell$.

Encoding covariant states in TN language is a simple extension of the invariant case [170]: Specifically, one can set every tensor to be invariant, as in Eq. (31), but then introduce an additional, special virtual link $\bar{\eta}$. This *selector link* is attached to a single tensor (any tensor fits, see e.g. Fig. 16), has dimension $D_{\bar{\eta}} = 1$ and its symmetry representation $R_{\bar{\eta}}(g)$ is fixed to the $\ell_{\text{select}}$ irrep $R_{\bar{\eta}}(g) = W^{[\ell_{\text{select}}]}(g)$. Equivalently, one can think of a covariant symmetric tensor network as a TN made of a single covariant tensor (i.e. an invariant tensor equipped with a selector link), as in Fig. 14, while all the other tensors are invariant, as in Fig. 13. This is an effective way to target the $\ell$ covariant subspaces while using the same data structure as used for invariant states: the symmetric tensor.

## 3.5 Symmetric Tensor Operations

So far, we focused on the upgraded data structures and fundamental group operations. Now, we will equip symmetric tensors with the same basic functionality that we provided for ordinary tensors in Sec. 2.2. Again, element-wise operations (such as scalar multiplication, complex conjugation or addition and subtraction) are not covered in detail. It is easy to cast them into a symmetric form by carrying out the respective ordinary operation on all degeneracy tensors independently. This strategy extends naturally to the linear algebra operations, discussed in Sec. 3.5.2.

In the modular programming of tensor network algorithms, these symmetric operations constitute an outer layer with respect to the basic tensor operations, as we sketched in Fig. 1.

### 3.5.1 Link and Tensor Manipulations

As for the non-symmetric tensors, we again first discuss operations that involve a single symmetric link or tensor.

(A)   **Input:** Links $(\ell_1, \partial_1)$, $(\ell_2, \partial_2)$, $(\ell_3, \partial_3)$ with directions $(\dagger, 1, 1)$ and a set of elements $\mathcal{T}^{\ell_1\ell_2\ell_3}_{\partial_1\partial_2\partial_3}$: $\left\{ \mathcal{T}^{0,0,0}_{2,1,1} = \pi, \mathcal{T}^{1,0,1}_{1,1,1} = i\pi \right\}$.

**Output:**

| $\mathfrak{L}$ | $\mathcal{R}^{\ell_1\ell_2\ell_3}$ | | | | |
|---|---|---|---|---|---|
| $(\ell_1^\dagger, \ell_2, \ell_3)$ | $\bar\partial_{\ell_1} \times \bar\partial_{\ell_2} \times \bar\partial_{\ell_3}$ | $\partial_1$ | $\partial_2$ | $\partial_3$ | $[\mathcal{R}^{\ell_1\ell_2\ell_3}]_{\partial_1\partial_2\partial_3}$ |
| $(0^\dagger, 0, 0)$ | $2 \times 1 \times 1$ | 1 | 1 | 1 | 0 |
| | | 2 | 1 | 1 | $\pi$ |
| $(1^\dagger, 0, 1)$ | $1 \times 1 \times 1$ | 1 | 1 | 1 | $i\pi$ |

Matches $\mathfrak{L} = \{(0,0,0),(1,0,1)\}$ from provided nonzero elements.

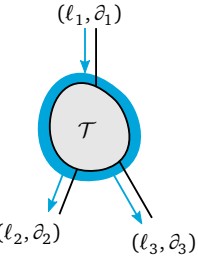

(B)   **Input:** Links $(\ell_1, \partial_1)$, $(\ell_2, \partial_2)$, $(\ell_3, \partial_3)$ with directions $(\dagger, 1, 1)$.

**Output:**

| $\mathfrak{L}$ | $\mathcal{R}^{\ell_1\ell_2\ell_3}$ | | | | |
|---|---|---|---|---|---|
| $(\ell_1^\dagger, \ell_2, \ell_3)$ | $\bar\partial_{\ell_1} \times \bar\partial_{\ell_2} \times \bar\partial_{\ell_3}$ | $\partial_1$ | $\partial_2$ | $\partial_3$ | $[\mathcal{R}^{\ell_1\ell_2\ell_3}]_{\partial_1\partial_2\partial_3}$ |
| *empty* | | | | | |

Matches $\mathfrak{L} = \{\}$ all non-present.

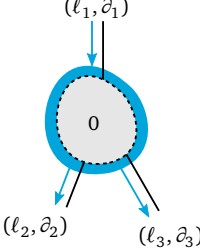

(C)   **Input:** Links $(\ell_1, \partial_1)$, $(\ell_2, \partial_2)$, $(\ell_3, \partial_3)$ with directions $(\dagger, 1, 1)$ and a random number generator.

**Output:**

| $\mathfrak{L}$ | $\mathcal{R}^{\ell_1\ell_2\ell_3}$ | | | | |
|---|---|---|---|---|---|
| $(\ell_1^\dagger, \ell_2, \ell_3)$ | $\bar\partial_{\ell_1} \times \bar\partial_{\ell_2} \times \bar\partial_{\ell_3}$ | $\partial_1$ | $\partial_2$ | $\partial_3$ | $[\mathcal{R}^{\ell_1\ell_2\ell_3}]_{\partial_1\partial_2\partial_3}$ |
| $(0^\dagger, 0, 0)$ | $2 \times 1 \times 1$ | 1 | 1 | 1 | $1.217 - 0.419i$ |
| | | 2 | 1 | 1 | $-0.311 + 1.520i$ |
| $(1^\dagger, 1, 0)$ | $1 \times 1 \times 1$ | 1 | 1 | 1 | $1.731 + 0.048i$ |
| $(1^\dagger, 0, 1)$ | $1 \times 1 \times 1$ | 1 | 1 | 1 | $1.096 + 0.654i$ |
| $(0^\dagger, 1, 1)$ | $2 \times 1 \times 1$ | 1 | 1 | 1 | $0.512 - 0.291i$ |
| | | 2 | 1 | 1 | $-0.442 - 0.673i$ |

Matches $\mathfrak{L} = \{(0,0,0),(1,1,0),(1,0,1),(0,1,1)\}$ all identified and present.

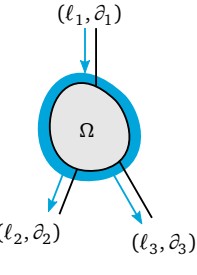

(D)   **Input:** Link $(\ell_a, \partial_a)$, will be mirrored to $(\ell_b, \partial_b) \equiv (\ell_a, \partial_a)$.

**Output:**

| $\mathfrak{L}$ | $\mathcal{R}^{\ell_a\ell_b}$ | | | |
|---|---|---|---|---|
| $(\ell_a^\dagger, \ell_b)$ | $\bar\partial_{\ell_a} \times \bar\partial_{\ell_b}$ | $\partial_1$ | $\partial_2$ | $[\mathcal{R}^{\ell_a\ell_b}]_{\partial_a\partial_b}$ |
| $(0^\dagger, 0)$ | $2 \times 2$ | 1 | 1 | 1.0 |
| | | 2 | 1 | 0.0 |
| | | 1 | 2 | 0.0 |
| | | 2 | 2 | 1.0 |
| $(1^\dagger, 1)$ | $1 \times 1$ | 1 | 1 | 1.0 |

Matches $\mathfrak{L} = \{(0,0),(1,1)\}$, one pair of each quantum number.

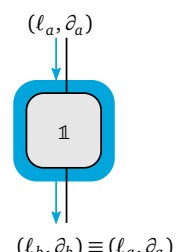

Figure 17: Symmetric tensor initialization examples for tensors with the same $\mathcal{G} = \mathbb{Z}_2$ symmetry and links $(\ell_1, \partial_1)$, $(\ell_2, \partial_2)$, $(\ell_3, \partial_3)$ as in Fig. 15. (A) from user provided elements, (B) filled with zeros, (C) from random elements, e.g. a Gaussian distribution, (D) an "identity matrix" on a given link $(\ell_a, \partial_a) := (\ell_1, \partial_1)$.

**Symmetric Tensor Initialization.** Four notable ways to create a symmetric tensor object are (see Fig. 17):

(A) *Initialize from user-provided elements.* Create the respective degeneracy tensors containing the provided elements and check the fusion rule of the corresponding matches.

(B) *Initialize with zeros.* No match will be present, hence no memory allocated for degeneracy tensors.

(C) *Fill all degeneracy tensors with random elements.* First, we search for all possible matches given the directed links. We make all of them present, and we initialize the respective degeneracy tensors with elements provided from a random number generator.

(D) *Set up an "identity matrix".* this requires a duplicate symmetric link. We then define a two-link tensor connected to the two links, with opposite directions. Then from all quantum numbers in the links we form matches $(\ell, \ell)$, and create the corresponding degeneracy tensors, each equal to the identity: $\left[ \mathcal{R}^{\ell,\ell} \right]_{\partial_1 \partial_2} = \delta_{\partial_1, \partial_2}$. Note that it might be advantageous to use sparse diagonal matrices for degeneracy tensors (see Sec. 2.2.2).

**Symmetric Link Inversion.** This operation inverts a symmetric link: it maps all quantum numbers $\ell$ from the link into their inverses $\ell \to \ell^\dagger$. Degeneracies remain unchanged, $\bar{\partial}_{\ell^\dagger}^{(\text{new})} = \bar{\partial}_\ell^{(\text{old})}$.

**Inversion of a Link in a Symmetric Tensor.** This operation inverts a link $r$ connected to a tensor while preserving the invariance of the tensor. It acts as follows: (i) The corresponding symmetric link $r$ is inverted (see above). (ii) The direction of link $r$ is flipped. (iii) The set of matches is preserved, however every match is manipulated so that $\ell_r \to \ell_r^\dagger$ at link $r$. Notice that this preserves the fusion rule. (iv) The degeneracy tensors are unaltered. Multiple links connected to a tensor can be inverted simultaneously or sequentially.

**Hermitian Conjugate of a Symmetric Tensor.** The Hermitian conjugation is an operation well distinct from link inversions, because it flips link directions without inverting the link representations. This operation becomes fundamental in e.g. calculating expectation values from symmetric tensor networks. It acts as follows: (i) The links are left unchanged. (ii) All the link directions are flipped. (iii) The set of matches is unchanged (thanks to $\ell = \ell^\dagger$ for $\ell = 0$). (iv) The degeneracy tensors are complex conjugated.

**Symmetric Link Index Manipulations.** Reducing, enlarging, and rearranging the set of indices of a symmetric link all belong to this class of transformations. For every irrep $\ell$ present in link $r$, its degeneracy indices $\partial_\ell \in \{1, \ldots, \bar{\partial}_\ell\}$ can be compressed, inflated, or rearranged as in Sec. 2.2.1. Additionally, quantum numbers can be entirely removed from a link, such as when $\bar{\partial}_\ell$ is compressed to zero (in which case $\ell$ should be explicitly removed from the internal set), or added anew to the link, in which case a nonzero degeneracy $\bar{\partial}_{\ell_{\text{new}}}$ should be assigned to it.

Two special cases of symmetric link index manipulations should be mentioned here, because they play an important role in algorithms:

(i) The intersection $(\ell, \partial) \cap (\ell', \partial')$ is the symmetric link obtained from the set intersection of the quantum numbers carried in the links $\ell$ and $\ell'$ with the minimal degeneracy dimensions $\min(\bar{\partial}_\ell, \bar{\partial}'_\ell)$.

(ii) Padding two links $(\ell, \partial) + (\ell', \partial')$ is defined as the disjoint union of their symmetric indices. The padded link has quantum numbers from the union of $\ell$ and $\ell'$ with corresponding

degeneracy dimensions $\bar{\partial}_\ell + \bar{\partial}'_\ell$ added up (taking e.g. $\bar{\partial}_\ell = 0$ if $\ell$ is not a quantum number in the respective link).

**Index Manipulations in a Symmetric Tensor.**   Index manipulation within a link of a symmetric tensor impacts the degeneracy tensors separately. On each degeneracy tensor, it acts as ordinary index inflation, reordering or restriction (as described in Sec. 2.2.1). Two special situations may arise: (i) When quantum numbers get entirely removed from a link (i.e. degeneracy drops to zero), corresponding matches and degeneracy tensors must be removed. (ii) When new quantum numbers appear on a link, new matches may become possible and can be assigned nonzero degeneracy tensors. For the sake of efficiency, creating new matches should be avoided unless the related degeneracy tensors are to be filled with nonzero elements.

**Symmetric Tensor Element Access.**   An individual symmetric tensor element can be accessed from its quantum numbers and degeneracy indices. Similarly, an entire degeneracy tensor can be addressed by providing the quantum numbers (match) alone. A more general interface to address tensor elements is provided by subtensors (see Sec. 2.2.1). Also in the symmetric case, a subtensor $\mathcal{S}$ is obtained from $\mathcal{T}$ by reducing the set of degeneracy indices in all links (see Sec. 3.5.1). As a consequence, $\mathcal{S}$ is a subtensor in $\mathcal{T}$ if all its degeneracy tensors are ordinary subtensors in the corresponding degeneracy tensors of $\mathcal{T}$.

   **Subtensor Readout.**   This operation reads out a subset of elements at specified indices from the tensor $\mathcal{T}$ and stores them in a new tensor $\mathcal{S}$, according to a link-index reduction given for every link $r$. Matches in $\mathcal{S}$ are the subset of all matches from $\mathcal{T}$ whose quantum numbers $\ell_r$ have not been deleted from the respective link. Since each degeneracy tensor of $\mathcal{S}$ is a subtensor in the respective degeneracy tensor of $\mathcal{T}$, this operation performs a block-wise ordinary subtensor extraction.

   **Subtensor Assignment.**   This operation overwrites elements at specified indices in a tensor $\mathcal{T}$ with the content of a subtensor $\mathcal{S}$. Again, the respective element indices are specified by a link-index reduction for every link $r$, such that we can perform block-wise ordinary subtensor assignment on degeneracy tensors of $\mathcal{S}$ and $\mathcal{T}$. It is however important to keep in mind that degeneracy tensors associated with non-present matches implicitly contain zeros. In order to perform the subtensor assignment, we compare the sets of matches $\mathfrak{L}_\mathcal{T}$ and $\mathfrak{L}_\mathcal{S}$ of both tensors $\mathcal{T}$ and $\mathcal{S}$, respectively. In detail, we distinguish three cases (see Fig. 18 for an example):

 (i) Matches in $\mathfrak{L}_\mathcal{T} \cap \mathfrak{L}_\mathcal{S}$, present in $\mathcal{T}$ and $\mathcal{S}$: We perform ordinary subtensor assignment between the respective degeneracy tensors of similar matches.

 (ii) Matches in $\mathfrak{L}_\mathcal{T} \setminus \mathfrak{L}_\mathcal{S}$, present in $\mathcal{T}$ but not $\mathcal{S}$: We set corresponding degeneracy tensor elements in $\mathcal{T}$ to zero, equivalent to ordinary subtensor assignment where the degeneracy tensor of $\mathcal{S}$ contains only zeros. In the special case where the respective degeneracy tensors of $\mathcal{T}$ and $\mathcal{S}$ are of same dimensions, we can instead remove the former from $\mathfrak{L}_\mathcal{T}$ as it becomes entirely zero.

 (iii) Matches in $\mathfrak{L}_\mathcal{S} \setminus \mathfrak{L}_\mathcal{T}$, present in $\mathcal{S}$ but not $\mathcal{T}$: We add these matches to the set $\mathfrak{L}_\mathcal{T}$, initialize the corresponding degeneracy tensors to zero, and finally inscribe the subtensor from $\mathcal{S}$ via ordinary subtensor assignment.

**Input:**

Tensor $\mathcal{T}$:

| $\mathfrak{L}$ $(\ell_1^\dagger, \ell_2, \ell_3)$ | $\mathcal{R}^{\ell_1\ell_2\ell_3}$ $\bar{\partial}_{\ell_1} \times \bar{\partial}_{\ell_2} \times \bar{\partial}_{\ell_3}$ | $\partial_1$ | $\partial_2$ | $\partial_3$ | $[\mathcal{R}^{\ell_1\ell_2\ell_3}]_{\partial_1\partial_2\partial_3}$ |
|---|---|---|---|---|---|
| $(1^\dagger, 1, 0)$ | $1 \times 1 \times 1$ | 1 | 1 | 1 | 1.0 |
| $(1^\dagger, 0, 1)$ | $1 \times 1 \times 1$ | 1 | 1 | 1 | 2.0 |
| $(0^\dagger, 1, 1)$ | $2 \times 1 \times 1$ | 1 | 1 | 1 | 3.0 |
|  |  | 2 | 1 | 1 | 4.0 |

Tensor links:

| $\ell_1$ | $\bar{\partial}_{\ell_1}$ | $\ell_2$ | $\bar{\partial}_{\ell_2}$ | $\ell_3$ | $\bar{\partial}_{\ell_3}$ |
|---|---|---|---|---|---|
| 0 | 2 | 0 | 1 | 0 | 1 |
| 1 | 1 | 1 | 1 | 1 | 1 |

Matches $\mathfrak{L}_{\mathcal{T}} = \{(1,1,0),(1,0,1),(0,1,1)\}$ are present in $\mathcal{T}$.

Subtensor $\mathcal{S}$:

| $\mathfrak{L}$ $(\ell_1^\dagger, \ell_2, \ell_3)$ | $\mathcal{R}^{\ell_1\ell_2\ell_3}$ $\bar{\partial}_{\ell_1} \times \bar{\partial}_{\ell_2} \times \bar{\partial}_{\ell_3}$ | $\partial_1$ | $\partial_2$ | $\partial_3$ | $[\mathcal{R}^{\ell_1\ell_2\ell_3}]_{\partial_1\partial_2\partial_3}$ |
|---|---|---|---|---|---|
| $(0^\dagger, 0, 0)$ | $1 \times 1 \times 1$ | 1 | 1 | 1 | 5.0 |
| $(1^\dagger, 1, 0)$ | $1 \times 1 \times 1$ | 1 | 1 | 1 | 6.0 |

Subtensor links:

| $\ell_1$ | $\bar{\partial}_{\ell_1}$ | $\ell_2$ | $\bar{\partial}_{\ell_2}$ | $\ell_3$ | $\bar{\partial}_{\ell_3}$ |
|---|---|---|---|---|---|
| 0 | 1 | 0 | 1 | 0 | 1 |
| 1 | 1 | 1 | 1 | 1 | 1 |

Matches $\mathfrak{L}_{\mathcal{S}} = \{(0,0,0),(1,1,0)\}$ are present in $\mathcal{S}$.

**Output:**

Tensor $\mathcal{T}$ with subtensor $\mathcal{S}$ inscribed:

| $\mathfrak{L}$ $(\ell_1^\dagger, \ell_2, \ell_3)$ | $\mathcal{R}^{\ell_1\ell_2\ell_3}$ $\bar{\partial}_{\ell_1} \times \bar{\partial}_{\ell_2} \times \bar{\partial}_{\ell_3}$ | $\partial_1$ | $\partial_2$ | $\partial_3$ | $[\mathcal{R}^{\ell_1\ell_2\ell_3}]_{\partial_1\partial_2\partial_3}$ |
|---|---|---|---|---|---|
| $(0^\dagger, 0, 0)$ | $2 \times 1 \times 1$ | 1 | 1 | 1 | 5.0 |
|  |  | 2 | 1 | 1 | 0.0 |
| $(1^\dagger, 1, 0)$ | $1 \times 1 \times 1$ | 1 | 1 | 1 | 6.0 |
| $(0^\dagger, 1, 1)$ | $2 \times 1 \times 1$ | 1 | 1 | 1 | 0.0 |
|  |  | 2 | 1 | 1 | 4.0 |

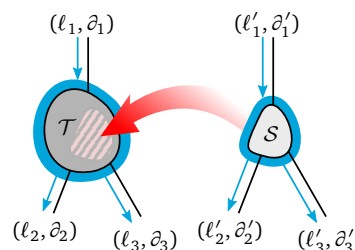

Matches $\mathfrak{L}_{\mathcal{T}} \cap \mathfrak{L}_{\mathcal{S}} = \{(1,1,0)\}$ present in both $\mathcal{T}$ and $\mathcal{S}$: Degeneracy tensors have been partially or entirely overwritten with elements from $\mathcal{S}$.

Matches $\mathfrak{L}_{\mathcal{T}} \setminus \mathfrak{L}_{\mathcal{S}} = \{(1,0,1),(0,1,1)\}$ present only in $\mathcal{T}$: Degeneracy tensors have been partially filled with zeros or entirely removed.

Matches $\mathfrak{L}_{\mathcal{S}} \setminus \mathfrak{L}_{\mathcal{T}} = \{(0,0,0)\}$ present only in $\mathcal{S}$: Degeneracy tensors have been initialized with zeros and partially or entirely overwritten with elements from $\mathcal{S}$.

Figure 18: Example of a symmetric subtensor assignment within a three-link tensor $\mathcal{T}$ with the same $\mathcal{G} = \mathbb{Z}_2$ symmetry and links as in Fig. 15. Top: $\mathcal{T}$ and the subtensor $\mathcal{S}$ come with similar symmetry and number of links, but the second degeneracy index $\partial_1 = 2$ has been removed from the degeneracy subspace $\mathbb{D}_{\ell_1=0}$ of the invariant quantum number in the first subtensor link. This is an exemplary link-index reduction, mapping all the lower degeneracy indices (here $\partial \equiv 1$) from $\mathcal{S}$ onto the same indices ($\partial \equiv 1$) in $\mathcal{T}$. Bottom: Output tensor $\mathcal{T}$ after assignment, with all elements from $\mathcal{S}$ inscribed into respective subtensors of $\mathcal{T}$.

**Permutation of Symmetric Links at a Tensor.**    The reordering of symmetric tensor links from $(\ell_r, \partial_r)$ into $(\ell_{\sigma(r)}, \partial_{\sigma(r)})$, given a permutation $\sigma$, is equal to a permuting all present matching quantum numbers and their associated degeneracy tensors according to $\sigma$, and finally keeping track of the symmetric tensor's link directions $\dagger_{\sigma(r)}$. The computational cost is dictated by the ordinary permutations on the degeneracy tensors, which is at most linear in the number of elements.

**Fusion and Splitting of Symmetric Links.**    The fuse- and split-operations in Sec. 2.2.1 can be conveniently adapted to symmetric tensors. However, taking into account the special structure of symmetric links, we cannot rely on the simple enumeration scheme of Eq. (8) anymore. Instead, we must resort to a more general permutation which is still computationally cheap but also respects the extra symmetry structure of the links.

We organize this paragraph as follows: First we define the symmetric link fusion as a bijection between multiple directed symmetric links and the combined index of the fused link. This mapping is then summarized in the form of a (symmetric) fuse-node. Finally, we describe how to obtain the corresponding fused or split tensor in full symmetric representation.

**Link Fusion.**    When fusing symmetric links, the goal is to replace multiple links of an $n$-link tensor $\mathcal{T}$ with given directions $\{\dagger_r\}$ at positions $r = k, \dots, m$ by a single, again symmetric *fused link* whose associated symmetric index $(\ell, \partial)$ is a bijection from the original link indices

$$(\ell, \partial) = \mathrm{fuse}(\ell_k, \dots, \ell_m; \partial_k, \dots, \partial_m). \tag{38}$$

We want to choose this mapping such that the group representation $W(g)$ associated with the fused link and each group element $g \in \mathcal{G}$ is the matrix direct product (or Kronecker product) of the representations on the original links

$$W(g) := \bigotimes_{r=k}^{m} W_r^{\dagger_r}(g) \quad \forall g \in \mathcal{G}, \tag{39}$$

i.e. such that in agreement with the defining Eq. (30) the *fused tensor* $\mathcal{T}'$ remains symmetric. This requirement fixes

$$\ell = \mathrm{fuse}(\ell_k, \dots, \ell_m) := \ell_k^{\dagger_k} \oplus \dots \oplus \ell_m^{\dagger_m} \tag{40}$$

and we recover the quantum-numbers fusion rule. Note that Eq. (40) holds for an outgoing fused link. To obtain an incoming fused link, $\ell$ has to be inverted.

For the degeneracy indices $\partial$, any enumeration over fused link indices with the same quantum number $\ell$ can be adopted. Such degeneracies arise from two sources: the degeneracies already present in the original links, and "collisions" of similar quantum numbers $\ell$ obtained from different combinations $\mathrm{fuse}(\ell_k, \dots, \ell_m) = \mathrm{fuse}(\ell_k', \dots, \ell_m')$, as Eq. (40) is not bijective for itself. We adopt the following strategy: On each combination $\ell_k, \dots, \ell_m$ we use the ordinary fusion $\mathbb{D}_{\ell_k} \times \dots \times \mathbb{D}_{\ell_m} \to \mathbb{D}_\ell^{\ell_k \dots \ell_m}$ (Eq. (8)), mapping the original degeneracies into a partial degeneracy index $\partial_\ell^{\ell_k \dots \ell_m} := \mathrm{fuse}(\partial_k, \dots, \partial_m)$. The partial degeneracy indices of colliding combinations $\{\ell_k, \dots, \ell_m : \mathrm{fuse}(\ell_k, \dots, \ell_m) = \ell\}$ are then concatenated (or stacked) into the fused degeneracy $\mathbb{D}_\ell \simeq \biguplus \mathbb{D}_\ell^{\ell_k \dots \ell_m} = \biguplus_\alpha \mathbb{D}_\ell^\alpha$ by adding the appropriate degeneracy offset $\Delta_\ell^\alpha$ for each collision:

$$\partial_\ell = \Delta_\ell^\alpha + \mathrm{fuse}(\partial_k, \dots, \partial_m). \tag{41}$$

Here $\alpha \longleftrightarrow (\ell_k, \dots, \ell_m)$ uniquely enumerates all collisions into a given $\ell$, and $\Delta_\ell^\alpha := \sum_{\alpha' < \alpha} \bar{\partial}_\ell^{\alpha'}$ amounts to the cumulated partial degeneracy obtained from fusing all preceding colliding degeneracy subspaces of dimensions $\bar{\partial}_\ell^\alpha := \#\mathbb{D}_\ell^\alpha = \bar{\partial}_{\ell_k} \times \dots \times \bar{\partial}_{\ell_m}$.

This choice of concatenated partial degeneracies makes fusing and splitting of consecutive links at symmetric tensors computationally cheap, because none of the original degeneracy tensor elements needs to be reordered in memory. It only involves moving contiguous sections of memory holding complete degeneracy tensors into, or from, the offset positions Eq. (41) within the fused degeneracy tensors.

Examples are given in Figs. (19) and (20), where we iterate over all possible combinations $\ell_k, \ldots, \ell_m$ in column-major ordering (fast increment in $\ell_k$) and obtain $\alpha = 0, 1, \ldots$ as collision counter for each fused quantum number $\ell$. Simultaneously, we increment the cumulated degeneracy offset with every collision according to $\Delta_\ell^{\alpha+1} = \Delta_\ell^\alpha + \prod_{r=k}^m \bar{\partial}_{\ell_r}$, such that we finally obtain the fused link's total degeneracy $\bar{\partial}_\ell$, provided that we started from $\Delta_\ell^0 = 0$.

**Fuse-Node.** The mapping of the symmetric link fusion in Eq. (38) is well-defined through Eqs. (40) and (41). Consequently, it is practical to store the relevant information of this mapping in a separate numerical object, the (symmetric) fuse-node $\mathcal{F}$. The fuse-node is a special symmetric tensor defined over the links $k, \ldots, m$ from $\mathcal{T}$ but with inverted directions, plus the fused link $(\ell, \partial)$ with the same direction as it appears on the fused tensor $\mathcal{T}'$:

$$\mathcal{F}_{\partial_k \ldots \partial_m, \partial}^{\ell_k \ldots \ell_m, \ell} = \begin{cases} 1 & \text{if } (\ell, \partial) = \text{fuse}(\ell_k, \ldots, \ell_m; \partial_k, \ldots, \partial_m), \\ 0 & \text{otherwise.} \end{cases} \tag{42}$$

The fused tensor $\mathcal{T}'$ is then obtained by simply contracting $\mathcal{F}$ into $\mathcal{T}$ over the original links which shall be fused, as shown in Fig. 21. Splitting up the fused link into the original links restores $\mathcal{T}$ and can be achieved by a second contraction of $\mathcal{T}'$ with the split-node $\mathcal{S} = \mathcal{F}^\dagger$ over the fused link. $\mathcal{S}$ equals the fuse-node but has all link directions inverted. Since splitting a link is the inversion of fusing, their contraction over the fused link is again the identity $\mathcal{F}^\dagger \mathcal{F} = \mathbb{1}$.

However, owing to its sparse structure, storing $\mathcal{F}$ in the form of a symmetric tensor and carrying out fuse- and split-operations by actually contracting such a fuse- or split-node is computationally inefficient. Therefore, we avoid expanding Eq. (42) into tensor elements and instead store the relation $(\ell, \alpha) \longleftrightarrow (\ell_k, \ldots, \ell_m)$ between the fused quantum number $\ell$ with corresponding collision index $\alpha$, and each possible combination of quantum numbers $\ell_k, \ldots, \ell_m$ from the original links. One possibility to achieve this is to enumerate the latter by a scalar index similar to Eq. (5), which can also easily be inverted.

Finally, we store the degeneracy offsets $\Delta_\ell^\alpha$ in the fuse-node, along with the original symmetric links and the fused link, including all link directions. Such a fuse-node can be viewed as a fully equipped network node. One can then define further basic operations for these fuse-nodes — just like we did for tensors. The permutation is a common example which is frequently combined with link fusion: Permuting a fuse-node merely involves rearranging some link positions, collision indices and degeneracy offsets, which is much faster than permuting the fused links in a tensor before fusing or after splitting.

We now outline two algorithms that, given one such fuse-node, create the fused- or split tensor. For efficiency, simultaneous fusions of arbitrary (not necessarily adjacent) disjoint groups of links of a tensor can be handled with minor adaptations, allowing the algorithm to process multiple fuse-nodes simultaneously. The total computational cost of both algorithms is composed of ordinary non-symmetric fuse- and split-operations on all degeneracy tensors, and thus remains at most linear in the number of symmetric tensor elements.

**Fused Tensor.** The complete link fusion at a symmetric tensor $\mathcal{T}$ could proceed as follows:

(i) Initialize the fused tensor $\mathcal{T}'$ with $n' = n - m + k$ links similar to $\mathcal{T}$. The links from positions $r = k, \ldots, m$ at $\mathcal{T}$ are replaced with the fused link at $\mathcal{T}'$.

| $b$ | $\ell_2$ | $\ell_3$ | $\partial_2$ | $\partial_3$ | $\ell_b$ | $\partial_b$ | $\alpha$ | $\Delta^\alpha_{\ell_b}$ |
|---|---|---|---|---|---|---|---|---|
| 1 | 0 | 0 | 1 | 1 | 0 | 1 | 0 | 0 |
| 2 | 1 | 0 | 1 | 1 | 1 | 1 | 0 | 0 |
| 3 | 0 | 1 | 1 | 1 | 1 | 2 | 1 | 1 |
| 4 | 1 | 1 | 1 | 1 | 0 | 2 | 1 | 1 |

(a)

| $\ell_b$ | $\bar{\partial}_{\ell_b}$ | $\partial_b$ |
|---|---|---|
| 0 | 2 | 1 |
|   |   | 2 |
| 1 | 2 | 1 |
|   |   | 2 |

(b)

| $\mathfrak{L}$ $(\ell_a^\dagger,\ell_b)$ | $\mathcal{R}^{\ell_a\ell_b}$ $\bar{\partial}_{\ell_a}\times\bar{\partial}_{\ell_b}$ | $\partial_a$ | $\partial_b$ | $[\mathcal{R}^{\ell_a\ell_b}]_{\partial_a\partial_b}$ |
|---|---|---|---|---|
| $(0^\dagger,0)$ | $2\times 2$ | 1 | 1 | $1/\sqrt{2}$ |
|  |  | 2 | 1 | 0 |
|  |  | 1 | 2 | $-1/\sqrt{2}$ |
|  |  | 2 | 2 | 1 |
| $(1^\dagger,1)$ | $1\times 2$ | 1 | 1 | $\sqrt{2}/\sqrt{3}$ |
|  |  | 1 | 2 | $1/\sqrt{3}$ |

(c)

$a \equiv (\ell_a,\partial_a)$

$\mathcal{M}$

$b \equiv (\ell_b,\partial_b)$

(e)

$$\mathcal{M}^{\ell_a\ell_b}_{\partial_a\partial_b}=\left(\begin{array}{cc|cc} 1/\sqrt{2} & -1/\sqrt{2} & 0 & 0 \\ 0 & 1 & 0 & 0 \\ \hline 0 & 0 & \sqrt{2}/\sqrt{3} & 1/\sqrt{3} \end{array}\right)$$

(d)

Figure 19: Symmetric link fusion $(\ell_b,\partial_b) = \text{fuse}((\ell_2,\partial_2),(\ell_3,\partial_3))$ of the last two links in the tensor $\mathcal{T}^{(\ell_1\ell_2)\ell_3}_{(\partial_1\partial_2)\partial_3}$ from the example in Fig. 15: In (a), we show the fuse mapping in detail; $b$ is the linear index in column-major enumeration of $\ell_2,\ell_3,\partial_2,\partial_3$. The collision index $\alpha$ and the corresponding degeneracy offset $\Delta^\alpha_{\ell_b}$ help us in the algorithm of Sec. 3.5.1 to place degeneracy indices of pairs $\ell_2,\ell_3$ that fuse to the same $\ell_b$ into successive intervals $\Delta^\alpha_{\ell_b}+1,\ldots,\Delta^{\alpha+1}_{\ell_b}$ up to $\Delta^2_{\ell_b}=\bar{\partial}_{\ell_b}$ of the degeneracy space $\mathbb{D}_b$. Here, the complete degeneracy $\bar{\partial}_{\ell_b}$ is induced by those collisions, but in Fig 20 we will encounter additional degeneracy already present in the links to be fused. The resulting matrix $\mathcal{M}$ shown in (e) has links $(\ell_a,\partial_a)=(\ell_1,\partial_1)$ (cf. Fig. 15) and the fused $(\ell_b,\partial_b)$ given in (b). The elements are detailed in (c). We have only two matching quantum numbers and hence two degeneracy tensors, which appear as blocks in the block-diagonal matrix-representation $\mathcal{M}$ shown in (d). Here we present rows and columns in ascending order (degeneracies being the fast index) according to $(\ell_a,\partial_a)=(0,1),(0,2),(1,1)$ and $(\ell_b,\partial_b)=(0,1),(0,2),(1,1),(1,2)$.

Note that in this example, $b$ also corresponds to the non-symmetric link index of Fig. 5, but here we have reordered the fused symmetric link indices $\ell_b,\partial_b$ according to quantum numbers $\ell_b$.

(ii) In all tuples of matching quantum numbers present in $\mathcal{T}$, replace $\ell_k,\ldots,\ell_m$ by the fused $\ell$ and remove possible duplicates ("collisions"). This yields the set of matches $\mathfrak{L}'$ of $\mathcal{T}'$.

(iii) Set the corresponding fused degeneracy tensors $\mathcal{R}'^{\ell_1\ldots\ell_{k-1},\ell,\ell_{m+1}\ldots\ell_n}$ in $\mathcal{T}'$ to zero, with degeneracy dimensions $\bar{\partial}_\ell$ queried from the fused link.

(iv) Go through the set $\mathfrak{L}$ of matching quantum numbers of $\mathcal{T}$ again, but this time fuse the links $(k,\ldots,m)$ at each degeneracy tensor $\mathcal{R}^{\ell_1\ldots\ell_n}$ and place these as subtensors into the respective degeneracy tensors of $\mathcal{T}'$, which might have a larger degeneracy dimension

| $j$ | $\ell_1^\dagger$ | $\ell_2$ | $\ell_3$ | $\partial_1$ | $\partial_2$ | $\partial_3$ | $\ell_j$ | $\partial_j$ | $\alpha$ | $\Delta_{\ell_j}^\alpha$ |
|---|---|---|---|---|---|---|---|---|---|---|
| 1 | 0 | 0 | 0 | 1 | 1 | 1 | 0 | 1 | 0 | 0 |
| 2 |   |   |   | 2 | 1 | 1 |   | 2 |   |   |
| 3 | 1 | 0 | 0 | 1 | 1 | 1 | 1 | 1 | 0 | 0 |
| 4 | 0 | 1 | 0 | 1 | 1 | 1 | 1 | 2 | 1 | 1 |
| 5 |   |   |   | 2 | 1 | 1 |   | 3 |   |   |
| 6 | 1 | 1 | 0 | 1 | 1 | 1 | 0 | 3 | 1 | 2 |
| 7 | 0 | 0 | 1 | 1 | 1 | 1 | 1 | 4 | 2 | 3 |
| 8 |   |   |   | 2 | 1 | 1 |   | 5 |   |   |
| 9 | 1 | 0 | 1 | 1 | 1 | 1 | 0 | 4 | 2 | 3 |
| 10 | 0 | 1 | 1 | 1 | 1 | 1 | 0 | 5 | 3 | 4 |
| 11 |   |   |   | 2 | 1 | 1 |   | 6 |   |   |
| 12 | 1 | 1 | 1 | 1 | 1 | 1 | 1 | 6 | 3 | 5 |

(a)

$$\mathcal{V}_{\partial_j}^{\ell_j} = \begin{pmatrix} 1/\sqrt{2} \\ 0 \\ \sqrt{2}/\sqrt{3} \\ 1/\sqrt{3} \\ -1/\sqrt{2} \\ \hline 1 \\ 0 \\ 0 \\ 0 \\ 0 \\ 0 \\ 0 \end{pmatrix}$$

(d)

| $\ell_j$ | $\bar{\partial}_{\ell_j}$ | $\partial_j$ |
|---|---|---|
| 0 | 6 | 1…6 |
| 1 | 6 | 1…6 |

(b)

| $\mathfrak{L}$ $(\ell_j)$ | $\mathcal{R}^{\ell_j}$ | | |
|---|---|---|---|
|  | $\bar{\partial}_{\ell_j}$ | $\partial_j$ | $[\mathcal{R}^{\ell_j}]_{\partial_j}$ |
| (0) | 6 | 1 | $1/\sqrt{2}$ |
|  |  | 2 | 0 |
|  |  | 3 | $\sqrt{2}/\sqrt{3}$ |
|  |  | 4 | $1/\sqrt{3}$ |
|  |  | 5 | $-1/\sqrt{2}$ |
|  |  | 6 | 1 |

(c)

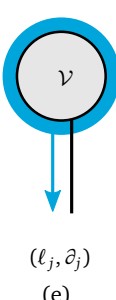

$(\ell_j, \partial_j)$

(e)

Figure 20: Fusing $\mathcal{T}$ from Fig. 15 into a vector $\mathcal{V}_{\partial_j}^{\ell_j}$: The three-link fusion $(\ell_j, \partial_j) = \text{fuse}((\ell_1, \partial_1), (\ell_2, \partial_2), (\ell_3, \partial_3))$ is detailed in (a). The fused link is summarized in (b) and (c), where the degeneracy $\bar{\partial}_{\ell_j}$ originates from two sources: First, the original link $(\ell_1, \partial_1)$ already comes with degeneracy $\bar{\partial}_{\ell_1=0} > 1$. Furthermore, the collisions of combinations $\ell_1, \ell_2, \ell_3$ into similar quantum numbers $\ell_j$ are cumulated in $\Delta_{\ell_j}^\alpha$ (see Sec. 3.5.1). The resulting vector elements vanish in all but the identical sector $\ell_j = 0$, as the vector must be invariant. If we again order the indices ascendingly in $\ell_j, \partial_j$, with fast increment in the degeneracy, we obtain the vector representation in (d). Graphically, the vector can be represented as shown in (e). Note that we would have obtained the same result by fusing links $(\ell_a, \partial_a)$ and $(\ell_b, \partial_b)$ at the matrix $\mathcal{M}$ from Fig. 19, as link fusions can be nested.

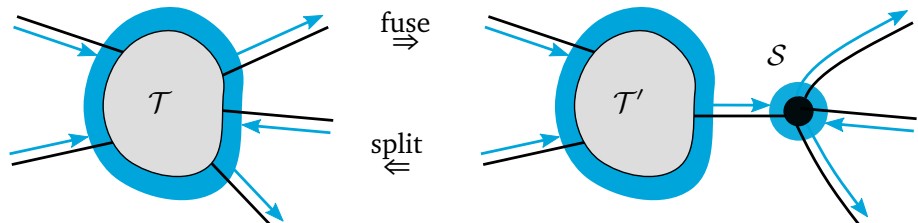

Figure 21: Fusion and splitting of a symmetric tensor. From the fuse-operation acting on a tensor $\mathcal{T}$ (left-to-right) we obtain the fused tensor $\mathcal{T}'$ together with a split-node $\mathcal{S} = \mathcal{F}^\dagger$ which encodes the symmetry fusion rule for the irreps (cyan filled circle) on the fused links and the index enumeration in the degeneracy space (black filled circle on top). The inversion of the process, splitting the fused link into the three original links (right-to-left), is equivalent to a contraction of the split-node over the fused link.

$\bar{\partial}_\ell \geq \prod_{r=k}^m \bar{\partial}_{\ell_r}$ due to collisions. The subtensor's exact position in the degeneracy subspace $\mathbb{D}_\ell$ is determined by the interval of degeneracy offsets $\Delta_\ell^\alpha + 1, \ldots, \Delta_\ell^{\alpha+1}$, where $\alpha$ is the enumeration index of collision $\ell_k, \ldots, \ell_m$ into $\ell$ as specified with the fuse-node $\mathcal{F}$.

Note that if the original tensor had non-present matches, the fused tensor also does not necessarily have all matching quantum numbers present in the fused set $\mathfrak{L}'$. This must be considered when e.g. fusing a tensor into a block-diagonal matrix $\mathcal{M}_{\ell_a \ell_b}^{\partial_a \partial_b}$ in order to perform matrix operations. For example, in the exponentiation of a square matrix with one incoming and one outgoing link (such as a time-evolution operation), non-present matches $(\ell_a, \ell_b = \ell_a) \notin \mathfrak{L}'$ must be identified and explicitly provided with identity degeneracy tensors $\mathcal{R}^{\ell_a \ell_b} = \mathbb{1}$, resulting from the exponentiation of the implicit zero blocks.

**Split Tensor.** Restoring the original links $(\ell_k, \partial_k), \ldots, (\ell_m, \partial_m)$ in place of a fused link $(\ell, \partial)$ at some splitting-position $k$ in the fused tensor $\mathcal{T}'$ yields the split tensor $\mathcal{T}''$. Step-by-step, the split tensor can be obtained from the $n'$-link tensor $\mathcal{T}'$ and the respective fuse-node $\mathcal{F}$ as follows:

(i) Initialize the split tensor $\mathcal{T}''$ with $n = n' + m - k$ links, copying links and directions from $\mathcal{T}'$, but replacing the link at position $k$ of $\mathcal{T}'$ with the "original links" $(\ell_k, \partial_k), \ldots, (\ell_m, \partial_m)$ including their directions as stored in the fuse-node.

(ii) In every tuple of matching quantum numbers present in $\mathcal{T}'$, expand $\ell$ at the splitting-position $k$ to all possible combinations $\ell_k, \ldots, \ell_m$ of quantum numbers in the "original links" (now at positions $k, \ldots, m$ in $\mathcal{T}''$) that fuse to $\ell$ (both the links and the respective combinations are stored as collisions in $\mathcal{F}$). This yields the set of matches $\mathfrak{L}''$ of $\mathcal{T}''$.

(iii) For each extended match, create the split degeneracy tensor $\mathcal{R}''^{\ell_1 \ldots \ell_{k-1}, \ell_k \ldots \ell_m, \ell_{m+1} \ldots \ell_n}$ by ordinarily splitting the $k$-th link of the subtensor $\mathcal{R}'^{\ell_1 \ldots \ell_{k-1}, \ell, \ell_{m+1} \ldots \ell_n}$ of $\mathcal{T}'$. All degeneracy dimensions can be queried from the links in the fuse-node. Again, the subtensor's exact position in the degeneracy subspace $\mathbb{D}_\ell$ is determined by the interval of degeneracy offsets $\Delta_\ell^\alpha + 1, \ldots, \Delta_\ell^{\alpha+1}$, available with the collision index $\alpha$ from $\mathcal{F}$.

Note that the tensor $\mathcal{T}'$, at which we split a link, does not necessarily have to be the result of a fuse-operation. The required fuse-node could also be obtained from the fusion of similar links at another tensor, or it might have been created manually. If however we consecutively fuse and split the same links, with no further operations performed in between, the original tensor is restored: $\mathcal{T}'' = \mathcal{T}$. However, degeneracy tensors with all elements equal to zero may appear in such a process. For the sake of efficiency, one can remove those from $\mathcal{T}''$ in a post-processing step after splitting.

**Fused Links and Symmetric Tensor Invariance.**

(i) The definition of the fused quantum number $\ell$ in Eq. (40) can be invoked to give an alternative definition of a symmetric tensor $\mathcal{T}$: After fusing all links on $\mathcal{T}$ into a single link $(\ell, \partial)$, the result is a symmetric vector $\mathcal{V} \equiv \left[ \mathcal{V}_\partial^\ell \times \tilde{\delta}_\ell \right]$ which has non-vanishing elements in the identical sector only, i.e. there is only one possible degeneracy tensor $\mathcal{V}_\partial^0$ (see Fig. 20 for an example).

(ii) Similarly, fusing a bipartition of links into symmetric row- and column-indices $(\ell_1, \partial_1)$ and $(\ell_2, \partial_2)$ yields, in accordance with Schur's Lemma, a block-diagonal matrix $\mathcal{M} \equiv \left[ \mathcal{M}_{\partial_1 \partial_2}^{\ell_1 \ell_2} \times \tilde{\delta}_{\ell_1 \ell_2} \right]$ (see Fig. 19 for an example).

(iii) For numerical purposes, it may be advantageous to avoid fusions and to keep as many links as possible, since that gives us the freedom *not* to store certain matching sectors $(\ell_1, \ldots, \ell_n) \in \mathfrak{L}$ with vanishing degeneracy tensors $\mathcal{R}^{\ell_1 \cdots \ell_n}$ that would otherwise appear explicitly as blocks of zeros within the fused degeneracy tensors $\mathcal{R}'^{\ell_1 \cdots \ell_{k-1}, \ell, \ell_{m+1} \cdots \ell_n}$ of the fused tensor.

(iv) The tensor symmetry invariance is not broken by adding a one-dimensional "dummy" link (in either direction) which carries only the identical quantum number $\ell = 0$ (cf. Fig. 22).

**Downgrade of a Symmetric Tensor to an Ordinary Tensor.** The operation of recasting a symmetric tensor (and tensor network) into an ordinary one can be useful for debugging and interfacing symmetric to non-symmetric tensor objects. For example, it can be used to measure a non-symmetric observable $O$ for a symmetric state. The links are first downgraded from symmetric links to ordinary links, where each downgraded link dimension $D = \sum_\ell \bar{\partial}_\ell$ is the sum of its previous degeneracy space dimensions $\bar{\partial}_\ell$, and an index mapping is chosen. The degeneracy subtensors $\mathcal{R}$ are written into the new ordinary tensor at the right positions, and the remaining tensor elements are filled with zeros.

**Link Clean-up.** This is a special operation upon a symmetric link, which employs information from the surrounding network links to wipe out redundant data on the link itself. Its goal is to remove indices while preserving the TN state manifold, and it finds extensive use in the random symmetric TN initialization (see Sec. 4.3.1). Namely, assume we want to clean-up the link connecting the network nodes $q$ and $q'$ (tensors at these nodes do not have to be defined). We then intersect the link representation with the fusion of all the other (directed) links connected to $q$ and again, with the fusion of all the other (directed) links connected to $q'$. The original link is then replaced by the resulting intersection.

### 3.5.2 Symmetric Tensor Linear Algebra Operations

**Symmetric Tensor Contraction.** In the contraction of two symmetric $n$- and $m$-link input tensors $\mathcal{A}$ and $\mathcal{B}$ over any $k$ shared symmetric links $(s, \partial_s)$, the $(m+n-2k)$-link output tensor $\mathcal{C}$ is again symmetric. It is however mandatory that the shared links are similar in quantum numbers and respective degeneracy dimensions $\bar{\partial}_s$, and that each of them is either directed from $\mathcal{A}$ to $\mathcal{B}$ or vice-versa. For example, the symmetric version of Eq. (14) reads

$$\mathcal{C}_{\partial_{a_1} \cdots \partial_{a_{n-k}} \partial_{b_{k+1}} \cdots \partial_{b_m}}^{a_1 \cdots a_{n-k} b_{k+1} \cdots b_m} = \sum_{s_1 \cdots s_k} \sum_{\partial_{s_1} \cdots \partial_{s_k}} \mathcal{A}_{\partial_{a_1} \cdots \partial_{a_{n-k}} \partial_{s_1} \cdots \partial_{s_k}}^{a_1 \cdots a_{n-k} s_1 \cdots s_k} \mathcal{B}_{\partial_{s_1} \cdots \partial_{s_k} \partial_{b_{k+1}} \cdots \partial_{b_m}}^{s_1 \cdots s_k b_{k+1} \cdots b_m}, \tag{43}$$

where $(a, \partial_a)$ and $(b, \partial_b)$ are now the symmetric indices of the remaining links not contracted over, attached to $\mathcal{A}$ and $\mathcal{B}$, respectively. The degeneracy tensors $\mathcal{R}^{a_1 \cdots a_{n-k} b_{k+1} \cdots b_m}$ in $\mathcal{C}$ are

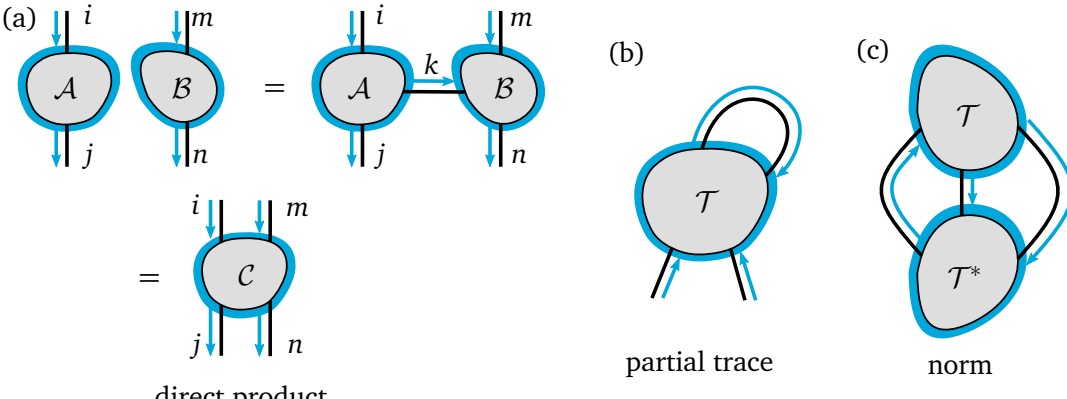

Figure 22: Variants of contractions (compare Sec. 2.2.2 and Fig. 8), now cast into a symmetric form: (a) Direct product. Between any two tensors, a dummy link can be inserted to indicate a contraction. With symmetric tensors, such intermediate product links may acquire a deeper meaning as quantum-number "transfer links" if they carry a non-degenerate quantum number $\ell \neq 0$. A typical application is the correlation function $c^\dagger \otimes c$ of particle creation and annihilation operators $c^\dagger$ and $c$ under a U(1) particle-conservation symmetry, which might, for example, occur in a Hubbard model. In this context, the transfer link carries a single quantum number $\ell = +1$ or $\ell = -1$ (depending on the direction of the link) which accounts for the particle hopping associated with the operation $c^\dagger \otimes c$. Contracting the (covariant) tensors representing the two operators over the intermediate link yields again a symmetric invariant tensor. (b) Partial trace, which can be carried out as a contraction over pairs of similar, oppositely directed symmetric links at the same tensor. (c) Hilbert–Schmidt norm, performed as contraction with the hermitian conjugate $\mathcal{T}^\dagger$ which extends the complex conjugation by the simultaneous inversion of all links.

obtained by adding up (outer sum in Eq. (43)) the standard contractions (inner sum) of degeneracy tensors from the input tensors, directly exposing the block-wise structure of such a procedure.

It seems natural to transform all involved tensors into (block-diagonal) symmetric matrices and carry out the contraction as a (block-wise) matrix-matrix multiplication, similar to ordinary contraction (Sec. 2.2.2). Indeed, symmetric fuse-, split- and permute-operations allow us to implement this sequence of operations in a simple and elegant manner. However, this may not always be the most efficient strategy, as it does not fully exploit the possible absence of matches within the fused tensor.

In the following, we present a specialized algorithm, and afterwards discuss the possible advantages over the "fuse-multiply-split" strategy: Namely, one first identifies pairs of subsets $\mathfrak{L}_{\mathcal{A}}^{s_1 \dots s_k} \subseteq \mathfrak{L}_{\mathcal{A}}$, $\mathfrak{L}_{\mathcal{B}}^{s_1 \dots s_k} \subseteq \mathfrak{L}_{\mathcal{B}}$ holding matches from $\mathcal{A}$ and $\mathcal{B}$ that share similar combinations of quantum numbers $s_1, \dots, s_k$ on the contraction links. Next, the degeneracy tensors associated with these subsets, given by $\mathcal{R}_{\mathcal{A}}^{a_1 \dots a_{n-k} s_1 \dots s_k}$ and $\mathcal{R}_{\mathcal{B}}^{s_1 \dots s_k b_{k+1} \dots b_m}$, are contracted for all available combinations $a_1, \dots, a_{n-k}$ and $b_{k+1}, \dots, b_m$ of the quantum numbers not contracted over. The resulting contracted tensors are then stored together with an intermediate list $\mathfrak{L}_{\mathcal{C}}'$ of possibly non-unique tuples $(a_1, \dots, a_{n-k}, b_{k+1}, \dots, b_m)$. As we required all contracted links to have pairwise opposite directions on tensors $\mathcal{A}$ and $\mathcal{B}$, these tuples are already matching quantum numbers obeying

$$a_1^{\dagger_1} \oplus \cdots \oplus a_{n-k}^{\dagger_{n-k}} \oplus b_{k+1}^{\dagger_{k+1}} \oplus \cdots \oplus b_m^{\dagger_m} = 0, \qquad (44)$$

where the †-superscripts yield the inverse quantum numbers only on the incoming links of $\mathcal{A}$ and $\mathcal{B}$, respectively. A final reduction step completes the contracted and again symmetric tensor $\mathcal{C}$: The set of matches $\mathfrak{L}_{\mathcal{C}}$ is obtained by identifying and removing all recurrences of matches from $\mathfrak{L}'_{\mathcal{C}}$ while adding up the associated degeneracy tensors into the final degeneracy tensors $\mathcal{R}^{a_1 \ldots a_{n-k} \, b_{k+1} \ldots b_m}$.

Even though our example Eq. (43) again resembles a matrix-multiplication (after fusing row- and column-links), it should be noted that the set of matches $\mathfrak{L}$ obtained by our specialized procedure usually contains only a subset of all possible matches $(a_1, \ldots, a_{n-k}, b_{k+1}, \ldots, b_m)$ that are allowed by Eq. (44). It is therefore not advisable to first permute the full symmetric input tensors $\mathcal{A}$ and $\mathcal{B}$ into (block-diagonal) matrices as in Sec. 2.2.2, since the symmetric link fusion might force us to represent some of the previously non-present degeneracy tensors as blocks of zeros in memory — thereby giving up computational speed[1] and precision.

The computational complexity $\mathcal{O}_{\text{tot}}$ of the symmetric tensor contraction is governed by the degeneracy tensor contractions, each of which can be carried out independently and is bounded linearly in all involved degeneracy dimensions by

$$\mathcal{O}_{\text{deg}} = \mathcal{O}\left( \prod_{r=1}^{k} \bar{\partial}_{s_r} \prod_{r=1}^{n-k} \bar{\partial}_{a_r} \prod_{r=k+1}^{m} \bar{\partial}_{b_r} \right). \tag{45}$$

Consequently, if the degeneracy dimensions of all involved links are peaked around certain quantum numbers, the total cost is governed by the largest contraction of degeneracy tensors involved. If, instead, degeneracies come in rather flat distributions over the active number of sectors, the number of contractions $S$ between degeneracy tensors may grow to $S \leq \left( \prod_{r=1}^{k} \bar{s}_r \prod_{r=1}^{n-k} \bar{a}_r \prod_{r=k+1}^{m} \bar{b}_r \right) / \left( \bar{a}_{\max} \cdot \bar{b}_{\max} \right)$, where $\bar{a}_{\max}$ and $\bar{b}_{\max}$ denote the maximum number of symmetry sectors among uncontracted links at $\mathcal{A}$ and $\mathcal{B}$ (compare also Sec. 3.5.3). The maximal speedup that can be achieved over the full, non-symmetric contraction therefore scales with $\bar{a}_{\max} \cdot \bar{b}_{\max}$.

In a similar way, we can also cast all contraction variants discussed in Sec. 2.2.2 into fully symmetric operations, as depicted in Fig. (22).

**Symmetric Tensor Decompositions.** Decompositions of symmetric tensors are in principle similar to decompositions of ordinary non-symmetric tensors (discussed in Sec. 2.2.2). We require that the output tensors of such symmetric decompositions have to be element-wise equal to the outcome of the corresponding symmetry-unaware operations. However, we must also require that these output tensors are all *symmetric* tensors again, and that they are mutually connected by *directed symmetric* links.

The usual QR- and singular-value matrix decompositions indeed fulfill these requirements. Focusing on the SVD, we proceed by restating the procedure of Sec. 2.2.2 for symmetric tensors. Specifically, we want to decompose an arbitrary $n$-link symmetric input tensor $\mathcal{T}$ over a certain link-bipartition $(1, \ldots, r),(r+1, \ldots, n)$ into two (semi-)unitary, symmetric output tensors $\mathcal{V}$ and $\mathcal{W}$ (holding the singular vectors) and a non-negative diagonal matrix $\lambda$ (holding the singular values). Both output tensors shall be connected to $\lambda$ through an intermediate symmetric link $(k, \partial_k)$,

$$\mathcal{T}^{\ell_1 \ldots \ell_n}_{\partial_1 \ldots \partial_n} = \sum_{k, \partial_k} \mathcal{V}^{\ell_1 \ldots \ell_r \, k}_{\partial_1 \ldots \partial_r \, \partial_k} \lambda^{k}_{\partial_k} \mathcal{W}^{k \, \ell_{r+1} \ldots \ell_n}_{\partial_k \, \partial_{r+1} \ldots \partial_n}. \tag{46}$$

In order to achieve this result, we do the following (see Fig. 23):

---

[1]Even though optimized matrix-matrix multiplications exhibit advantageous performance on those larger blocks, in practice the performance gain by not having to contract non-present degeneracy tensors can be significantly larger.

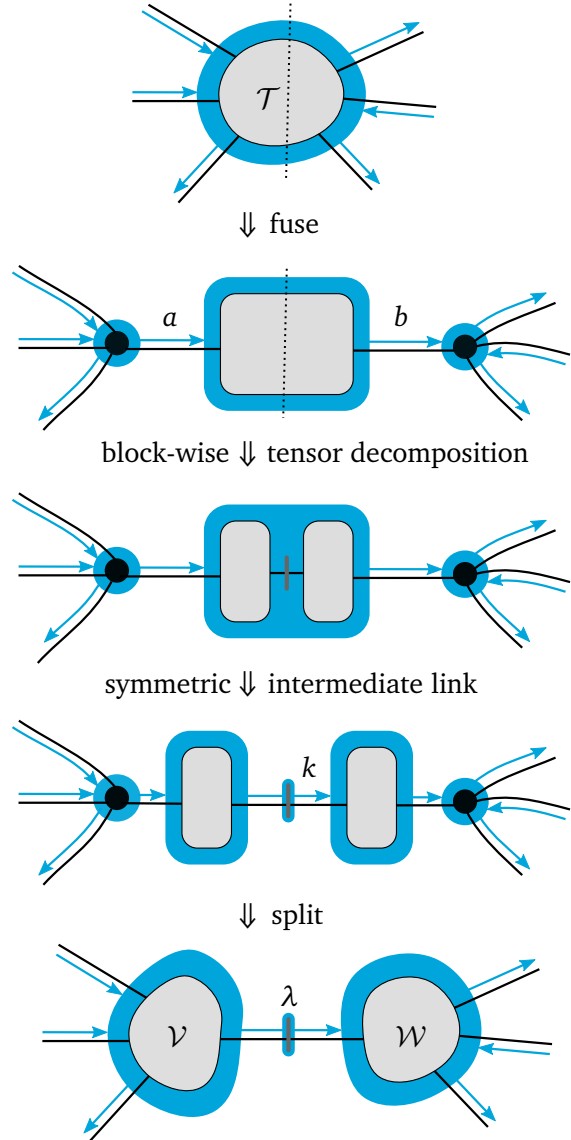

Figure 23: Steps in symmetric tensor decomposition (e.g. QR or singular value decomposition), from top to bottom: First the input tensor $\mathcal{T}$ is fused into a matrix over the indicated link bipartition $a,b$. The resulting degeneracy tensors form a block-wise matrix and can be decomposed independently, often at greatly reduced computational cost. Their factors are separated into two symmetric tensors $\mathcal{V}$ and $\mathcal{W}$, connected through a new intermediate symmetric link $k$, and weighted by singular values $\lambda$ in the case of an SVD. Finally, the original links are restored with the help of fuse-nodes that were obtained in the initial step.

(i) Fuse $\mathcal{T}$ into a matrix with incoming row-link $(a, \partial_a) = \text{fuse}(\ell_1, \ldots, \ell_r; \partial_1, \ldots, \partial_r)$ and outgoing column-link $(b, \partial_b) = \text{fuse}(\ell_{r+1}, \ldots, \ell_n; \partial_{r+1}, \ldots, \partial_n)$. Keep the fuse-nodes. Due to the symmetry invariance, we obtain a block-diagonal matrix

$$\mathcal{T}^{ab}_{\partial_a \partial_b} = \left[ \mathcal{R}^{ab} \right]_{\partial_a \partial_b} \times \delta_{ab} \,, \tag{47}$$

in which the degeneracy tensors $\mathcal{R}^{ab}$ play the role of blocks.

(ii) Create the intermediate symmetric link from the intersection $(k, \partial_k) = (a, \partial_a) \cap (b, \partial_b)$

(see Sec. 3.5.1).

(iii) Carry out the ordinary matrix decomposition on all present matrix blocks $\mathcal{R}^{ab}$ independently. In the case of an SVD, we obtain block-wise matrix factors $\mathcal{V}^{ak}$ and $\mathcal{W}^{kb}$ holding the left- and right singular vectors, and vectors (or diagonal matrices) $\lambda^k$ holding their respective singular values:

$$\left[\mathcal{R}^{ab}\right]_{\partial_a \partial_b} = \sum_{\partial_k=1}^{\bar{\partial}_k} \left[\mathcal{V}^{ak}\right]_{\partial_a \partial_k} \left[\lambda^k\right]_{\partial_k} \left[\mathcal{W}^{kb}\right]_{\partial_k \partial_b} , \tag{48}$$

where $k = a = b$ are the quantum numbers associated with each block in the intermediate link, directed from $\mathcal{V}$ to $\mathcal{W}$. The degeneracy dimensions $\bar{\partial}_k = \min\left\{\bar{\partial}_a, \bar{\partial}_b\right\}$ of the factorization can be truncated, e.g. by the exclusion of small singular values. To this end it may be necessary to first compute $\lambda^k$ for all blocks $k$ and then globally identify the singular values to be discarded among them. This aspect is particularly relevant when the implemented Abelian symmetry is a subgroup of a larger, possibly non-Abelian one (as we discussed in Sec. 3.1): singular values which are quasi-degenerate should either be all kept, or all discarded, even if they are distributed across different sectors, because they might form a multiplet of an underlying non-Abelian symmetry.

(iv) Compose the symmetric output tensors $\mathcal{V}^{ak}_{\partial_a \partial_k}$, $\mathcal{W}^{kb}_{\partial_k \partial_b}$ and $\lambda^k_{\partial_k}$ from all obtained degeneracy tensors (or subtensors therein if selected singular values shall be discarded):

$$\mathcal{V}^{ak}_{\partial_a \partial_k} := \left[\mathcal{V}^{ak}\right]_{\partial_a \partial_k} \times \delta_{ak}, \tag{49a}$$

$$\mathcal{W}^{kb}_{\partial_k \partial_b} := \left[\mathcal{W}^{kb}\right]_{\partial_k \partial_b} \times \delta_{kb}, \tag{49b}$$

$$\lambda^k_{\partial_k} := \left[\lambda^k\right]_{\partial_k}. \tag{49c}$$

If desired, the direction of the intermediate link can be inverted afterwards.

(v) With the help of the fuse-nodes obtained in step (i), split the links $a$ and $b$ into the original links $\ell_1 \dots \ell_r$ on $\mathcal{V}$ and $\ell_{r+1} \dots \ell_n$ on $\mathcal{W}$.

We note that a QR decomposition follows the exact same procedure, yielding two output tensors, one of which is unitary with respect to the intermediate link. Since the computational costs of QR- and singular value decomposition behave highly nonlinearly in their operands' dimensions, their symmetric versions can gain a lot in efficiency by operating independently on multiple smaller blocks instead of one large matrix. For example, a square matrix defined over two similar links comes at a full decomposition cost of $\mathcal{O}\left(d_a^3\right)$ in the absence of symmetric inner structure. On a symmetric tensor one can gain a factor $\bar{a}^2$, quadratic in the number of active sectors of a flat or only slightly peaked distribution of degeneracy dimensions. Furthermore, these block-wise operations allow for a natural parallelization of the algorithm implementation.

### 3.5.3 Computational Cost Estimation

The resources spent in a symmetric tensor operation depend on the number and dimensions of the degeneracy tensors, and are thus highly problem-dependent. Nevertheless, we can isolate common scenarios and give cost estimates for the respective operations. As it will turn out, the symmetric operations presented above often display a highly favorable computational complexity compared to equivalent ordinary tensor operations. The latter usually constitute the worst-case scenario for symmetric tensors. Therefore, provided that data structures and

algorithms with only a negligible overhead are used for the internal quantum-number book-keeping (as outlined in Sec. 3.3), one will always benefit from exploiting the additional inner structure.

We stress that all basic tensor operation listed in Sec. 3.5, on the level of degeneracy tensors, make use of the ordinary procedures presented for non-symmetric tensors. Therefore, we can easily estimate the cost of symmetric operations. Namely, if the complexity of a certain operation on degeneracy tensors scales like

$$\mathcal{O}_{\mathrm{deg},\ell_1\dots\ell_n} = \mathcal{O}\Big(\bar{\partial}_{\ell_1}^{p_1}\cdots\bar{\partial}_{\ell_n}^{p_n}\Big), \tag{50}$$

where $p_r$ are the maximal power coefficients for the degeneracy dimensions $\bar{\partial}_{\ell_r}$ and, in total, $n$ links are involved, then its cost on a symmetric tensor is bounded by

$$\mathcal{O}_{\mathrm{sym}} = \sum_{\ell_1\dots\ell_n} \mathcal{O}_{\mathrm{deg},\ell_1\dots\ell_n} \times \hat{\bar{\delta}}_{\ell_1\dots\ell_n}, \tag{51}$$

where $\hat{\bar{\delta}}_{\ell_1\dots\ell_n} \in \{0,1\}$ is an operation-specific symmetry constraint on the involved quantum numbers. Depending on the operation under consideration (we saw complexity examples in Sec. 3.5.2), this constraint can be even stricter than the "structural tensor"-constraint $\tilde{\delta}$ formulated in Eq. (34) (which, for example, would be valid for a block-wise operation on a single tensor). In any case, we have $\hat{\bar{\delta}}_{\ell_1\dots\ell_n} \leq \tilde{\delta}_{\ell_1\dots\ell_n}$.

We can therefore provide an upper bound for the computational cost of a symmetric operation:

$$\mathcal{O}_{\mathrm{sym}} \leq \sum_{\ell_1\dots\ell_n} \tilde{\delta}_{\ell_1\dots\ell_n} \times \mathcal{O}\Big(\bar{\partial}_{\ell_1}^{p_1}\cdots\bar{\partial}_{\ell_n}^{p_n}\Big) \leq \mathcal{O}\left(\left(\sum_{\ell_1}\bar{\partial}_{\ell_1}\right)^{p_1}\cdots\left(\sum_{\ell_n}\bar{\partial}_{\ell_n}\right)^{p_n}\right) = \mathcal{O}_{\mathrm{full}}, \tag{52}$$

where $\mathcal{O}_{\mathrm{full}}$ is the complexity of the ordinary, non-symmetric operation. We observe that the symmetric operations outperform the ordinary ones for two reasons:

(i) Thanks to the constraints $\hat{\bar{\delta}}$ posed by symmetry (invariance), the number of block-wise operations between degeneracy tensors, $S = \sum_{\ell_1\dots\ell_n} \hat{\bar{\delta}}_{\ell_1\dots\ell_n}$, is rigorously restricted. Due to the sparsity of the allowed block-wise operations on degeneracy tensors, we can naively upper-bound their number by $S \leq \bar{\ell}_1\cdots\bar{\ell}_n$. However, since those operations are only performed over symmetry-invariant quantum numbers, they at least fulfill the constraint $\tilde{\delta}$. This allows us to further tighten down the number of combinations contributing to $S$:

$$S \leq \bar{\ell}_1\cdots\bar{\ell}_n/\bar{\ell}_{\mathrm{max}}, \tag{53}$$

where $\bar{\ell}_{\mathrm{max}} = \max_r \bar{\ell}_r$ is the largest number of sectors found among all links $r = 1,\dots,n$. When specifying an operation, one can possibly formulate even tighter restrictions, as we encountered e.g. for the symmetric contraction (Sec. 3.5.2). In practice, one often finds even smaller $S$ due to non-present matches.

(ii) The most costly operations are those that scale nonlinearly ($p_r > 1$) with the dimensions. Those operations are also the ones which benefit most from operating on smaller blocks.

In order to provide a more concrete estimate of computational costs, we need to account for the actual degeneracy dimensions $\bar{\partial}_{\ell_r}$. To this end, we sketch two extremal and one common degeneracy distributions of symmetric links and study their cost estimates for all basic symmetric tensor operations. In particular, we demonstrate that we can benefit the more from symmetric tensors operations, the broader the indices of links are spread over different quantum numbers.

**Flat distribution.** We define a *flat distribution* as a distribution characterized by constant degeneracy dimensions $\bar{\partial}_{\ell_r} = d_r / \bar{\ell}_r$ which only depend on the total dimension $d_r$ and the number of sectors (different quantum numbers) $\bar{\ell}_r$ of a given link $r$, but not on the quantum numbers $\ell_r$ itself. Consequently, all degeneracy tensor operations come at a similar cost, totaling to

$$\mathcal{O}_{\text{sym}} = \mathcal{O}\left( \bar{\partial}_{\ell_1}^{p_1} \cdots \bar{\partial}_{\ell_n}^{p_n} \cdot S \right) \tag{54a}$$

$$= \mathcal{O}\left( d_1^{p_1} \cdots d_n^{p_n} \cdot \bar{\ell}_1^{-p_1} \cdots \bar{\ell}_n^{-p_n} \cdot S \right) \tag{54b}$$

$$= \mathcal{O}_{\text{full}} \cdot \mathcal{O}_{\text{gain}}. \tag{54c}$$

$\mathcal{O}_{\text{full}}$ is again the complexity of the equivalent non-symmetric operation, and we separate all improvement from using symmetric tensors into $\mathcal{O}_{\text{gain}} = \mathcal{O}\left( \bar{\ell}_1^{-p_1} \cdots \bar{\ell}_n^{-p_n} \cdot S \right)$. Using Eq. (53), which was derived for unary operations on a single symmetric tensor, results in a conservative upper bound of

$$\mathcal{O}_{\text{gain}} \leq \mathcal{O}\left( \bar{\ell}_1^{1-p_1} \cdots \bar{\ell}_n^{1-p_n} / \bar{\ell}_{\text{max}} \right), \tag{55}$$

i.e. the computational cost scales at least reciprocally in the maximal number of sectors $\bar{\ell}_{\text{max}}$, but possibly significantly better depending on the operation ($p_r$, $S$) and the number of non-present matches in the tensor(s).

Consider for example the memory consumption of a symmetric tensor in a flat degeneracy distribution. The number of elements in Eq. (36) is linear in the degeneracy dimensions (meaning $p_r = 1$), hence according to Eq. (55) this number is reduced by a factor of $\bar{\ell}_{\text{max}}$ compared to the number of elements in an ordinary tensor.

Among the degeneracy distributions discussed here, the flat distribution is optimal for symmetric tensor operations, because it allows computations to benefit from the additional inner structure to the greatest possible extent. Tensors with this distribution usually contain many small degeneracy tensors, which significantly reduces the cost in operations.

The flat distribution is occasionally realized in symmetries with small finite numbers of available quantum numbers (e.g. $\mathbb{Z}_n$ with small $n$), such as in the Ising model (see Sec. 5.3).

**Highly peaked distribution.** In the extremal case of a narrowly peaked distribution, we can expect one degeneracy tensor to have dimensions much larger than all the others. The computational cost is then dominated by these maximal degeneracy dimensions, approaching the full dimensions $d_r$ of the peaked links. In this case, the cost of performing the symmetric operation becomes roughly equal to performing the equivalent operation on non-symmetric tensors: $\mathcal{O}_{\text{sym}} = \mathcal{O}_{\text{deg}} = \mathcal{O}_{\text{full}}$. In this worst-case scenario, symmetric tensors cannot distinguish themselves by any relevant inner structure any more, and basic operations cannot benefit from symmetry invariance in a noticeable way.

**Slightly peaked distribution.** The most common scenario in groups with more than a few accessible quantum numbers is a distribution of degeneracy over a wider, but effectively limited, range of quantum numbers $\bar{\ell}_{\text{eff}}$. If the degeneracy dimensions are more or less evenly distributed within such a range, we can resort to the results obtained for the flat distribution, replacing $\bar{\ell}$ by the distribution width $\bar{\ell}_{\text{eff}}$. Sometimes, degeneracy concentrates around certain quantum numbers, with a distribution resembling e.g. a Gaussian. The narrower such a distribution becomes, the more the cost will be dominated by the largest degeneracy dimensions in the involved links. Then, the effective number of active quantum numbers $\bar{\ell}_{\text{eff}}$ on those links becomes very small, resulting in less advantageous cost estimates. We will encounter such scenarios in Sec. 5.3.

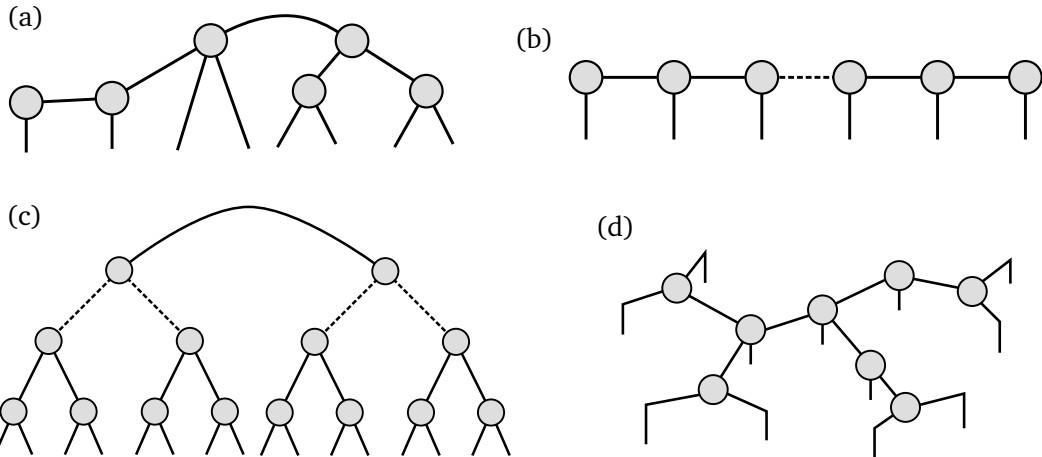

Figure 24: Loop-free TN geometries: (a) a generic tree tensor network, (b) an MPS, (c) a binary tree tensor network (bTTN), (d) a molecular TTN, shaped according to an arbitrary loop-free graph geometry, finding application in physical chemistry.

# 4 Loop-Free Tensor Networks

In this section, we address general properties of loop-free tensor networks, i.e. TN states whose network graphs contain no cycles. In Fig. 24 some examples are given, including the linear MPS [25, 27], the hierarchical binary Tree Tensor Network (bTTN, discussed in detail in Sec. 5.2) [70, 73, 75], and the flat tensor product state, which finds applications in quantum chemistry [181].

Tensor networks without cycles can exploit the full power of the TN gauge transformations: Smart gauging establishes rules on the network which can tremendously reduce the number of necessary contractions, thereby enhancing computational speed (see Fig. 25). It has been argued that the computational advantage of working in such a loop-free framework in fact enables the use of larger link dimensions, and, in turn, may partially compensate for the lower scaling of entanglement captured with respect to a "loopy" tensor network [182]. The textbook example is the comparison between a tree tensor network [70, 73–75, 183], which allows for efficient algorithms, but suffers from the fact that the maximal entanglement between two subsystems depends on the chosen bipartition of the system (entanglement clustering) [184], and a MERA [64–67] where the entanglement distribution is "smooth", but calculations are expensive.

Loop-free TNs exhibit advantages also in relation to symmetries. In fact, while upgrading a generic TN (describing a covariant state) to its symmetric version may require an overall increase in link dimensions [180], for loop-free TNs the upgrade can always be performed in a link dimension-preserving way (see a proof in Sec. 4.2.4).

In the following, we first introduce some general aspects of the structure of loop-free TNs (Sec. 4.1). We then discuss different possible gauges in loop-free TNs and their application (Sec. 4.2). Finally, we show how to randomly initialize symmetric ansatz states for loop-free TNs (Sec. 4.3.1).

## 4.1 General Aspects of Loop-Free Tensor Networks

The underlying graph structure of a loop-free TN defines a distance $\text{dist}(a, b)$ between any two nodes $a$ and $b$ of the network, which is equal to the number of links encountered on the

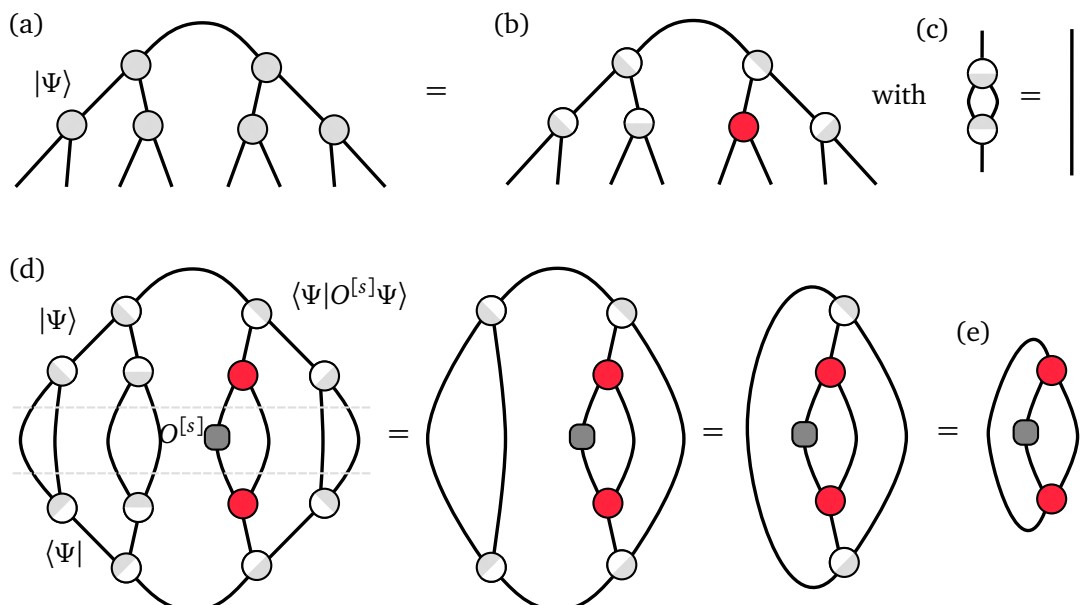

**Figure 25:** (a) A binary tree TN as an example of a loop-free TN describing some QMB state $|\Psi\rangle$: Writing it in a certain gauge (b) imposes certain rules (c) on the network. Knowledge about the gauge then allows us to simplify the numerical calculations, here the calculation of the expectation value of some local observable $\langle\Psi|O^{[s]}|\Psi\rangle$. The bra TN is drawn as the vertically flipped version of the ket TN, indicating hermitian conjugation of all its tensors (d): By applying the rules, we see that most contractions (which would be necessary in a network without a gauge) do not have to be performed explicitly numerically. We are left with a much simpler network (e) where only the contractions that cannot be "annihilated" using gauge rules have to be executed.

shortest path between them. In particular, for loop-free TNs such a path is always unique. It is practical to call all nodes which share the same distance $d$ to some specified center node $c$ a *level*, $L(d;c)$.

We further introduce the maximal link dimension $D := \max_\eta\{D_\eta\}$ among all virtual links, which usually exceeds the physical link dimensions $\{d_s\}$. As it will become apparent in the following, the maximal link dimension plays a fundamental role in any TN ansatz. The computational complexity of algorithms operating on loop-free TNs is a smooth polynomial of $D$. This can be seen, for instance, in the calculation of the scalar product between two QMB states represented in the same loop-free TN geometry. Following a level-structure towards some center node, one can successively contract all tensors at the currently outermost level of the ket-state with their bra-state counterparts into temporary nodes, while absorbing any temporary nodes on links in between. The cost of these contractions is then upper-bounded by $\mathcal{O}(D^{Z+1})$, where $Z$ is the maximal number of links of a tensor in the network. Tighter bounds can be identified in specific situations, e.g. when certain link dimensions are known to differ substantially from $D$ (such as in an MPS, where one link at each tensor carries the physical dimension).

Another immediate consequence of the absence of loops is that every link between two nodes $q$ and $q'$ induces a bipartition of the QMB system. The Hilbert spaces of the two partitions are spanned by the canonical basis states $\{|i_s\rangle : \text{dist}(s,q) < \text{dist}(s,q')\}_s$ and $\{|i_s\rangle : \text{dist}(s,q) > \text{dist}(s,q')\}_s$, respectively, i.e. by fusing all physical links $s$ ultimately connected to either $q$ or $q'$ after removing the bipartition link between the two nodes from the network.

At any virtual link $\eta$, we can therefore rewrite the loop-free TN state by means of a Schmidt decomposition

$$|\Psi\rangle = \sum_{k=1}^{D_\eta} |A_k\rangle |B_k\rangle \lambda_k \,, \tag{56}$$

where $|A_k\rangle$ and $|B_k\rangle$ are states described in the (contracted) TNs of the two partitions and we assume, without loss of generality, that the Schmidt values $\lambda_k$ are sorted descendingly. On this basis, we make the following observations:

- A loop-free TN can be viewed as a simultaneous Schmidt decomposition over all virtual bonds. This point of view is emphasized in the canonical gauge of Sec. 4.2.3.

- At each network bipartition, the bond dimension $D$ provides an upper bound to the Schmidt rank, which measures bipartite entanglement.

- Discarding the smallest singular values achieves the optimal single link compression. More precisely, suppose that given a virtual bond $\eta$, we wish to reduce its prior dimension $D_\eta$ to some smaller dimension $\chi_\eta$. We achieve this compression by simply truncating the summation at $k = \chi_\eta$ in Eq. (56) and discarding (setting to zero) the smallest Schmidt values in the decomposition. Thus, we obtain the following truncated state

$$|\Psi^{\text{trunc}}\rangle := \sum_{k=1}^{\chi_\eta} |A_k\rangle |B_k\rangle \lambda_k / \mathcal{N}_{\text{kept}} \,, \tag{57}$$

with normalization constant

$$\mathcal{N}_{\text{kept}} := \sqrt{\sum_{k=1}^{\chi_\eta} \lambda_k^2} \,. \tag{58}$$

Any other way of discarding $\lambda_k$'s leads to a larger error in quantum state fidelity, according to the Eckart–Young–Mirsky theorem [185]. Note that while being optimal for a single link, compressing a quantum state by discarding Schmidt values on *multiple* bonds simultaneously does not lead to the optimal fidelity (as each compression is a local non-unitary operation; see techniques for an enhanced compression approach, e.g. in Ref. [39] and Sec. 5.1.6). The most common purpose for link compression in loop-free TN algorithms, such as the ground state search outlined in Sec. 5.1, is to preserve the bond dimension $D$ throughout the simulation: This is often ensured by re-compressing bond dimensions after temporary growths.

- Similar to performing a bipartition through link-cutting, detaching an $n$-link network node from a loop-free TN induces a partition into $n$ disjunct subspaces. Each virtual index then contributes variational degrees of freedom in form of associated tensor elements.

## 4.2 Gauges in Loop-Free Tensor Networks

In Sec. 4.2.1 we review the freedom introduced by gauges specifically in the framework of loop-free TNs [26, 27, 39, 70]. We present the specific gauges that are related to the orthogonality properties of the tensors in the network. We dub these gauges *center gauges* and discuss in particular the *unitary gauge* (Sec. 4.2.2) and its refinement, the *canonical gauge* (Sec. 4.2.3). Both are of great practical and conceptual importance. They install isometries in each tensor, which strongly simplify contractions of the network with its conjugate (see again Fig. 25). They also allow for efficient local error estimation and efficient application of local operators in time-evolution algorithms and measurements. In variational algorithms, they maintain the network

state's normalization during local optimization (see Sec. 5). Conceptually, the canonical gauge provides an understanding of the loop-free TN ansatz as simultaneous, independent, truncated Schmidt decompositions over all lattice bipartitions induced by the network geometry.

Then, we introduce the *symmetry gauge* (Sec. 4.2.4): We demonstrate that any loop-free TN ansatz state that transforms in- or covariantly under a global pointwise symmetry (as introduced in Sec. 3.4) can be recast into a symmetric TN, i.e. made entirely of symmetric tensors, through gauge transformations. Remarkably, this is compatible with both unitary and canonical gauge, meaning that all those gauges can be installed at the same time, and without increasing the bond dimension. Once installed, the symmetry gauge is strictly protected for any algorithm formulated via the symmetric tensor operations that we described in Sec 3.5. This last property is shared by the *real gauge* (Sec. 4.2.5), which can be useful in the presence of time-reversal symmetry.

### 4.2.1 Gauge Freedom

When searching for the optimal gauge, it is helpful to keep in mind that the whole freedom we have in choosing the tensor elements to encode a specific ansatz state $|\Psi\rangle$ in a loop-free network with given link dimensions is entirely determined by link-local gauge transformations as introduced in Sec. 1.7. This means that the only possible difference in the choice of tensors are matrices $X, Y$ inserted as products $X \cdot Y$ on virtual links and contracted into the adjacent tensors, as we demonstrate in the following.

**Gauges on Virtual Links and Minimal Bond Dimension.**  Let the TN have a virtual link $\eta$ of dimension $D_\eta$. We can perform a Schmidt decomposition over the original state $|\Psi\rangle$ between the two subnetworks connected by $\eta$, and possibly obtain a lower link dimension $\chi_\eta \leq D_\eta$. We now discuss how to gauge-transform the tensor network via local operations to obtain the explicit decomposition. The reduction of dimension is non-trivial as it requires the insertion of a pair of matrices $X$ and $Y$ of dimensions $(D_\eta \times \chi_\eta)$ and $(\chi_\eta \times D_\eta)$ on the link. Theoretically, the existence of the gauge matrices can be shown as follows:

(i) In the given TN, we contract over all links except for the selected bond $\eta$ and obtain two tensors $\tilde{\mathcal{A}}$ and $\tilde{\mathcal{B}}$ for the respective bipartitions. We can rewrite these tensors as matrices $\tilde{A}$ and $\tilde{B}$. As every physical site of the system belongs to one of the two partitions, we collect the physical degrees of freedom of the respective subspaces in indices $a$ and $b$. We can then write the coefficient tensor of the TN state as $\Psi_{a,b} = \sum_k \tilde{A}_{a,k}\tilde{B}_{k,b}$, where $k \in \{1, \ldots, D_\eta\}$ is the index of bond $\eta$.

(ii) Starting from the original state $|\Psi\rangle$, we can also compute an exact Schmidt decomposition $\Psi_{a,b} = \sum_{k=1}^{\chi_\eta} A_{a,k}B_{k,b}$, where we already absorbed the Schmidt values into the matrices $A$ or $B$ and the indices $a$ and $b$ label the physical degrees of freedom as above. $\chi_\eta$ denotes the Schmidt rank of $\Psi$ and we now assume that $\chi_\eta < D_\eta$, i.e. for the description of $\Psi$ a lower bond dimension than given in the TN above suffices.

(iii) We will now determine two gauge matrices $X$ and $Y$ which can be inserted on bond $\eta$ and then be absorbed into adjacent nodes, thereby reducing the original link dimension from $D_\eta$ to $\chi_\eta$, without modifying the encoded state. By exploiting $\tilde{A} \cdot \tilde{B} = A \cdot B$ we arrive at:

$$A = \tilde{A}\left[\tilde{B}B^{-1R}\right] =: \tilde{A}X \tag{59}$$

and

$$B = \left[A^{-1L}\tilde{A}\right]\tilde{B} =: Y\tilde{B}, \tag{60}$$

(a) (b) (c)

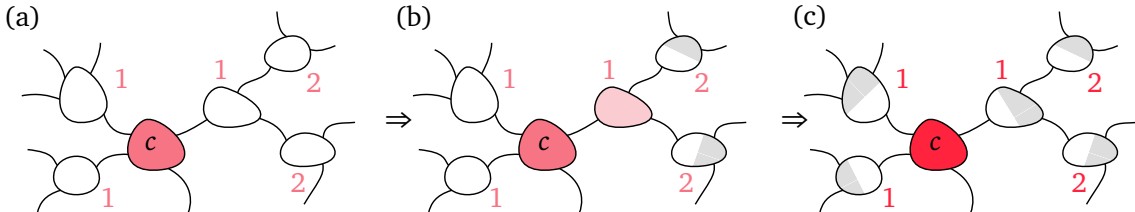

Figure 26: Example for a loop-free TN gauging procedure. The numbers indicate the distance of each node from the center $c$, determining the order of isometrization: Starting from the un-isometrized network (a), tensors with largest distance are isometrized first, and the next tensor towards the center is updated (b). Then the isometrization continues on the next level (c). This procedure is repeated until the center (red) is reached.

where the expressions in square brackets now define the anticipated gauges $X$ and $Y$. The right-inverse (left-inverse) $B^{-1R}$ ($A^{-1L}$) exists due to the full row-rank (column-rank) $\chi_\eta$ of the matrix $B$ ($A$) resulting from the Schmidt decomposition.

While $X \cdot Y \neq \mathbb{1}$ in general, these gauges have the property $Y \cdot X = \mathbb{1}$:

$$YX = A^{-1L}\tilde{A}\tilde{B}B^{-1R} = A^{-1L}ABB^{-1R} = \mathbb{1}. \tag{61}$$

(iv) By repeating the procedure on all virtual links of the network, we obtain the gauges that transform an arbitrary given TN into an equivalent TN of minimal bond dimensions equal to the Schmidt rank on the respective bond.

Consequently, two loop-free TN representations $TN_1$ and $TN_2$ of the same state $|\Psi\rangle$, with the same geometry, are connected by a link-local gauge transformation: We can explicitly construct two sets of local gauges $\{X_1, Y_1\}$ and $\{X_2, Y_2\}$ which transform either one into the (same) TN with minimal bond dimensions.

**Gauges on Physical Links.** On physical links, gauge transformations shall be restricted to unitary transformations to keep the associated basis states orthonormal. Such a change of basis can e.g. be helpful for entirely real Hamiltonians, as these allow us to solve eigenproblems with entirely real TNs, therefore consuming less computational resources. Gauge transformations on physical links can also be used to install the symmetry gauge (more on this in Sec. 4.2.4). Note that within symmetric TNs, we are restricted to symmetric gauge matrices, and the local basis can only be changed within degenerate manifolds associated to the quantum numbers.

### 4.2.2 Unitary Gauge

The unitary gauge is always defined with respect to a single selected node $c$ of the TN and therefore also referred to as "central gauge" (some literature refers to this gauge as the canonical gauge, while we prefer to adopt this term for the gauge defined in Sec. 4.2.3). Starting with nodes of maximal distance $d = d_{\max}$ from $c$, the unitary gauge can be installed on this level-structure $\{L(d;c) : d = 0, \ldots, d_{\max}\}$ as follows (see also Fig. 26):

(i) On each node $q \in L(d;c)$ perform the QR decomposition

$$\mathcal{T}^{[q]}_{\{s\},\{\alpha\},\alpha_\eta} = \sum_{\beta_\eta=1}^{\tilde{D}_\eta} \mathcal{Q}^{[q]}_{\{s\},\{\alpha\},\beta_\eta} \mathcal{R}^{[q]}_{\beta_\eta,\alpha_\eta} \tag{62}$$

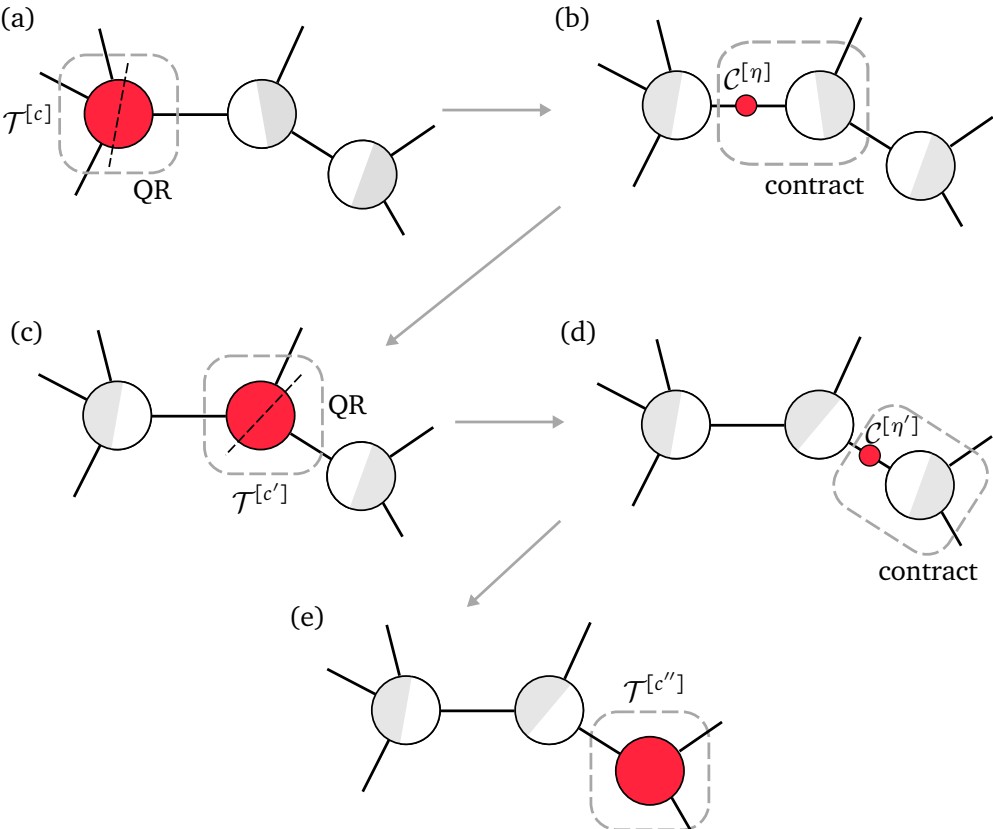

Figure 27: Moving the center in unitary gauge: For an excerpt from a possibly larger TN, we show how the gauge center (red) is moved from network node $c$ to $c''$. Network center-tensors on the path of links $\eta$, $\eta'$ between the old and the new designated center node are QR-decomposed (a,c) and the non-unitary parts $\mathcal{C}^{[\eta]}$, $\mathcal{C}^{[\eta']}$ are subsequently contracted into the respective next node on the path (b,d). Tensor isometrization (gray-white shading) is always directed to the current gauge center.

Figure 28: (a) Re-isometrization of a node in the unitary gauge (essentially steps (b)→(c) and (c)→(d) from Fig. 27), achieved by QR decomposition. (b) Formulation of the same transformation in terms of a link-local gauge transformation (see Sec. 1.7), by contracting the respective non-unitary center matrices $\mathcal{C}^{[\eta]}$ and $(\mathcal{C}^{[\tilde{\eta}]})^{-1}$ (both red) into their adjacent node. However, this is only possible if their (single-sided) inverses exist, i.e. if the bond dimensions equal the matrix ranks.

on the bipartition between the virtual link $\eta$ that leads towards $c$ and all remaining (physical and virtual) tensor links of $q$, which provide the remaining indices $\{s\}$ and $\{\alpha\}$.

(ii) Update $\mathcal{T}^{[q]} \to \tilde{\mathcal{T}}^{[q]} := \mathcal{Q}^{[q]}$ with the semi-unitary isometry of the QR decomposition.

(iii) Contract the (upper trapezoidal) matrix $\mathcal{C}^{[\eta]} := \mathcal{R}^{[q]}$ over the corresponding virtual link $\eta$ into the adjacent tensor from the inner layer $L(d-1;c)$.

(iv) Advance inwards to $d \to d-1$ and repeat from step (i) until $d = 0$ is reached.

After this procedure all tensors have been updated to $\tilde{\mathcal{T}}^{[q]}$. Also, virtual links might turn out reduced in bond dimension from $D_\eta$ to $\tilde{D}_\eta$, since the QR decomposition applied in step (i) automatically selects the minimum between $D_\eta$ and the product of the remaining link dimensions.

Note that the iteration presented above indeed implements a global gauge transformation, because it does not change the physical state of the network: Neither QR decompositions nor the propagation of $\mathcal{R}$-transformations over virtual bonds alter the contraction of the affected tensors over shared virtual links. Computational complexity is dominated by the QR factorizations (see Sec. 2.2.2), upper-bounded by $\mathcal{O}\left(D^{Z+1}\right)$, where $D$ is the maximal bond dimension and $Z$ the maximal number of links of a tensor.

For algorithms operating in unitary gauge, a common task is to move the center from one node $c_1$ to another node $c_2$ (as we will see in the variational search for ground states in Sec. 5). This can be achieved by a sequence of QR decompositions $\mathcal{T}^{[q]} = \tilde{\mathcal{T}}^{[q]}\mathcal{C}^{[\eta]}$ of all nodes $q$ on the path between $c_1$ and $c_2$, scaling linearly with the distance, $\mathcal{O}(\text{dist}(c_1, c_2))$. In every step, we obtain a possibly different non-unitary part $\mathcal{C}^{[\eta]}$ which is passed over the shared bond $\eta$ from the re-isometrized node $\tilde{\mathcal{T}}^{[q]}$ to the next tensor on the path towards $c_2$. This process is shown in Fig. 27, and yields the isometry relation of the unitary gauge, shown in Fig. 28(a),

$$\sum_{\beta_\eta} \mathcal{T}^{[q]}_{\{s\},\{\alpha\},\beta_\eta} \mathcal{C}^{[\eta]}_{\beta_\eta,\alpha_\eta} = \sum_{\beta_{\tilde{\eta}}} \tilde{\mathcal{T}}^{[q]}_{\{s\},\{\alpha\},\beta_{\tilde{\eta}}} \mathcal{C}^{[\tilde{\eta}]}_{\beta_{\tilde{\eta}},\alpha_{\tilde{\eta}}}, \tag{63}$$

where $\mathcal{C}^{[\eta]}$ and $\mathcal{C}^{[\tilde{\eta}]}$ are the (non-unitary) center matrices defined on any two links $\eta$ and $\tilde{\eta}$ of $q$ respectively, as they might be encountered on a path moving the center. This relation holds for every network node $q$ and transforms the tensor $\mathcal{T}^{[q]}$, which is isometric over $\eta$, into the tensor $\tilde{\mathcal{T}}$, which is isometric over $\tilde{\eta}$.

If we had access to the single-sided inverses $(\mathcal{C}^{[\eta]})^{-1}$, the re-isometrization in Eq. (63) could be performed by local contractions avoiding costly matrix factorizations (see Fig. 28(b)) and the unitary gauge transformations could be understood in terms of link-local gauges as discussed in Sec. 1.7. However, the inversion requires the matrix $\mathcal{C}$ to be of full rank, and we need to find a way to reduce the possibly oversized matrix dimensions first: There exists a very elegant gauge that realizes such minimal bond dimensions together with a very efficient re-isometrization rule. We will discuss this gauge in the following section.

### 4.2.3 Canonical Gauge

A canonical tensor network [26] is obtained from singular-value decomposing the unitary-gauge center matrices $\mathcal{C}^{[\eta]}$ on all bonds $\eta$, as in

$$\mathcal{C}^{[\eta]}_{\alpha_\eta,\beta_\eta} \overset{\text{SVD}}{=} \sum_{\gamma_\eta=1}^{\chi_\eta} \mathcal{V}^{[\eta]}_{\alpha_\eta,\gamma_\eta} \lambda^{[\eta]}_{\gamma_\eta} \mathcal{W}^{[\eta]}_{\gamma_\eta,\beta_\eta}. \tag{64}$$

The isometries $\mathcal{V}^{[\eta]}$ and $\mathcal{W}^{[\eta]}$ merely act as link-local gauge transformations on $\eta$ (and pairs of the form $\mathcal{V}^{[\eta]}\mathcal{V}^{[\eta]\dagger}$ or $\mathcal{W}^{[\eta]\dagger}\mathcal{W}^{[\eta]}$ can therefore be introduced on the respective links at will, as

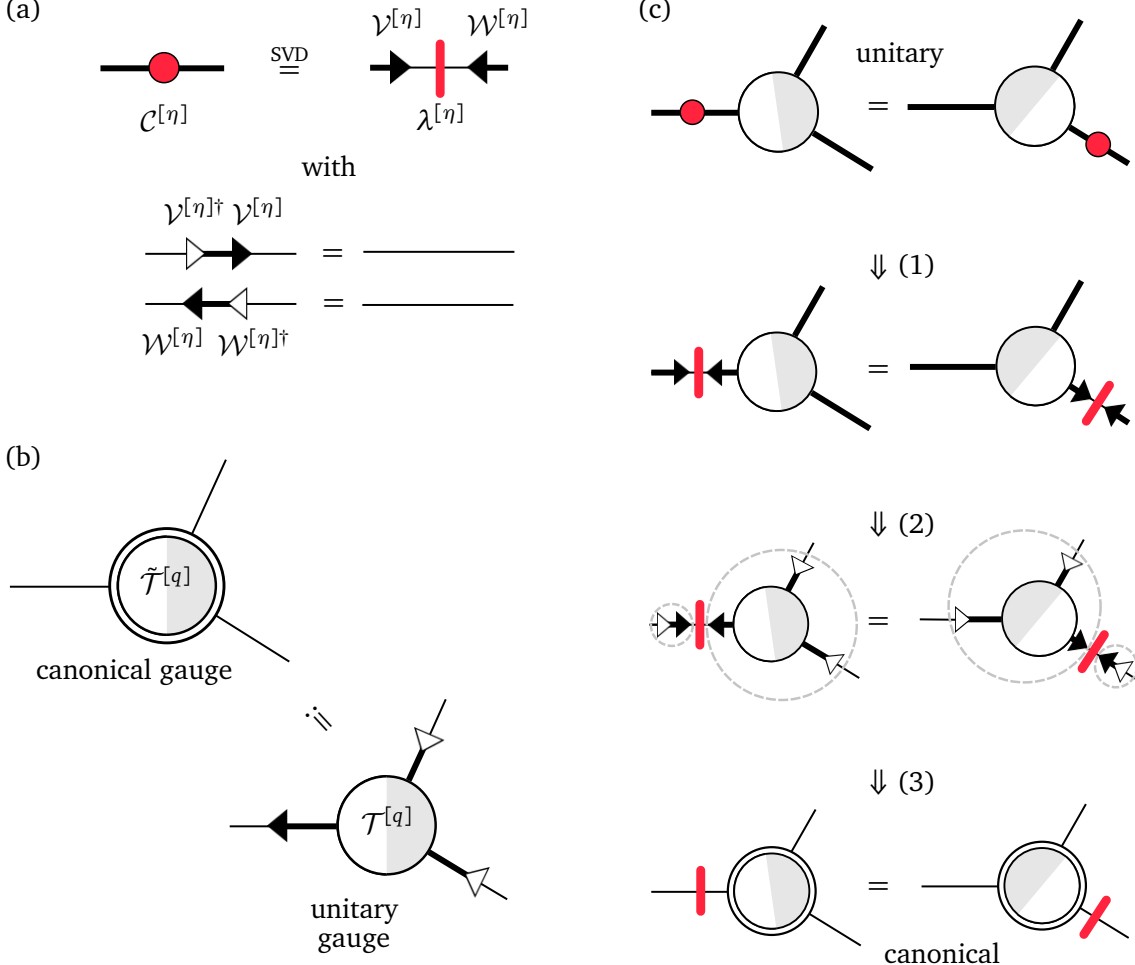

Figure 29: Transformation from unitary- to canonical gauge: (a) We define isometries $\mathcal{W}^{[\eta]}$, $\mathcal{V}^{[\eta]}$ (black triangles, pointing towards the center singular values) and their one-sided inverses (white) from left- and right singular vectors of the unitary-gauge center matrices $C^{[\eta]}$ (red) on all virtual links $\eta$ in a TN. (b) The canonical-gauge tensor (double outline) is obtained from the unitary-gauge tensor by absorbing, from all virtual links, those isometries that point towards the current TN gauge center $C^{[\eta]}$ sitting on some network link $\eta$. Inverses are taken if tensor links lead away from the center. When repeated on each tensor, this procedure establishes the canonical gauge by insertion of link-local gauges $\mathcal{V}\mathcal{V}^{\dagger}$ or $\mathcal{W}^{\dagger}\mathcal{W}$ on each non-center virtual link (cf. Fig. 30), and $\mathcal{V}^{[\eta]}\lambda^{[\eta]}\mathcal{W}^{[\eta]}$ on the center link $\eta$. (c) The isometry relation of the canonical gauge (bottom) follows from the isometry relation in the unitary gauge (top): We (1) perform SVDs on the center matrices and (2) multiply all (virtual) links on both sides with the respective inverse isometries pointing towards the tensor node. (3) By exploiting (a) and (b), we recover an identity and the definition of a canonical tensor. The result (bottom) is the isometry relation in the canonical gauge picture. In order to visualize the compression to minimal bond dimension, entailed by the SVD, we assign different line widths to links of different dimension.

Figure 30: Proof that pairs of isometries $\mathcal{W}^\dagger\mathcal{W}$ act indeed as link-local gauges in the transformation from unitary- to canonical gauge (see also Fig. 29(b)): In unitary gauge, we first move the center (red) onto the selected virtual link (here depicted with two adjacent nodes as part of a larger TN) and compute its SVD (top left to right). The SVD yields a left-unitary term $\mathcal{V}$ and a right-unitary term $\mathcal{W}$ with $\mathcal{V}^\dagger\mathcal{V} = \mathcal{W}\mathcal{W}^\dagger = \mathbb{1}$, but the commuted products do not yield identities in general. We then insert such an identity (here $\mathcal{W}\mathcal{W}^\dagger$), restore the center matrix, and move it back to its original position (bottom left to right). This leaves the gauges $\mathcal{W}^\dagger\mathcal{W}$ inserted on the link, while none of the steps changed the network state. The same is true for an insertion of $\mathcal{V}\mathcal{V}^\dagger$.

shown in Fig. 30). They are absorbed into the adjacent nodes, thereby updating the network tensors $\mathcal{T}^{[q]} \to \tilde{\mathcal{T}}^{[q]}$ to the canonical tensors as illustrated in Fig. 29. Singular values that are zero or fall below a numerical threshold are discarded, so that the virtual links $\eta$ attain their minimal bond dimensions $\chi_\eta \leq D_\eta$ equal to the respective center-matrix ranks.

A practical approach to installing the canonical gauge does not have to find the center matrices $\mathcal{C}^{[\eta]}$ on all bonds explicitly. It is sufficient to install the unitary gauge with a single center-bond $\eta$ towards which all tensors are isometries. Then a series of SVDs (instead of simple QRs) for moving the unitary center, performed in reverse order from the center outwards, establishes the center matrix successively on all bonds (as in Fig. 31).

Due to the orthonormality induced by the isometries, every SVD of the center matrix, which can be considered as a reduced amplitudes tensor, also achieves a Schmidt decomposition of the complete TN state, with Schmidt values $\lambda^{[\eta]}$ and Schmidt rank $\chi_\eta$ on every virtual link [52]. Hence, the canonical gauge can be viewed as the explicit and simultaneous Schmidt decomposition of the QMB state on all lattice bipartitions induced by the loop-free network geometry.

By uniqueness of the Schmidt decomposition (up to phases and degeneracies) when ordering the Schmidt values descendingly, the canonical gauge becomes similarly unique. This gauge implements minimal bond dimensions simultaneously on all network links, which cannot be reduced further without altering the state. The main reason for actually installing the more costly canonical gauge is that the isometry relation Eq. (63) in this case translates into a mere multiplication with diagonal matrices (which can be implemented as a scaling of tensors with link-weights, cf. Sec. 2.2.2), as shown in Fig. 32:

$$\mathcal{T}^{[q]}_{\{s\},\{a\},\alpha_\eta} \lambda^{[\eta]}_{\alpha_\eta} = \tilde{\mathcal{T}}^{[q]}_{\{s\},\{a\},\alpha_{\tilde{\eta}}} \lambda^{[\tilde{\eta}]}_{\alpha_{\tilde{\eta}}} . \tag{65}$$

Again, the tensor $\mathcal{T}^{[q]}$, which originally is isometric over $\eta$, turns into an isometry $\tilde{\mathcal{T}}^{[q]}$ over another of its virtual links $\tilde{\eta}$. As we can easily afford to store and invert the Schmidt values

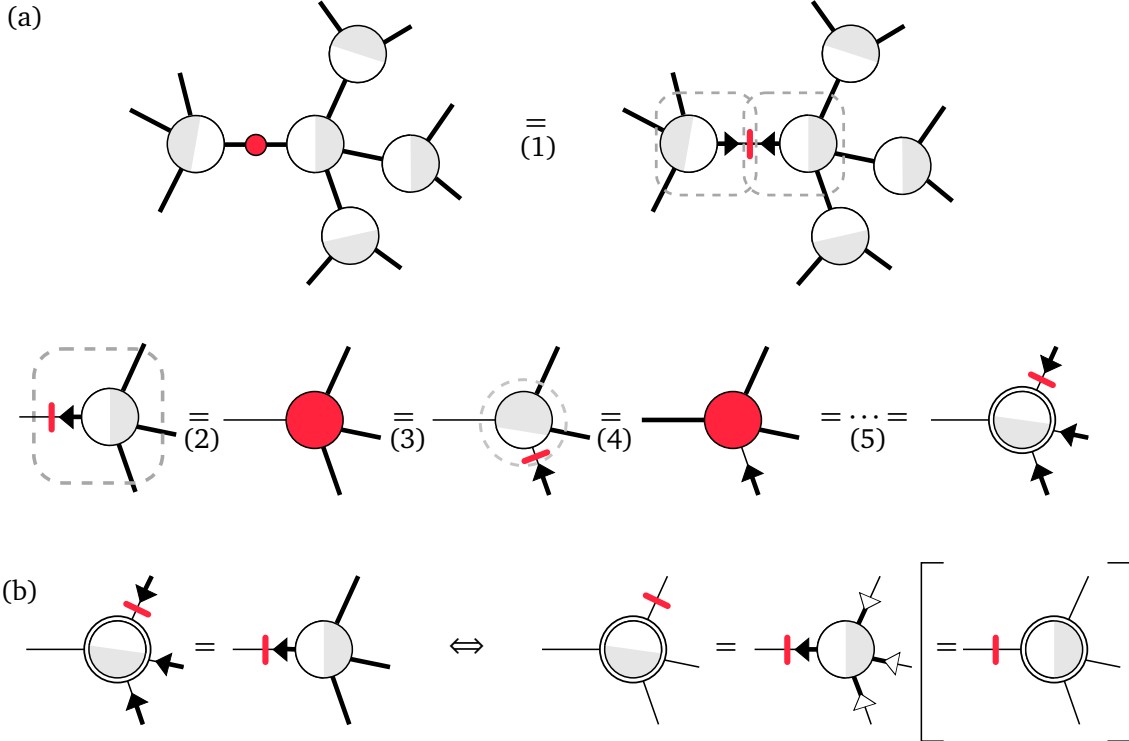

(a)

(b)

Figure 31: (a) Practical installation of the canonical gauge. (1) The gauge center (always red) of a TN in unitary gauge is SVD-decomposed. Starting with the nodes adjacent to the center, we perform the following steps (resembling a higher-order SVD [186]): (2) Absorb unitary matrix and singular values from parent link, (3) perform SVD with respect to a first child link, (4) re-absorb singular values, (5) repeat previous two steps on remaining child links. The final tensor (without the unitaries on the child links) is now in canonical gauge. (b) Comparison of the final and the initial tensor allows us to recover the definition of the canonical gauge.

Figure 32: Re-isometrization of a node in the canonical gauge: If we store the Schmidt values $\lambda^{[\eta]}$ (red) for all links of a network, we can perform a re-isometrization of a node (and consequently of the full network) using only numerically efficient contractions with these "link-weights" (see Sec. 2.2.2) and their inverses. This approach is much faster than the iterative sequence of QR decompositions in Fig. 27. Due to the minimal bond dimension in canonical gauge, the inversion is always possible. Moreover, the Schmidt values can be efficiently stored.

$\lambda^{[\eta]}$ of all bonds, we are able to quickly re-isometrize the TN with only local operations. A possible danger when implementing this step is that the inversion of small Schmidt values (if not discarded in a compression) can result in numerical instabilities. However, workarounds for this issue have been discussed [187].

The re-isometrization relation of the canonical gauge allows for parallelized algorithms (see Ref. [52]) and eliminates directional dependency of errors, arising for instance in the compression of virtual links performed by many TN algorithms. Therefore, the canonical gauge is a favorable choice in algorithms driven by (local) unitary evolution, e.g. real-time evolution algorithms, because a unitary operation does not alter the Schmidt decomposition except for the bonds it acts on [52]. Hence, (costly) re-installation can be avoided. To a certain extent, this can also be true for operations which are only approximately unitaries or identities. This includes the aforementioned compression of virtual links, if it is carried out by truncated SVDs which locally restore the gauge.

### 4.2.4 Symmetry Gauge

In the symmetric gauge, all tensors in the TN representing a covariant state become symmetric tensors as discussed in Sec. 3.3. Contrary to the gauges discussed so far, its installation may require a (fixed) transformation in the physical basis. While in practice one usually never explicitly transforms into symmetry gauge, and no simple recipes exist for this operation, we show that such a transformation is theoretically possible for any TN state:

(i) Identify the group of unitary, global pointwise symmetry operations $\{V_g : g \in \mathcal{G}\}$ and the global quantum number $\ell_{\text{select}}$ of the encoded QMB state.

(ii) The global pointwise operations $V_g^{[j]}$ define on every physical site $j$ a representation of $\mathcal{G}$ itself. Focusing again on Abelian groups for simplicity, we can diagonalize $V_g^{[j]}$ and obtain a new basis of possibly degenerate irrep states $|\tilde{s}\rangle_j$. We choose these as our new physical basis states.

(iii) Contract the whole TN into a single tensor $\Psi$. This tensor is now by definition invariant or covariant and can directly be written as a symmetric tensor as in Eq. (31). In the covariant case, i.e. in the case $\ell_{\text{select}} \neq 0$, we need to attach a corresponding quantum-number selector link to this tensor.

(iv) Decompose this tensor again into a TN of the original geometry, resorting to exact symmetric tensor decompositions as described in Sec. 3.5.2. The details of these decompositions (bond dimensions etc.) are irrelevant. We can always obtain the original (loop-free) geometry, representing the same state (because basic decompositions do not alter the state); hence we have already shown that it is a gauge transformation.

(v) Install the canonical gauge in the symmetric TN. As this task requires only basic QR, SVD, fusing (and subsequent splitting) operations, and contractions — all of which are well-defined symmetric tensor operations — we can always install the canonical gauge in a symmetric TN. By uniqueness of the Schmidt decomposition (up to degeneracy and ordering of singular values), we can obtain the same minimal bond dimension as in a non-symmetric TN. Note that the local basis transformations in step (ii) explicitly do not alter the bond dimensions because, being local operations, they cannot increase the entanglement (Schmidt rank).

Note that such a construction is also possible in loopy geometries. However, the canonical gauge of Sec. 4.2.3 does not exist in such networks and the loops introduce further constraints

on symmetric links that may lead to larger bond dimensions. The symmetric tensor ansatz can still be beneficial for a loopy network, but a trade-off between the speed-up through the use of symmetric tensors and the unfavorable increase of bond dimensions must be made [180].

### 4.2.5 Real Gauge

In the *real gauge* all tensor elements are in the real domain. As such, it is not restricted to loop-free networks. If it can be installed in the tensors of a TN state and the operators (Hamiltonian) acting on it, the real gauge can increase computational speed and reduce the variational space by 1/2. This is, e.g., the case in diagonalization problems, where it can usually be installed in the presence of a time-reversal symmetry (see Appendix A). As for the symmetry gauge, the TN operations we are considering can always be implemented such that they yield a real output if the input is real.

When time-reversal *and* global pointwise Abelian symmetries are present, one can often find a local basis transformation that simultaneously installs the real gauge on top of the symmetry gauge, by exploiting the freedom of phases, or more generally unitary transformations in degeneracy spaces, in the local physical basis.

## 4.3 Preparation of Symmetric TN Ansatz States

The initialization of symmetric tensor network states presents an additional level of difficulty with respect to initializing ordinary TNs: This is due to the fact that we need to assign a symmetry group representation to every virtual link, and these representations must be such that the symmetric tensors contain actual matches. In the pathological scenario of a tensor not containing any match due to the link representations, the TN is representing no state at all (zero vector). As as rule of thumb, the more matches are allowed, the less information in the TN is wasted. Constructing meaningful initial representations for virtual links is in general a non-trivial, non-local problem that involves fusion rules of the symmetry, but can nevertheless be treated on a general ground.

The simplest approach is to initialize the symmetric TN state as a *product state*, i.e. in any configuration which respects the symmetry and for which every virtual link holds only a single sector of degeneracy one. Consider, for instance, a system with $N$ physical sites, each of which can hold up to $n$ particles, and a U(1)-symmetry, which fixes the total particle number $N_p$. We could assign these particles to physical sites and, following the selection rules of the symmetry, assign a single sector of degeneracy one to every virtual link of the network.

In a more sophisticated approach, we could use knowledge about the expected spatial particle distribution in the state of interest (usually the ground state). As discussed earlier, any link $\eta$ of a loop-free tensor network introduces a bipartition of the underlying physical system into e.g. $x$ and $N - x$ sites. Typically, the expected value of particles in the subsystems then establishes the most relevant sector of $\eta$ with the largest degeneracy. Around that sector, and in the range of possible quantum numbers, one can now distribute degeneracy dimensions with a certain variance until the maximal bond dimension is exhausted. Suppose we have no prior knowledge about the distribution of the particles and their correlations. For fermions or hard-core bosons we could then make a binomial ansatz and consider initializing the sector $\ell$ (equal to the number of particles in the subsystem of length $x$) with degeneracy

$$\bar{\partial}_\ell \propto \min\left\{\binom{x}{\ell}, \binom{N-x}{N_p-\ell}\right\}. \tag{66}$$

Additional information, such as a harmonic confinement of the system, can be taken into account by centering the largest degeneracies around particle expectations from e.g. a Thomas-Fermi density distribution [188]. If we expect an insulating behavior where the quantum numbers

SciPost Phys. Lect. Notes 8 (2019)

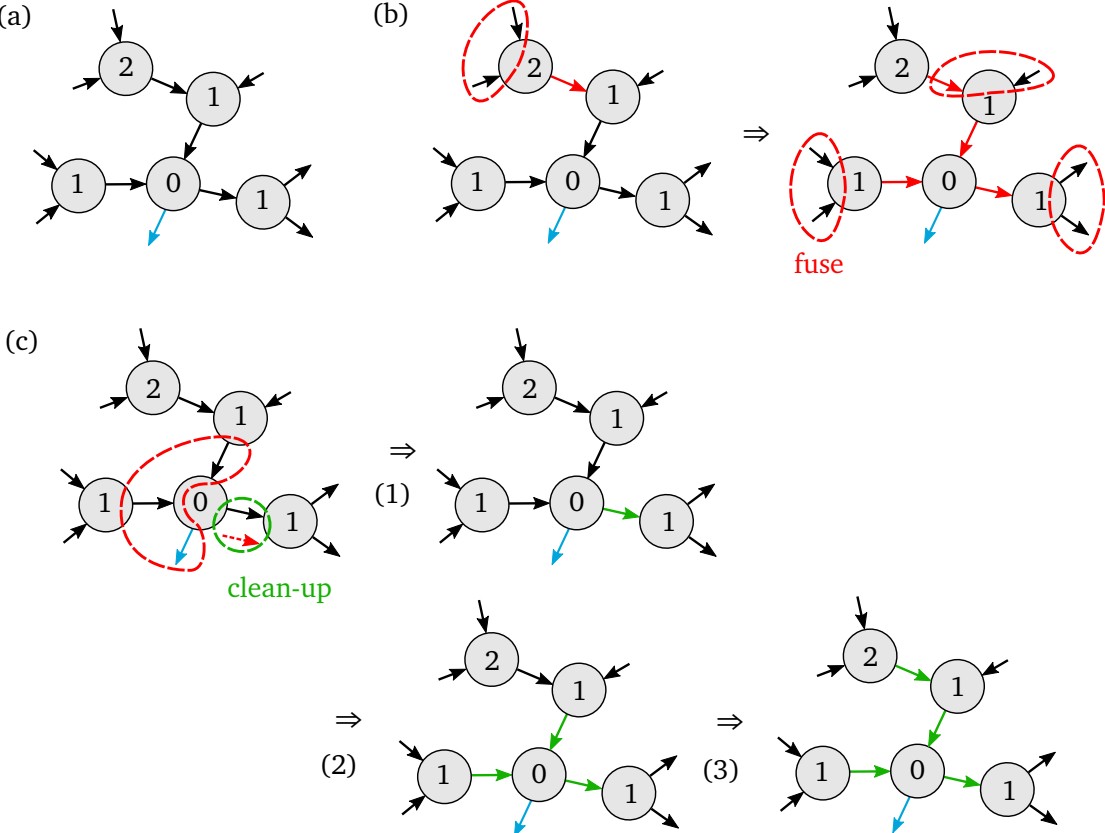

Figure 33: First steps in the preparation of a randomized symmetric ansatz state: (a) Set up a level-structure, assign directions for symmetric links and equip the TN with a selector link (blue). (b) Build up virtual links by means of fusing: Starting from the physical links, fuse links at nodes of the highest level to obtain the links towards the next lower level; successively repeat this procedure until reaching the center. (c) Clean-up round: (1) Intersection (green circle) with fused link (red dashed) yields cleaned-up link (green). (2) Clean up remaining virtual links at the center node. (3) Propagate the clean-up outwards.

are spatially locked, a narrowly peaked degeneracy distribution with small variance might be a good ansatz. Obviously, such initialization schemes are highly problem-dependent.

As a very general initialization method, we now introduce a scheme based on randomization, which makes no *a priori* assumption and thus can be always adopted.

### 4.3.1 Randomized Symmetric Ansatz States

In this construction, we start with a loop-free TN where the geometry, the physical links and the selector link are defined and a maximal bond dimension $D$ is given, while other virtual links and symmetric tensors are to be constructed. The recipe we present guarantees that the network is non-trivial in the end, and that all the degeneracy tensors in the network may indeed contribute to the state (and do not drop out in a contraction to the complete amplitudes tensor). This recipe can be applied to an arbitrary symmetry group. It makes extensive use of the symmetric link clean-up operation defined in Sec. 3.5.1, and it is sketched in Fig. 33.

(i) We choose a center node $c$ and assign a level-structure to the network, as we discussed in Sec. 4.1.

(ii) Starting from the outer levels (branches) and moving inwards towards the center, we fuse the outer links (already defined) of a tensor and assign the inner link to be the fusion of the outer links. We repeat this step towards the center node, so that the outer links are always well-defined (they possess a representation), by recursion.
**Option:** We can limit the dimensions of the inner links by intersecting the fusion of the outer links with a user-defined "maximal" link (for symmetric link intersection see Sec. 3.5.1). This option presents a possible bias to the random state, but it can make the approach numerically more efficient.

(iii) Once we reach the center node and all the links have been defined by the previous step, we start to clean up the links from the center outwards. First, we clean up all the virtual links connected to the center node. Then we move outwards into the branches until all virtual links have been cleaned up. At this point, no (physical) truncation has occurred yet.

(iv) Select randomly a virtual link whose total dimension is larger than $D$. Within this link, remove randomly a single symmetric link index. Then clean up all the other virtual links, starting from the closest ones in the distance and moving outwards, to update this change. No link is completely emptied by the clean-up procedure as long as $D > 0$. Repeat this step until all virtual links have dimension $D$ or less. This completes the virtual links setup.

(v) Finally, generate the tensors. Within each tensor, include all possible matches given the links, and assign a random degeneracy tensor to every match (see Sec. 3.5.1). Eventually, install a gauge.

Note that in the reduction step (iv), at every iteration we reduce a link dimension by one and then propagate the truncation: This is a conservative choice, which allows every state $|\Psi\rangle$ within the given bond dimensions $D$ to be generated. Nevertheless, quicker reduction strategies can be considered. Furthermore, during clean-up, propagation into a branch can be stopped as soon as the link leading outwards remains unchanged by the clean-up operation.

Now we have highlighted the main features of loop-free TNs, and identified the complete freedom in defining the tensors and links therein, exploited in the unitary, canonical and finally the symmetric gauge. In the next section we take advantage of those features and describe exemplarily a TN algorithm which searches ground states in a loop-free TN.

## 5  Ground State Search with Loop-Free Tensor Networks

In this section we discuss a robust scheme for achieving ground states (GS) of Hamiltonians by means of tensor networks with no loops (or cycles) in their geometry. This is a natural extension of the DMRG algorithm [39–41] adapted to any tensor network architecture without cycles [73, 74, 79, 181–183, 189–194]. The absence of cycles leads to several computational advantages: We have already encountered one of them, namely the TN gauge freedom, in the previous section and we will exploit it extensively in what follows. Another striking advantage of a loop-free geometry is the possibility to formulate local optimization problems (which are at the heart of efficient variational energy minimization procedures) as simple eigenvalue problems. This feature is not present in TNs with cycles, where often no exact strategies for the solution of local optimization problems are available, and one has to rely on approximate approaches (as occurring e.g. for MERA [66], PEPS [128, 195]). Moreover, loopy tensor networks often imply performing more expensive tensor contractions, and dealing with symmetries is less straightforward (as mentioned in Sec. 4.2.4): This is, for instance, the

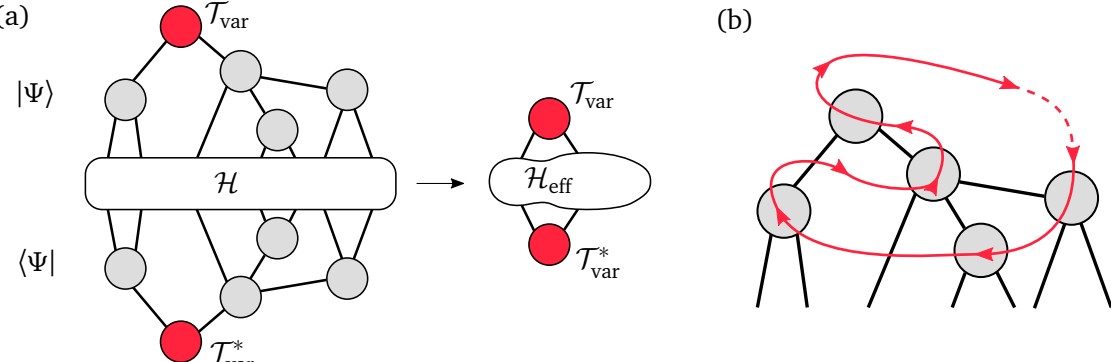

Figure 34: Basic idea of the variational ground state search: (a) In order to reduce the complexity of the minimization problem, only the parameters of one tensor $\mathcal{T}_{\text{var}}$ (red) are taken as variational while all other tensors are taken as fixed. The variational tensor forms the "optimization center", while the fixed tensors form an "environment". The latter can be efficiently contracted with the Hamiltonian $\mathcal{H}$, leading to a reduced "effective" Hamiltonian $\mathcal{H}_{\text{eff}}$. (b) By solving the reduced optimization problem sequentially (for varying locations of the optimization center), the energy expectation value is gradually reduced. This iterative minimization is commonly called "sweeping".

case for periodic boundary DMRG, where specific methods must be engineered to work around these issues [196, 197].

For these reasons, we choose to focus on ground state search algorithms for loop-free TN setups instead. This will break down the linear algebra demands to only three basic procedures (contraction, SVD/QR decomposition, simple eigenvalue problem) and take advantage of adaptive TN gauges [181, 182].

## 5.1 Ground State Search Algorithm for the Adaptive-Gauge Loop-Free TN

In this section we describe in detail the algorithm we adopt to achieve the QMB ground state variationally with loop-free TNs. The basic idea is to break down the diagonalization problem for the full QMB ansatz into a sequence of smaller, manageable diagonalization problems, each of which is formulated for a small subset of tensors (in the simplest case for one single tensor, as illustrated in Fig. 34(a)). Starting from a possibly random initial state in the manifold parametrized by the TN ansatz, we thus employ an iterative scheme of local tensor optimizations ("sweeping", see Fig. 34(b)) that converges to the state which best approximates the true QMB ground state. This approach is akin to what is known in applied mathematics as domain decomposition [198, 199].

Before listing the algorithm in Sec. 5.1.2, we first describe in detail the core concepts and high-level operations (based on the elementary ones defined in Sec. 2 and Sec. 3) that it will rely on. Note that our discussion applies both to symmetric and non-symmetric loop-free TNs, with the obvious requirement that in the symmetric case all tensors and tensor operations have to be formulated in a symmetry-aware fashion as detailed in Sec. 3.

### 5.1.1 High-Level Operations

In what follows, we consider a loop-free TN of tensors $\mathcal{T}^{[q]}$ sitting at network nodes $q = 1 \ldots Q$, mutually joined by virtual links $\eta$ (with dimension $D_\eta$). We can arbitrarily choose any node to be the center node: Here, we exclusively work in the unitary (center) gauge (see Sec. 4.2.2),

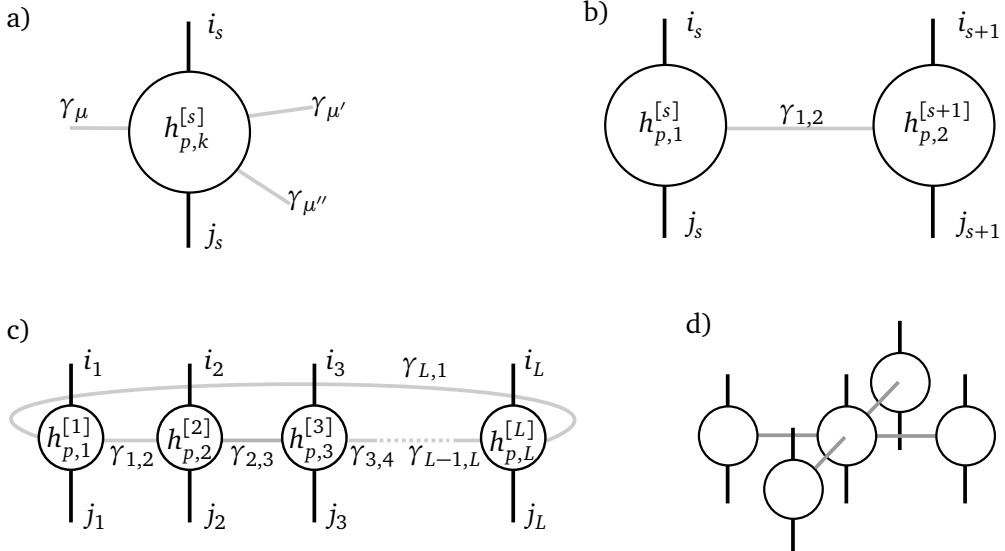

**Figure 35:** The TPO interaction terms $\mathcal{H}_p$ which sum up to the full Hamiltonian. (a) Each interaction $p$ is composed of local operators $h_{p,k}^{[s]}$ where $[s]$ stands for the TN physical link (or site) at which it is applied. The index $k$ enumerates the TPO operators from $k = 1 \dots K_p$, with $K_p$ being the TPO operator length (in general different from the interaction range), and $p$ distinguishes different interactions in the full Hamiltonian. Any two TPO operators $k$, $k'$ can be interconnected by a TPO link $\mu = (k, k')$, drawn in gray, but efficiency mandates to keep their overall dimension small. (b)-(d) Examples of interactions: (b) a nearest-neighbor product interaction (such as the Heisenberg Hamiltonian), (c) a global TPO and (d) a possible interaction of a two-dimensional model, where the center-tensor has however four TPO links.

and introduce the notion of an effective Hamiltonian.

**Hamiltonians.**     For the GS algorithm we consider Hamiltonians of the form

$$\mathcal{H} = \sum_{p=1}^{P} \mathcal{H}_p \, , \tag{67}$$

and we focus on Tensor Product Operator (TPO) [181] interactions of the form

$$\left[ \mathcal{H}_p \right]_{\{i_s\},\{j_s\}} = \sum_{\{\gamma_\mu\}} \prod_s \left[ h_{p,k(s)}^{[s]} \right]_{i_s,j_s}^{\{\gamma_\mu\}_{k(s)}} \, , \tag{68}$$

where $h_{p,k(s)}^{[s]}$ is the $k$-th local operator of the TPO acting on site $s$ and the operators are interconnected by links $\mu = (k, k')$, labeled by the two operators $k$ and $k'$ they connect. The sum runs over all TPO links $\{\mu\}$, and we define $\{\mu\}_{k(s)} \subseteq \{\mu\}$ as the set of all links $\mu$ attached to the $k$-th operator of the TPO. The outer product can be restricted to sites on which $\mathcal{H}_p$ acts non-trivially. TPOs can be viewed as a generalization of Matrix Product Operators (MPOs) [39, 56, 200], widely used in the context of MPS, to arbitrary TN geometries. Some examples for how a TPO $\mathcal{H}_p$ can look like are given in Fig. 35.

Computational efficiency is maintained by keeping the number and bond dimensions of TPO links small. Specifically, we consider interactions to be efficient if they are *compatible with*

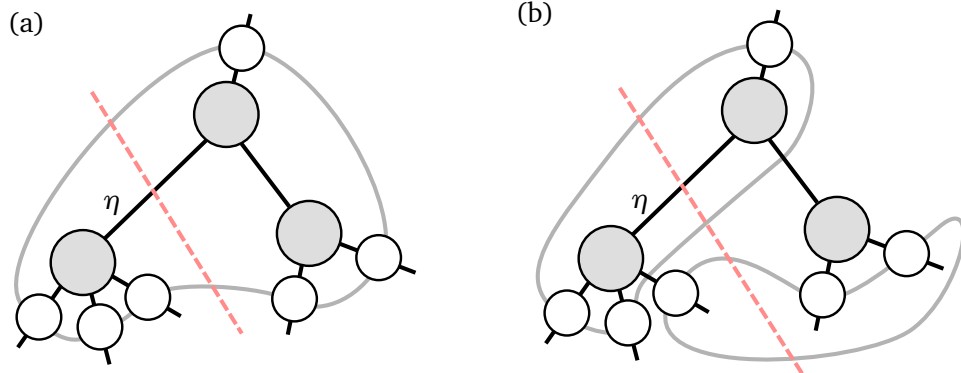

**Figure 36:** Two possible configurations for global TPO interactions (white tensors) acting on a TN state (gray tensors). Every local operator has two non-trivial TPO links: While the TPO configuration in (a) is considered efficient, in (b) there are four TPO links joining the network bipartition over the virtual link $\eta$ (indicated by the dashed red line).

*the TN geometry,* in that every network bipartition over any virtual link $\eta$ corresponds to a cut of two non-trivial TPO links (see Fig. 36). More generally, the condition to be met in order to keep computational efforts manageable is that the maximal total dimension $\prod D_\mu$ of links crossing any bipartition should be non-scaling (or at least weakly scaling) with the system size. Typical 1D lattice Hamiltonians fulfill this criterion, including sums of nearest-neighbor interactions, long-range two-site interactions, and TPOs in alignment with (i.e. having the same arrangement as) the physical sites, as exemplified in Fig. 35.

As the representation in Eqs. (67) and (68) is not unique, it has to be decided individually whether e.g. a large number $P$ of lightweight interactions (i.e. with small bond dimensions $D_\mu$) is more efficient than an equivalent representation with smaller $P$ but more or larger TPO links $\{\mu\}$. On the one hand, there are certain interaction types (e.g. exponentially decaying interactions) which are known to have a very efficient global TPO representation, characterized by a constant, non-scaling link dimension $D_\mu$ [39]. On the other hand, it has been noted that a single global TPO representation can be less efficient than treating the individual interaction terms separately [181], for example when interactions are long-range and inhomogeneous (e.g. in spin glass models or quantum chemistry applications). In particular, the algorithm can easily exploit the locality (i.e. the spatially restricted non-trivial action) of individual interaction terms, as shown in the next paragraph.

**Renormalization.** Once a Hamiltonian TPO has been provided, we assign each non-trivial local operator $h_{p,k}^{[s]}$ from every "bare" Hamiltonian term $\mathcal{H}_p$ to the physical link $s$ it acts on. We now associate with each *virtual* network link $\zeta$ a *renormalized* Hamiltonian, which describes the effective action of the physical Hamiltonian onto the subsystem (or bipartition part) $S_\zeta$ composed of all lattice sites separated over $\zeta$ from the current network center $c$. A single renormalized Hamiltonian operator takes the form $h_{p,\Bbbk}^{[\zeta]}$, where $\Bbbk$ is the set of all local operators of the TPO in the renormalization $\Bbbk(\zeta) := \cup_{s \in S_\zeta} k(s)$; in particular, $\Bbbk(s) = k(s)$ on physical sites.

The following recursive relation defines the renormalized Hamiltonian operators associated with the virtual link $\zeta$ that leads towards the current network center $c$ at some node $q$. It must be performed for each contribution $p$ separately: We start with collecting all (physical or renormalized) Hamiltonian terms on all remaining (physical or virtual) links $\{\xi\}_q$, $\xi \neq \zeta$, of $q$.

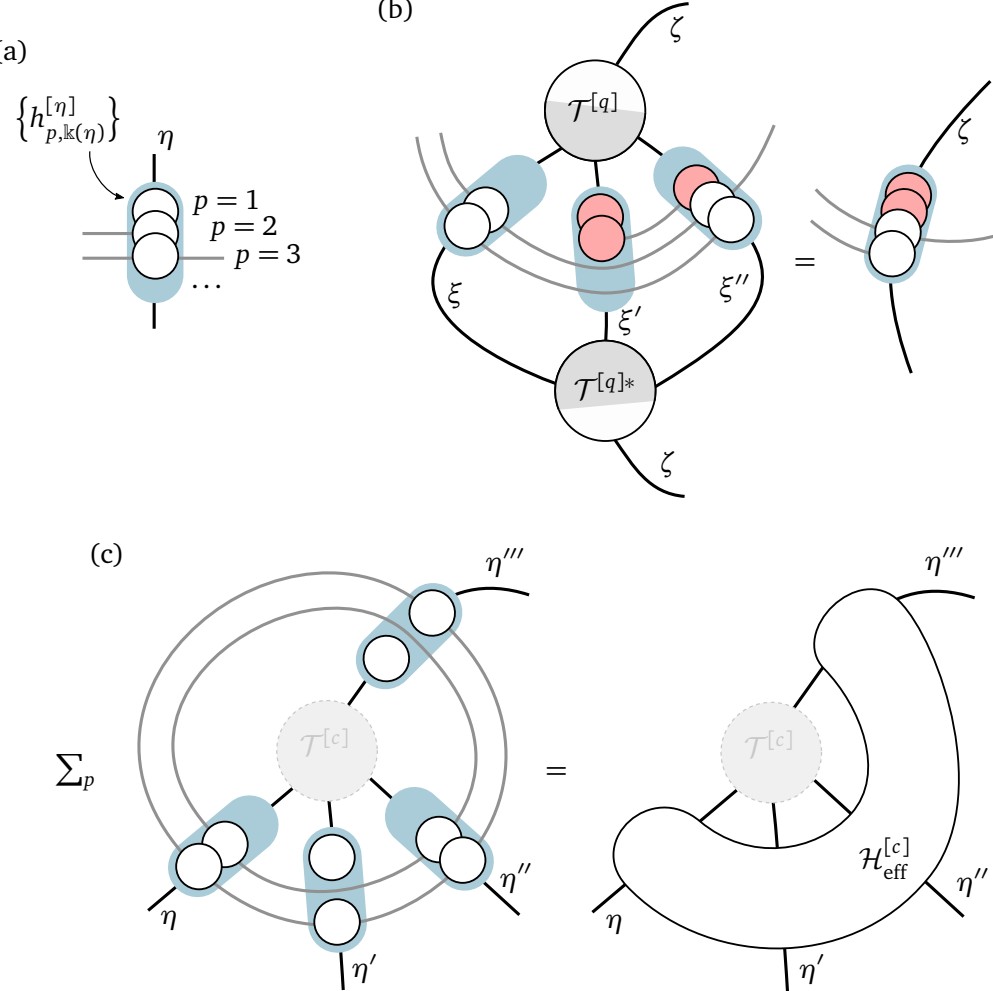

Figure 37: Building blocks of the GS algorithm: (a) Local interaction operators are defined on the (physical and virtual) network links. Tensors inside the blue shaded area on a link $\eta$ represent local operations $h^{[\eta]}_{p,\Bbbk(\eta)}$, each with a different $p$. Not for every $p$ in the Hamiltonian's sum Eq. (67) there must be a tensor defined at every link $\eta$; the missing ones are implicitly treated as identities. (b) Renormalizing the local interactions from the links $\xi, \xi', \xi''$ towards the virtual link $\zeta$. Each interaction $p$ is contracted independently into a renormalized term, and the global sum over $p$ is not carried out explicitly. An exception are renormalized operators $p$ that have no TPO links left (for example the two independent interactions indicated in red), which can already be summed into a single new renormalized tensor. (c) The effective Hamiltonian involves the contraction of all renormalized Hamiltonian terms over *all* tensor links $\{\eta\}_c$ at the center node $c$. Often, the final sum over $p$ is postponed, e.g. until contracting the individual terms $h^{[c]}_{p,\text{eff}}$ into the tensor $\mathcal{T}^{[c]}$ as described in Sec. 5.1.3 (solving the reduced optimization problem). The figure also shows the center tensor (semi-transparent) to highlight how it is acted upon by the effective Hamiltonian, which is a key ingredient for the GS algorithm.

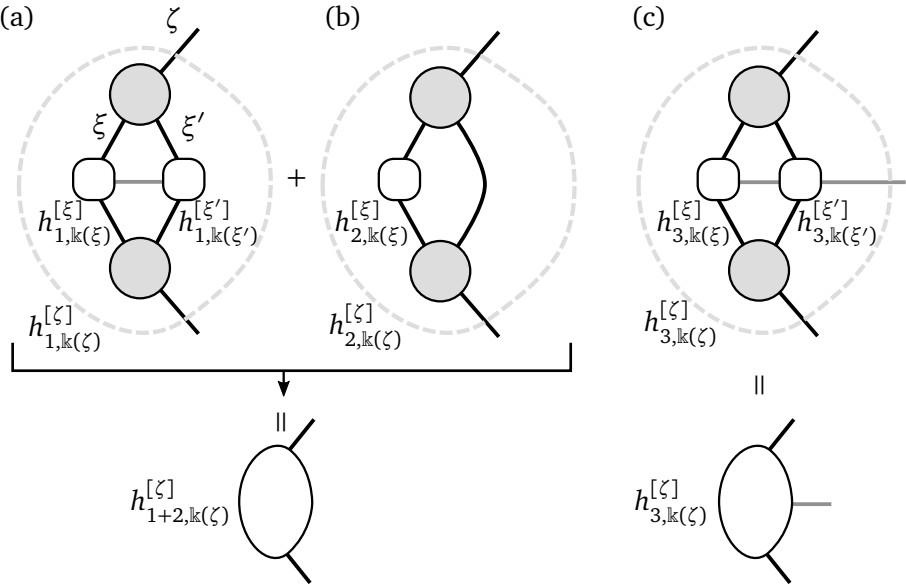

Figure 38: Example for summation over renormalized Hamiltonians without open TPO links: the resulting TPO tensors on link $\zeta$ (dashed) in (a) and (b) have no open TPO links, they can therefore be summed over, and saved in a single tensor. In (c), this is not the case.

They are generally joined by some mutual TPO-links $\{\mu\}$, representing the contraction

$$\left[h_{p,\Bbbk(\zeta)}\right]^{\{\gamma_\mu\}_{\Bbbk(\zeta)}}_{\{\alpha_\xi\},\{\beta_\xi\}} = \sum_{\{\gamma_\mu\}_{\text{paired}}} \prod_{\xi \neq \zeta} \left[h^{[\xi]}_{p,\Bbbk(\xi)}\right]^{\{\gamma_\mu\}_{\Bbbk(\xi)}}_{\alpha_\xi,\beta_\xi} . \tag{69}$$

The left-hand side ultimately combines $\Bbbk(\zeta) = \cup_{\xi \neq \zeta} \Bbbk(\xi)$ renormalized physical operators of the TPO. To obtain the renormalized Hamiltonian over link $\zeta$ we must additionally contract the Hamiltonian operators in Eq. (69) over the network links $\xi \neq \zeta$ with the node-tensor $\mathcal{T}^{[q]}$, as well as with its complex conjugate, as depicted in Fig. 37(b):

$$\left[h^{[\zeta]}_{p,\Bbbk(\zeta)}\right]^{\{\gamma_\mu\}_{\Bbbk(\zeta)}}_{\alpha_\zeta,\beta_\zeta} = \sum_{\{\alpha_\xi\}} \sum_{\{\beta_\xi\}} \mathcal{T}^{[q]}_{\alpha_\zeta,\{\alpha_\xi\}} \left[h_{p,\Bbbk(\zeta)}\right]^{\{\gamma_\mu\}_{\Bbbk(\zeta)}}_{\{\alpha_\xi\},\{\beta_\xi\}} \mathcal{T}^{[q]*}_{\{\beta_\xi\},\beta_\zeta} . \tag{70}$$

With small TPO-link dimensions, an efficient order in which to perform the contractions in Eqs. (69) and (70) is to generally first absorb the Hamiltonian operators by successively contracting them over all shared links into $\mathcal{T}^{[q]}$, and finally contract with the conjugate node tensor.

When working in the unitary gauge, the tensor $\mathcal{T}^{[q]}$ appears as an isometry over the links $\xi \neq \zeta$. Therefore, it is immediately clear that the result of Eq. (70) will be the identity whenever Eq. (69) produced an identity, i.e. whenever the TPO acts trivially on all links $\xi \neq \zeta$. This feature can save a lot of computational time on renormalizing interactions with spatially restricted (local) support. Another optimization which can be applied at this stage is to carry out the sum $\sum_p$ over all renormalized, *non-interacting* parts $h^{[\zeta]}_{p,\Bbbk_\zeta}$, i.e. those without any open TPO links $\{\mu\}$. All other renormalized interaction operators are stored in separate tensors on link $\zeta$ (see also Fig. 38).

Given that the bare Hamiltonians are *compatible* with the TN geometry, the renormalized Hamiltonians on the left-hand side of Eq. (70) have at most two non-trivial TPO links $\{\mu\}$ by construction, and all contractions remain efficient in $\mathcal{O}\left(D_\zeta^2 \prod D_\mu \prod D_\xi\right)$.

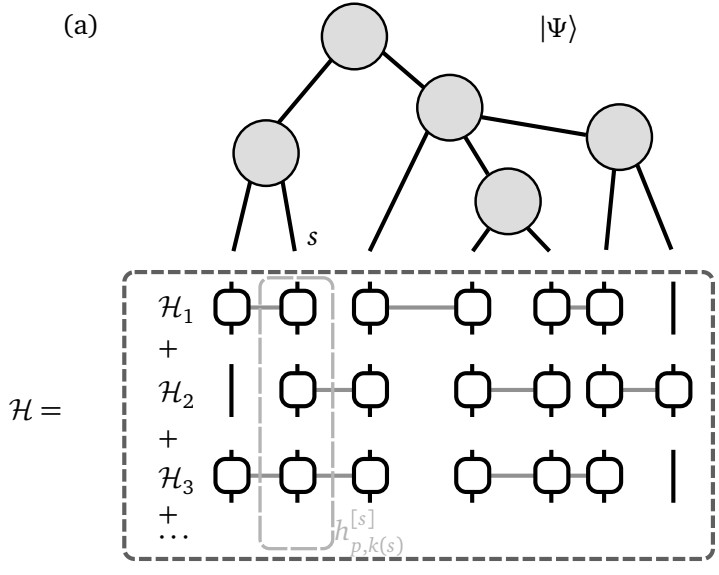

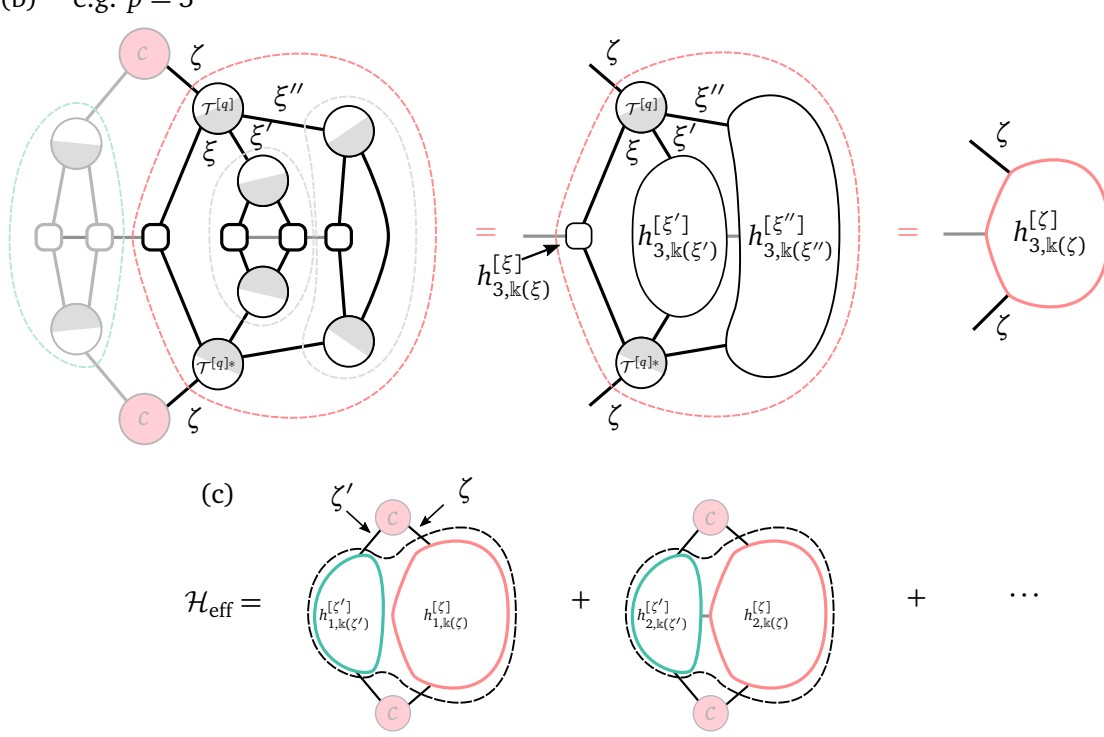

Figure 39: Example for the renormalization process that leads to the renormalized Hamiltonians: (a) The Hamiltonian $\mathcal{H}$ acting on the TN representation of some state $|\Psi\rangle$ can be decomposed into TPOs $\mathcal{H}_p$. (b) In a bottom-up approach, departing from the state-tensors closest to the bare Hamiltonian tensors, we can build the renormalized terms $h^{[\eta]}_{p,\Bbbk(\eta)}$ on all links $\eta$ below the chosen center $c$ by iteratively contracting state tensors with the renormalized Hamiltonian terms of the previous level (dashed circles). (c) We can then determine the effective Hamiltonian $\mathcal{H}_{\text{eff}}$ at the center node $c$ by summing over the renormalized Hamiltonian terms acting on the links of $c$.

**Effective Hamiltonian.** At any given center node $c$ (a unitary-gauge center), we define an effective Hamiltonian

$$\mathcal{H}_{\text{eff}}^{[c]} = \langle \Psi_{\text{env}} | \mathcal{H} | \Psi_{\text{env}} \rangle \,, \tag{71}$$

where "env" stands for the TN contraction over all tensors except $\mathcal{T}^{[c]}$, i.e. the environment of the center node. With this definition, the energy expectation value $E$ of the TN state can be written as (cf. Fig. 34(a))

$$E = \langle \Psi | \mathcal{H} | \Psi \rangle = \langle \mathcal{T}^{[c]} | \langle \Psi_{\text{env}} | \mathcal{H} | \Psi_{\text{env}} \rangle | \mathcal{T}^{[c]} \rangle = \langle \mathcal{T}^{[c]} | \mathcal{H}_{\text{eff}}^{[c]} | \mathcal{T}^{[c]} \rangle \,. \tag{72}$$

In practice, to obtain $\mathcal{H}_{\text{eff}}^{[c]}$, we do not need to contract the entire environment, if we have already the renormalized Hamiltonians (as defined in the previous paragraph) on all links of node $c$ available. Then, Eq. (71) reduces to effective Hamiltonian terms (see Fig. 37(c))

$$\left[ h_{p,\text{eff}}^{[c]} \right]_{\{\alpha_\eta\},\{\beta_\eta\}} = \sum_{\{\gamma_\mu\}} \prod_\eta \left[ h_{p,\Bbbk(\eta)}^{[\eta]} \right]_{\alpha_\eta,\beta_\eta}^{\{\gamma_\mu\}_{\Bbbk(\eta)}} \,, \tag{73}$$

defined over all tensor links $\{\eta\}$ at node $c$ with corresponding indices $\{\alpha_\eta\}$, $\{\beta_\eta\}$. The contraction sum runs over all TPO links $\{\mu\}$ with corresponding indices $\{\gamma_\mu\}$. To complete the effective Hamiltonian, we have to sum over all interactions $p$:

$$\mathcal{H}_{\text{eff}}^{[c]} = \sum_p h_{p,\text{eff}}^{[c]}. \tag{74}$$

The whole process of building the effective Hamiltonian is summarized in Fig. 39.

**Projectors.** Adding projective operators to the Hamiltonian opens the possibility to explore degenerate GS manifolds and to target excited states (we will comment further on this paradigm in Sec. 5.1.5 below). More precisely, we apply the ground state search algorithm to the Hamiltonian $\mathcal{H}' = \mathcal{H} + \mathcal{H}_p$, where $\mathcal{H}_p = \varepsilon_p |\Psi_p\rangle\langle\Psi_p|$, to find the first excited state of $\mathcal{H}$, assuming its ground state $|\Psi_p\rangle$ is known, and where $\varepsilon_p$ is much larger than the energy gap. Here we discuss the explicit construction of projective summands $\mathcal{H}_p = \varepsilon_p |\Psi_p\rangle\langle\Psi_p|$, where $|\Psi_p\rangle$ is expressed as a TN with the same geometry as the variational TN ansatz state. The scaling factor $\varepsilon_p$ will play the role of an energy penalty for occupying the state $|\Psi_p\rangle$.

The projectors are thus defined via the (not necessarily isometric) tensors $\mathcal{T}_p^{[q]}$ with possibly different dimensions in the virtual links (and symmetry sectors if they are symmetric; but the global symmetry group has to be the same). Projective operators can be implemented within the same structure and almost the same equations as for the renormalized and effective Hamiltonians. For a graphical representation, see Fig. 40. In the recursion relation of Eq. (70), we replace $\mathcal{T}^*$ with $\mathcal{T}_p^*$, and set identities for $h_p^{[s]}$ on physical sites:

$$\left[ h_p^{[\zeta]} \right]_{\alpha_\zeta,\beta_\zeta} = \sum_{\{\alpha_\xi\}} \sum_{\{\beta_\xi\}} \mathcal{T}_{\alpha_\zeta,\{\alpha_\xi\}}^{[q]} \left( \prod_\xi \left[ h_p^{[\xi]} \right]_{\alpha_\xi,\beta_\xi} \right) \mathcal{T}_{p,\{\beta_\xi\},\beta_\zeta}^{[q]*} \,. \tag{75}$$

Note that here we dropped the internal TPO label $k$ because the $h_p$ are defined on every link without TPO links. It is important to keep in mind, however, that the projective terms $\mathcal{H}_p$ have *global* support. Therefore here we can never carry out the sum $\sum_p$, in contrast to renormalized TPO interactions with *local* support which have no TPO links left. Furthermore, isometry can no longer be exploited here, and the resulting renormalized projectors $h_p^{[\zeta]}$ will in general be different from the identity.

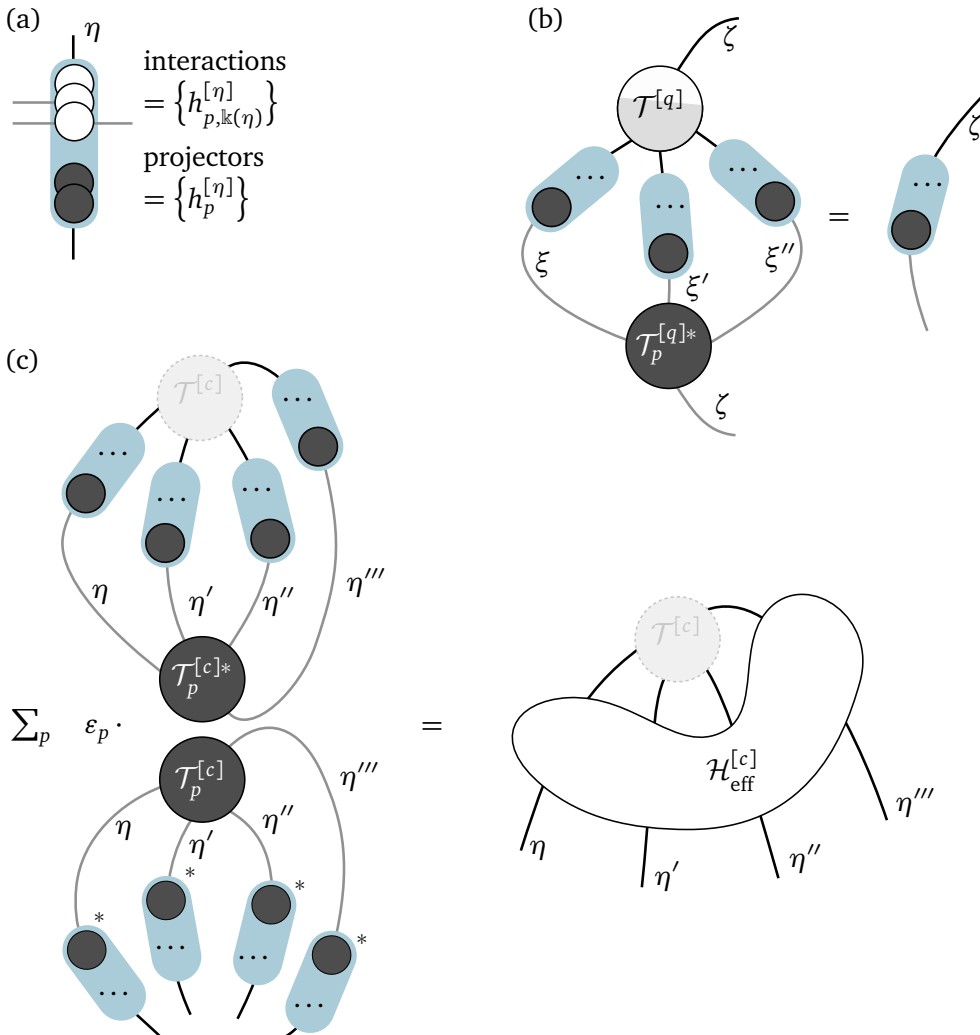

Figure 40: Adding projectors to the Hamiltonian: a) (Renormalized) local projection operators are put on the links. Contrary to interactions which act non-trivially on physical links, projective terms are always identities on physical links. b) Similarly to the renormalization of TPO interactions, projectors are renormalized to a virtual link $\zeta$ of a node $q$ by contracting over all local projection operators on the remaining node links $\eta$, $\eta'$, $\eta''$, with the conjugate tensor $\mathcal{T}_p^{[q]*}$ now taken from the projector TN. The renormalized projectors $h_p^{[\zeta]}$ have no TPO links, but in contrast to the rule for TPO interactions they cannot be summed up over $p$ (every projector results in a separate renormalized two-link tensor on each link). c) The effective projector at the center $c$ is given by the contraction of the renormalized projectors over all links with the conjugated projector-tensor $\mathcal{T}_p^{[c]*}$. The resulting expression is then multiplied by its complex conjugate. This contraction has to be repeated for every $p$ in the global Hamiltonian sum individually, constituting one summand of the complete effective Hamiltonian $\mathcal{H}_{\text{eff}}^{[c]}$. As before, we show the center tensor (semi-transparent) to demonstrate how the effective projector acts upon it.

Figure 41: Example for the renormalization process that leads to the renormalized projection operators: (a) Renormalized projector terms $h_p^{[\eta]}$ are built on all links $\eta$ below the center node $c$ by iteratively contracting state tensors with the renormalized projector terms of the previous level (dashed circles). (b) For a given center $c$, we can then determine the projector contributions to the effective Hamiltonian $\mathcal{H}_{\text{eff}}$ by summing over the renormalized projector terms acting on the links of $c$.

The effective projector at a node $c$ becomes $\varepsilon_p \langle\Psi_{\text{env}}|\Psi_p\rangle \langle\Psi_p|\Psi_{\text{env}}\rangle$, which is analogous to the definition of the effective Hamiltonian. Written out explicitly it reads

$$\left[h^{[c]}_{p,\text{eff}}\right]_{\{\alpha_\eta\},\{\beta_\eta\}} = \varepsilon_p \, \tilde{\mathcal{T}}^{[c]}_{p,\{\alpha_\eta\}} \cdot \tilde{\mathcal{T}}^{[c]*}_{p,\{\beta_\eta\}}, \tag{76}$$

where

$$\tilde{\mathcal{T}}^{[c]}_{p,\{\alpha_\eta\}} = \sum_{\{\beta_\eta\}} \left(\prod_\eta \left[h^{[\eta]}_p\right]_{\alpha_\eta,\beta_\eta}\right) \mathcal{T}^{[c]*}_{p,\{\beta_\eta\}}. \tag{77}$$

Note that this lengthy expression is rather simple to compute, by contracting all renormalized projectors into $\mathcal{T}^{[c]*}_p$, and multiplying with the complex conjugate of the outcome. The resulting links constitute the bra- and ket- links of the effective projector $h^{[c]}_{p,\text{eff}}$, see Fig. 40(c). The whole process of building effective projectors is summarized in Fig. 41.

Some specific applications, such as variational addition of TNs in Sec. 5.1.6, employ projective operators $|\Psi_p\rangle\langle\Psi_p|$ onto *superpositions* of TN states. Those are of the form $|\Psi_p\rangle = \sum_i \varepsilon_{p,i}|\Psi_{p,i}\rangle$ with all $|\Psi_{p,i}\rangle$ given as separate TNs sharing the same geometry. In such a case, one computes renormalized projectors $h^{[\zeta]}_{p,i}$ for each state in the superposition separately — just as before, by simply replacing the single projector index $p$ with $p,i$ in Eq. (75). A more relevant adjustment is to be made in the effective Hamiltonian, which now reads $\sum_{i,j} \varepsilon_{p,i}\varepsilon^*_{p,j} \langle\Psi_{\text{env}}|\Psi_{p,i}\rangle \langle\Psi_{p,j}|\Psi_{\text{env}}\rangle$. Correspondingly, as a replacement for the single factor $\varepsilon_p$ in Eq. (76), one additionally has to sum over superposition states in Eq. (77):

$$\tilde{\mathcal{T}}^{[c]}_{p,\{\alpha_\eta\}} = \sum_i \varepsilon^*_{p,i} \sum_{\{\beta_\eta\}} \left(\prod_\eta \left[h^{[\eta]}_{p,i}\right]_{\alpha_\eta,\beta_\eta}\right) \mathcal{T}^{[c]*}_{p,i,\{\beta_\eta\}}. \tag{78}$$

Nevertheless, evaluating such an effective Hamiltonian remains a simple computation by successive absorption of all effective projectors into $\mathcal{T}^{[c]*}_{p,i}$.

### 5.1.2 Ground State Search Algorithm in Detail

We now describe the algorithm, which is based on the high-level operations introduced in the previous section, and follows the flowchart shown in Fig. 42. In the following, we list additional explanations and remarks, useful when implementing such an algorithm.

**Setup and Configuration Tasks.**

(i) Definition of the TN geometry: From the physical point of view, the TN should be designed such that it complies best with the system and the Hamiltonian being targeted. On the one hand, the TN has to be compatible with the interaction terms in the Hamiltonian. On the other hand, it has to be capable of hosting the required amount and spatial distribution of entanglement, controlled by its virtual links. Finally, the entanglement arises from the interactions and (at least for ground states) can often be theoretically proven to be upper-bounded (e.g. by area laws [37]). Fig. 43 summarizes schematically these requirements. At the purely technical level, any network fulfilling the following requirements is allowed:

- The network has $N$ physical links, supporting the system Hamiltonian $\mathcal{H}$ (defined in Eq. (67)).
- The network is acyclic: for any pair of nodes $q$, $q'$ there exists a unique path connecting $q$ and $q'$.

Initialization

- define network geometry

- define sweep sequence

- initialize TN state

Target 1st opt. center

- install unitary gauge

- renormalize physical Hamiltonian
  to effective Hamiltonian

see Eq. (73) and Fig. 37

Optimization loop

$s \mapsto s+1$

- increment sweep counter

see Sec. 5.1.3

- optimize center node

$n$   $n+1$

- target next optimization center

  - adjust gauge

see Fig. 44

  - adjust eff. Hamiltonian

sweep sequence end?    no

yes

energy converged?    no

yes

Finalization

- measure observables

- compute (bi)partitions entanglement

A   B

- ...

Figure 42: Flowchart of the GS algorithm. The left column provides illustrations/references for the corresponding items in the bullet list.

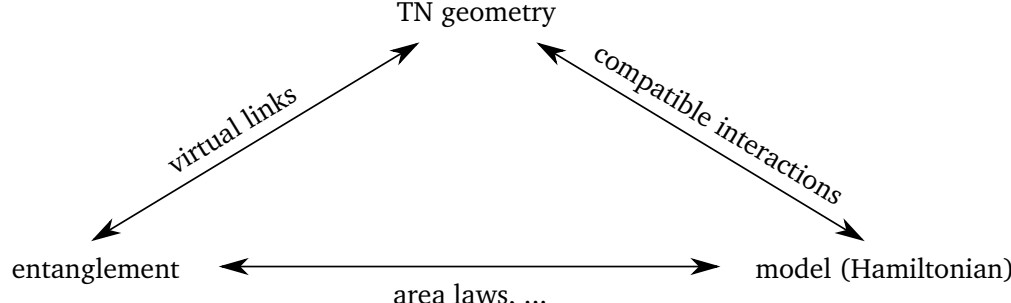

Figure 43: Elements influencing the choice of the TN geometry: The network has to be compatible with the interactions present in the Hamiltonian, and at the same time must be able to capture the ground state entanglement, which in turn is determined by the Hamiltonian itself.

- The dimension of any virtual link in the network does not exceed $D$ (bond dimension of the TN).

- While in principle it is not prohibited, the employment of tensors with less than three links is redundant and may deadlock symmetries. Absorbing one-link and two-link tensors into adjacent tensors is strongly suggested, as it leaves the variational TN state manifold unchanged while reducing the technical issues with symmetries (see Sec. 5.1.3).

(ii) Definition of *sweep sequence*: This is the rule according to which the optimization centers are chosen sequentially in the optimization loop. In principle, any sequence that visits every tensor of the TN (and every variational link, in case of multi-tensor optimization) is allowed. The specific form of the optimal sweep sequence, i.e. the one that achieves fastest energy convergence within the shortest computation time, depends on the TN structure and the model under consideration. For regular hierarchical networks (e.g. binary TTNs, see Sec. 5.2) it is natural to select a sweep sequence based on the distance from the physical links: Assign an optimization precedence $p := \min_s [\text{dist}(c,s)]$ to every optimization center $c$, where $s$ is a physical link. Within a sweep sequence, optimization centers with smaller $p$ take precedence over centers with larger $p$. Centers sharing the same $p$ are chosen in such a way that the distance that has to be covered in order to visit all of them is minimal. For less regular networks (e.g. TNs for quantum chemistry problems) other schemes (such as depth-first search with backtracking) have been proposed [181].

(iii) Initialization of TN state: An obvious choice is to start from a random state (see Sec. 4.3.1 on how to construct such a random state for a symmetric TN with a defined global quantum number). This choice guarantees the least possible bias and therefore bears the smallest risk of ending up in a local energy minimum state, which would corrupt the convergence to the GS (by definition the global energy minimum state). Another valid option is to start from an already computed GS, previously obtained with the same algorithm on another Hamiltonian, assuming the symmetries and quantum numbers are the same.

(iv) Targeting of initial optimization center $c$:

- Transform the network into unitary gauge with respect to $c$ (see Sec. 4.2.2).

- Build up the effective Hamiltonian for $c$, according to the procedure outlined in Sec. 5.1.1.

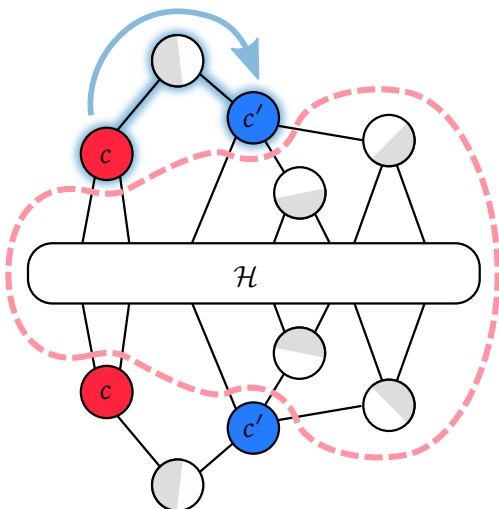

Figure 44: Moving the effective Hamiltonian from an optimization center $c$ to another optimization center $c'$. Only the renormalized Hamiltonian terms (Eq. (70)) defined on the links located on the path from $c$ to $c'$ have to be updated (blue shaded). All other renormalized Hamiltonian terms (contractions within the red-dashed area) are left unchanged by this move and therefore do not need to be recomputed.

In case of multi-tensor optimization, we extend the notion of the network center to two or more network nodes. This can be achieved by temporarily contracting some adjacent nodes $\{q\}$ into a single center node $c$. The renormalized Hamiltonian terms and isometry properties have to be computed as usual for all remaining links and tensors, respectively. After the optimization center has been prepared, the effective Hamiltonian is given by Eq. (73). The energy expectation value $E$ of the TN state is then (see Eq. (72))

$$E = \langle \mathcal{T}^{[c]} | \mathcal{H}_{\text{eff}}^{[c]} | \mathcal{T}^{[c]} \rangle \,, \tag{79}$$

where $\mathcal{T}^{[c]}$ is the tensor of the center node. Now, the generic local (reduced) optimization problem consists in determining the variational tensor $\mathcal{T}^{[c]}$ in such a way that it minimizes $E$. This optimization problem is solved sequentially for all centers $c$ in the following optimization loop.

**Optimization Loop.**

(i) Optimize $\mathcal{T}^{[c]}$, i.e. minimize Eq. (79). The optimal $\mathcal{T}^{[c]}$ is given by the eigenvector of $\mathcal{H}_{\text{eff}}^{[c]}$ corresponding to the lowest eigenvalue (see a proof in Ref. [89]). While this basic observation remains always the same, there are technical differences for different types of optimization centers (e.g. single-tensor, double-tensor, etc.). Sec. 5.1.3 is dedicated to the different flavors of the "optimizer routine".

(ii) Target the next optimization center $c'$ according to the sweep sequence by moving the isometry center to $c'$ via gauge transformations and updating the renormalized Hamiltonian terms between $c$ and $c'$ (see Fig. 44).

(iii) Loop back to step (i) and repeat the optimization with the new center node $c'$, until the end of the sweep sequence is reached.

(iv) Check convergence. If not converged, start a new sweep. Typical convergence criteria are: change in energy expectation value, variance of energy, distance of TN states from

subsequent sweeps. Most commonly, one monitors the energy expectation value $E(s)$ after sweep $s$: Since $E(s)$ is expected to approach the GS energy asymptotically from above, a possible convergence criterion is to demand

$$E(s-1) - E(s) < \epsilon \,, \tag{80}$$

where $\epsilon$ is a constant (positive) threshold. In practice $\epsilon$ has to be adapted according to the bond dimension $D$ in order to avoid unnecessary sweeps that would push the convergence too far beyond the accuracy limit imposed by finite bond dimension. Usually, the bond dimension dependence of this accuracy limit (and hence the form of $\epsilon(D)$) is well modeled by some polynomial or exponential decay (or combination thereof), e.g. one might choose the ansatz $\epsilon(D) = a + b \, D^{-c} \exp(-D/d)$. In practice, the parameters $a$, $b$, $c$, $d$ need to be determined empirically by some fitting procedure because the form of the decay highly depends on the model parameters (system size $N$, proximity to a critical point, etc.).

### 5.1.3 Optimizer – Strategies

Here we describe in detail various possible strategies when interfacing with the reduced optimization problem (while how to practically perform the numerical optimization routine will be discussed in Sec. 5.1.4). The most commonly employed schemes for this purpose are the *single-tensor update* and the *double-tensor update*, where the optimization centers consist of one tensor and two tensors, respectively. We additionally introduce a *single-tensor update with subspace expansion*.

The three schemes differ in the complexity of the variational problem they solve, and have different requirements on computational resources. The first two schemes, single- and double-tensor update, are a generalization of DMRG to arbitrary loop-free TN architectures. They are directly related to the traditional single-center site and double-center site DMRG [40] respectively. Finally, the subspace expansion scheme allows us to reconcile the improved convergence behavior of the double-tensor update with the favorable computational cost scaling of the single-tensor update.

In the single-tensor update, the optimization center consists of just a single network tensor, and it is the most straightforward and computationally inexpensive scheme. However, it can experience serious shortcomings: If the Hamiltonian under consideration exhibits a certain symmetry, the same symmetry will usually be present in the effective Hamiltonian of the reduced diagonalization problem. As a consequence, local optimizations are constrained to preserve the link representations on the individual links of the optimization center. The present quantum numbers (sectors) and their associated degeneracies can therefore not be handled variationally. This issue is particularly acute for symmetries that are encoded in the TN ansatz (e.g. by the methods described in detail in Sec. 3): in that case the mentioned constraints are hard and will be *exactly* (by construction of the tensors) respected in each local diagonalization. For TNs made from tensors that are unaware of symmetry rules the constraints are softer: In this case they are only metastable configurations which can be violated through numerical fluctuations, caused e.g. by finite precision floating point operations or by the fact that the environment (i.e. the tensors surrounding the optimization center) does not obey the symmetry. Nevertheless, the existence of these metastabilities still often leads to poor convergence behavior (as observed for example in the standard single-site DMRG algorithm [201]).

The standard way to cure this issue is to formulate the diagonalization problem not for a single tensor, but for several (usually two) adjacent tensors that previously have been contracted together. After optimizing this compound tensor, the original tensor structure is restored by means of SVDs. The accompanying truncation of the state spaces associated with the

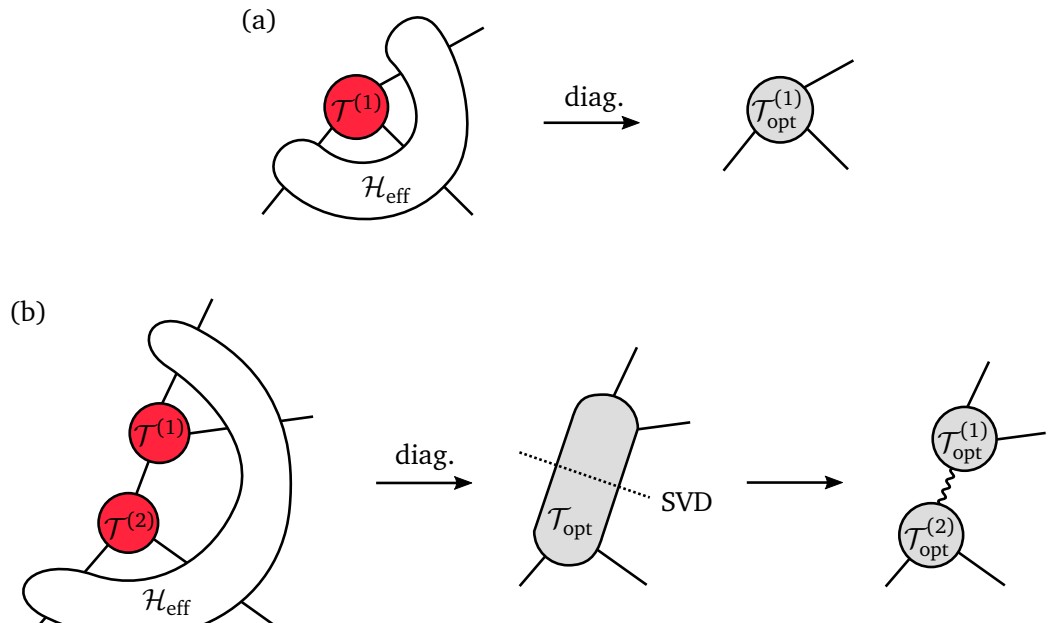

Figure 45: (a) Single-tensor update: the reduced diagonalization problem is formulated for one tensor. The optimized tensor is given by the eigenvector of $\mathcal{H}_{\text{eff}}$ with the smallest eigenvalue. Note that link representations are preserved if $\mathcal{H}_{\text{eff}}$ obeys some symmetry. (b) Double-tensor update: the reduced diagonalization problem is formulated for two tensors. The number of links of the optimized tensor is given by the number of links on which $\mathcal{H}_{\text{eff}}$ acts. Link representations of these links are again preserved, in contrast to the one of the link which joins $\mathcal{T}^{(1)}$ and $\mathcal{T}^{(2)}$: This link results from an SVD and can differ from the original link, if the SVD performs a non-trivial separation of the links of $\mathcal{T}_{\text{opt}}$. Note that if one of the two tensors is a two-link tensor, this link separation is trivial and the link created by the SVD will be the same as the external link of the two-link tensor.

compound's internal links provides the variational handle lacking for external links, as illustrated in Fig. 45. The downside of the multi-tensor optimization strategy is its higher computational cost resulting from the increased number of links of the involved tensors.

In order to avoid the increase in computational costs, we can employ a so-called *subspace expansion* at a single tensor $\mathcal{T}^{(1)}$, by temporarily expanding the state space of one selected virtual link $\eta_o$ of $\mathcal{T}^{(1)}$ in order to allow for a variation of the symmetry representation on that link. While the same ideas apply to both ordinary and symmetric TNs, we now describe the technique explicitly for symmetric tensors since some extra steps are required for this case. As a first step, an expansion of the selected link's state space can be achieved in the following way: Start by fusing all the links of $\mathcal{T}^{(1)}$ other than $\eta_o$ to one large link $\eta_1$. Repeat the operation for tensor $\mathcal{T}^{(2)}$, which is connected to $\mathcal{T}^{(1)}$ through $\eta_o$, to obtain another large link $\eta_2$. Now determine the intersection $\eta^{\text{max}} := \eta_1 \cap \eta_2$ (see Sec. 3.5.1). Note that the resulting $\eta^{\text{max}}$ has the full link representation accessible to the double-tensor optimization scheme, i.e. it can capture any link resulting from an SVD of the optimized double-tensor. Since our goal is to reduce the computational cost with respect to the double-tensor optimization, we introduce a parameter $d_{\text{pad}}$ to truncate all degeneracies $\bar{\partial}_{\ell_{\eta^{\text{max}}}} = d_{\text{pad}}$ (for all sectors $\ell_{\eta^{max}}$ in $\eta^{\text{max}}$). The resulting link $\eta_{\text{pad}}^{\text{max}}$ is used to pad (see Sec. 3.5.1) the link $\eta_o$, according to $\eta_o' := (\eta_o + \eta_{\text{pad}}^{\text{max}}) \cap \eta^{\text{max}}$ (the additional intersection makes sure that no redundant degeneracies are generated by the padding operation). The parameter $d_{\text{pad}}$ now allows to smoothly interpolate between a single-

tensor update and a double-tensor update. The latter limiting case can be achieved by choosing $d_{\text{pad}}$ sufficiently large such that $\eta'_o = \eta^{\max}$, while the former limiting case is obtained by setting $d_{\text{pad}} = 0$ which results in $\eta'_o = \eta_o$. Note that in case the link $\eta_o$ exceeds the minimal bond dimension determined by the Schmidt rank (see Sec. 4.2.1), that is to say, in case there exist some sectors $\ell$ for which $\bar{\partial}_{\ell_{\eta_o}} > \bar{\partial}_{\ell_{\eta^{\max}}}$, the procedure to obtain $\eta'_o$ may also cause some sectors to have reduced degeneracies, effectively removing redundant indices from $\eta_o$.

The actual optimization within the single-tensor update with subspace expansion is carried out in the following way (see Fig. 46): First the link $\eta_o$ at the tensors $\mathcal{T}^{(1)}$ and $\mathcal{T}^{(2)}$ is replaced by the expanded link $\eta'_o$. The hereby newly created tensor elements in the two tensors are filled (via subtensor assignment, see Sec. 3.5.1) with random entries (this step corresponds to the subspace expansion). Then, the tensor $\mathcal{T}^{(1)}$ is made the center node of the TN and it is optimized as usual (by diagonalizing $\mathcal{H}_{\text{eff}}$). Since the subspace expansion also affects $\mathcal{T}^{(2)}$, this tensor has to be optimized as well (analogously to $\mathcal{T}^{(1)}$), keeping the same expanded link $\eta'_o$. The two alternating optimizations of $\mathcal{T}^{(1)}$ and $\mathcal{T}^{(2)}$ are repeated until convergence is achieved (usually one or two repetitions are enough). Finally, the original bond dimension $D$ is restored via an SVD, discarding the smallest singular values (see Fig. 46). We will show and discuss numerical results obtained with this method in Sec. 5.3. A diagrammatic comparison of the various optimization strategies is presented in Fig. 46.

Subspace expansion techniques in single- and double tensor updates can become an even more powerful tool when combined with perturbative approaches, such as the procedure introduced in Refs. [143, 201] for curing the shortcomings of the traditional single-center site DMRG. These approaches typically differ from the single-tensor update with subspace expansion presented above in the following three aspects: First, the padded elements are generated from a contraction of the effective Hamiltonian (or a component thereof) into the center tensor, such as to introduce first-order contributions to $\mathcal{H}|\Psi\rangle$ ("enrichment step"). Second, the perturbation coupling (in the simplest approach, a scalar factor) has to be driven to zero in a controlled way to achieve convergence of the algorithm. Third, after some form of local bond truncation, the perturbation introduced by the padded elements persists within the network until the enriched node is revisited in a later sweep. This strategy enables the subspace expansion to propagate through the network, making it potentially more powerful than the double-tensor update discussed above without such perturbative approach [143].

### 5.1.4 Optimizer – Execution

A crucial step of the optimization routine (common to all update schemes) is the diagonalization of the effective Hamiltonian $\mathcal{H}_{\text{eff}}$ as this step usually constitutes the computational bottleneck of the whole GS algorithm. Thanks to the central gauging we are merely dealing with a standard eigenvalue problem (as opposed to a generalized eigenvalue problem arising in TNs with loops in their geometry). While this already provides a computational advantage, the identification of a fast eigensolver is still of utmost importance. Two observations are helpful in this context: $\mathcal{H}_{\text{eff}}$ is usually sparse[2], and only the ground-eigenstate of $\mathcal{H}_{\text{eff}}$ has to be computed. Therefore, direct diagonalization algorithms (such as tridiagonal reduction, implemented by e.g. LAPACK's *zheev* [136]) are not convenient since they compute the whole spectrum, for which they usually require the explicit construction of the full matrix $\mathcal{H}_{\text{eff}}$. It is instead advantageous to employ iterative algorithms based on power methods, e.g. Lanczos/Arnoldi iteration [157, 159]. These algorithms exploit the sparsity of $\mathcal{H}_{\text{eff}}$, only require knowledge of the action $\mathcal{H}_{\text{eff}}|\mathcal{T}^{[c]}\rangle$ on the center node tensor $|\mathcal{T}^{[c]}\rangle$ (see Fig. 37(c)) and, in addition, are specialized to return only

---

[2]More precisely, the two conditions that $\mathcal{H}_{\text{eff}}$ has to fulfill in order to be sparse are: (i) The dimensions of the TPO links $\{\mu\}$ should be much smaller than the bond dimensions of the virtual links ($D_\mu \ll D_\eta$), and (ii) the total number of interactions $p$ should be small (see Eq. (73)). Many important Hamiltonian classes (e.g. short-range interacting models) meet these requirements.

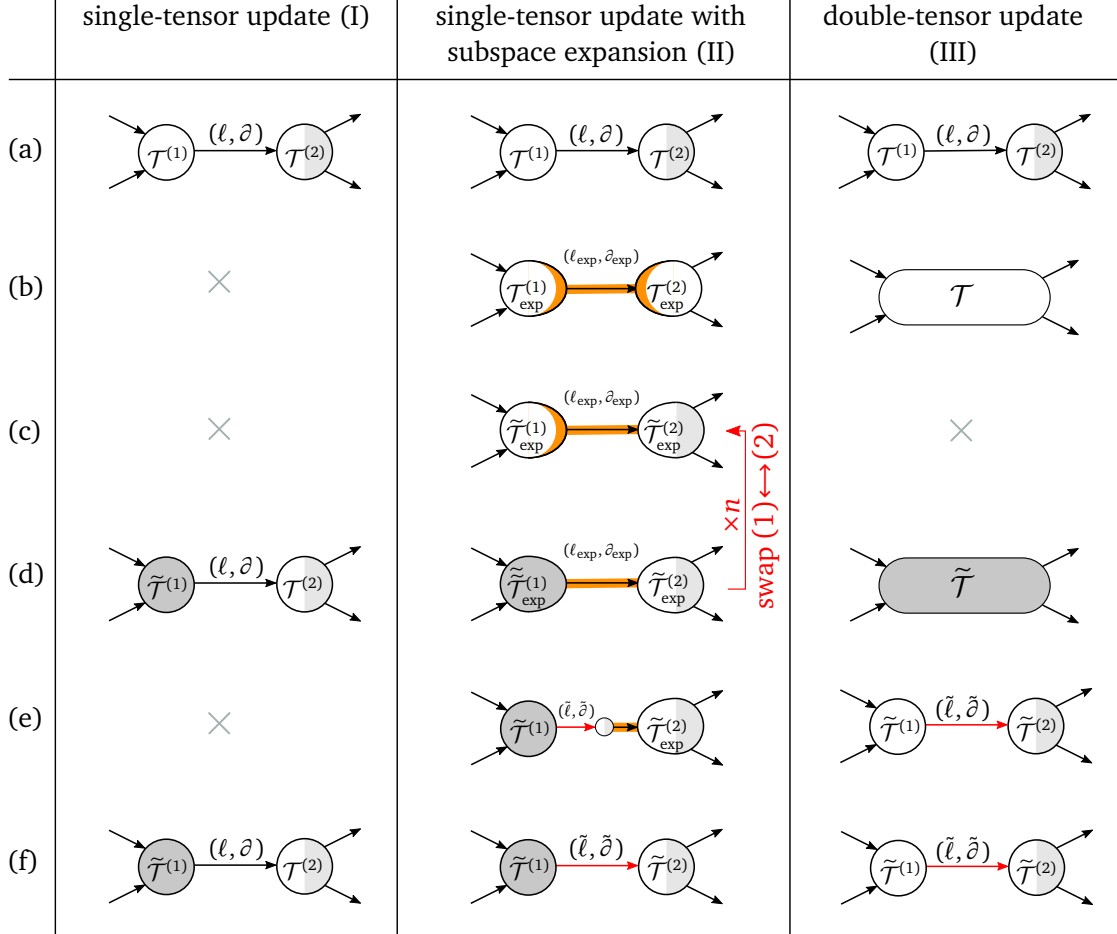

Figure 46: Comparison of different update schemes. (a) Initial configuration (identical in all three schemes). Note that scheme I only operates on one tensor, which means $\mathcal{T}^{(2)}$ is left unchanged throughout this scheme. (b) Making the link representation of the intermediate link variational: scheme I does not have this step; scheme II realizes it via link expansion (see text), while scheme III achieves it via contraction with $\mathcal{T}^{(2)}$. (c) Restore gauge and compute effective Hamiltonian for $\mathcal{T}^{(1)}$ (only scheme II). (d) Update center node with lowest eigenvector of $\mathcal{H}_{\text{eff}}$. Only scheme II: repeat (c) and (d) until converged, the center node being alternately $\mathcal{T}^{(1)}$ and $\mathcal{T}^{(2)}$. (e) Restoring the original bond dimension of the intermediate link: scheme I does not have this step; scheme II and III realize it via an SVD, discarding the smallest singular values. (f) Final configuration. Note that in scheme II and III the link representation of the intermediate link may have changed, while in scheme I it is constant.

a few eigenpairs. The computational costs of the different update schemes are thus given by the numerical effort of evaluating $\mathcal{H}_{\mathrm{eff}}|\mathcal{T}^{[c]}\rangle$: All schemes share a factor $f = N_{\mathrm{app}} P D_{\mathrm{TPO}}^3$, where $N_{\mathrm{app}}$ is the number of operator applications in the iterative eigensolver (see Sec. 5.2 for an analysis of the behavior of $N_{\mathrm{app}}$ in a specific TN architecture) and $P$ is the number of terms in the effective Hamiltonian (including possible projectors), labeled by the index $p$ in Eqs. (73) and (76). Here we assumed for simplicity that each TPO tensor (see Eq. (68)) has two virtual links $\{\mu\}$, all of uniform dimension $D_{\mathrm{TPO}}$. The leading cost for the diagonalization in the single-tensor update is then

$$\mathcal{O}\left( f \cdot \max_{\{\eta\}_1} D_\eta \cdot \prod_{\{\eta\}_1} D_\eta \right), \tag{81}$$

where $\{\eta\}_1$ is the set of links of the tensor $\mathcal{T}^{(1)}$ under optimization. For the single-tensor update with subspace expansion we get a diagonalization cost of

$$\mathcal{O}\left( n \cdot f \cdot \max_{j \in \{1,2\}} \left[ \max_{\{\eta\}_j} D_\eta \cdot \prod_{\{\eta\}_j} D_\eta \right] \right), \tag{82}$$

where $n$ is the number of times that the tensors $\mathcal{T}^{(1)}$ and $\mathcal{T}^{(2)}$ are alternately optimized, and the sets $\{\eta\}_{1,2}$ contain one expanded link $\eta_o'$ with bond dimension $D_{\eta_o'} = D_{\eta_o} + d_{\mathrm{pad}}$. For the double-tensor update the cost for the diagonalization scales like

$$\mathcal{O}\left( f \cdot \max_{\{\eta\}_{1\cup2}\setminus\eta_o} D_\eta \cdot \prod_{\{\eta\}_{1\cup2}\setminus\eta_o} D_\eta \right). \tag{83}$$

The SVD needed for the re-compression of the expanded link at the end of the single-tensor update with subspace expansion has a cost of (see Sec. 2.2.2)

$$\mathcal{O}\left( \left( \min\left[ D_{\eta_o'}, \prod_{\{\eta\}_1\setminus\eta_o'} D_\eta \right] \right)^2 \max\left[ D_{\eta_o'}, \prod_{\{\eta\}_1\setminus\eta_o'} D_\eta \right] \right), \tag{84}$$

while the final SVD in the double-tensor update has a computational cost scaling of

$$\mathcal{O}\left( \left( \min_{j\in\{1,2\}} \prod_{\{\eta\}_j\setminus\eta_o} D_\eta \right)^2 \max_{j\in\{1,2\}} \prod_{\{\eta\}_j\setminus\eta_o} D_\eta \right). \tag{85}$$

In Sec. 5.2 we will see how the general computational cost estimations given here specialize for specific TN architectures. This will also allow for an easier, more evident comparison of the various costs for the different update schemes.

### 5.1.5 Low Excited States

The GS algorithm listed in Sec. 5.1.2 can be extended to determine low excited states, with very little overhead in terms of both implementation effort and computational cost. The basic idea for targeting a specific excited state is to penalize overlap with all lower eigenstates by adding energy penalty terms to the Hamiltonian $\mathcal{H}$. It should be stressed that this method works iteratively, starting from the bottom end of the spectrum. Hence, the $n$-th excited state $|\Psi_n\rangle$ can only be targeted if all lower eigenstates $|\Psi_p\rangle$, $p \in [0, n-1]$ have been determined previously.

More in detail, while for obtaining the GS $|\Psi_0\rangle$ of the Hamiltonian $\mathcal{H}$ we solved the minimization problem

$$E_0 = \langle\Psi_0|\mathcal{H}|\Psi_0\rangle \overset{!}{=} \min. \, , \tag{86}$$

for targeting the $n$-th excited state we now need to solve the minimization problem

$$E_n = \langle\Psi_n|\mathcal{H}|\Psi_n\rangle + \sum_{p=0}^{n-1} \epsilon_p \langle\Psi_n|\mathcal{P}_p|\Psi_n\rangle \overset{!}{=} \min. \, . \tag{87}$$

Here, $\mathcal{P}_p = |\Psi_p\rangle\langle\Psi_p|$ is the projector on the $p$-th eigenstate and $\epsilon_p$ is an energy penalty which has to be set large enough, i.e. at least as large as the energy difference $|E_p - E_n|$ to the target state. Since the target state is not known at the beginning of the algorithm, this energy difference is also unknown; in practice, one therefore simply estimates a value for $\epsilon_p$ which is guaranteed to be large enough (e.g. by setting $\epsilon_p$ one order of magnitude larger than a typical energy scale of the system). For solving Eq. (87), one runs again the algorithm listed in Sec. 5.1.2, additionally taking into account the *projective terms* defined in Eq. (75), as detailed in Sec. 5.1.1. Each projective term contributes exactly one additional interaction term $p$ (see Eq. (76)) to the effective Hamiltonian.

### 5.1.6 Variational Compression, Addition and Orthogonalization

An interesting feature of the presented variational ground state search algorithm is that we can readily use it for *compressing* a generic loop-free tensor network state. This feature then enables further operations such as the addition and orthogonalization of entire tensor networks. For the case of MPSs, the procedure has been described in detail in [39].

**Compression of a Tensor Network.**  Let a state $|\Psi'\rangle$ be given by some loop-free tensor network of bond dimension $D'$. We then seek the best approximation $|\Psi\rangle$ encoded in the same network geometry, but with a smaller bond dimension $D < D'$.

As long as the change in bond dimension affects only one virtual link, an efficient and optimal strategy is to truncate the Schmidt values associated with that bond, as discussed in Sec. 4.1. In all other cases, the optimal compressed state can be found by running the GS algorithm of Sec. 5.1 with bond dimension $D$ for the Hamiltonian $\mathcal{H} \equiv -|\Psi'\rangle\langle\Psi'|$, i.e. with a single projector as described in Sec. 5.1.1. Ideally, the algorithm then converges to the lowest energy $E = -\left|\langle\Psi|\Psi'\rangle\right|^2$ that can be achieved with bond dimension $D$, maximizing the quantum fidelity between compressed and uncompressed TN.

In practice, depending on the severity of compression and the chosen initial state, this approach may improve fidelity but fail to find the optimal solution, even if extended single- or double-tensor update schemes are used as discussed in Sec. 5.1.3. In order to increase robustness and efficiency, it has been suggested to run a sequence of local compressions first, e.g. to successively truncate Schmidt values, bringing all bond dimensions of $\Psi'$ down to $D$ [39]. The accumulation of consecutive truncation errors may prevent that state from being optimal, but it is usually sufficiently well-conditioned to serve as initial state for a few more iterations in the GS algorithm.

For the sole purpose of compression, the GS algorithm no longer requires the sophisticated eigensolvers from Sec. 5.1.4: At the level of tensors, projection onto the compressed state results in effective Hamiltonians taking the form of outer products

$$\left[h_{\text{eff}}^{[c]}\right]_{\{\alpha_\eta\},\{\beta_\eta\}} = -\tilde{\mathcal{T}}_{\{\alpha_\eta\}}^{[c]} \cdot \tilde{\mathcal{T}}_{\{\beta_\eta\}}^{[c]*} \tag{88}$$

as written in full detail in Eqs. (76) and (77). Up to a normalization factor, the lowest eigenstate solution for the center tensor is thus given by $\mathcal{T}^{[c]} = \tilde{\mathcal{T}}^{[c]*}$, which is easily constructed.

**Addition of Tensor Networks.**   A particular need for compression arises in the addition of two or more TNs $|\Psi^{(i)}\rangle$, $i = 1,\ldots,k$. We require that these TNs share the same geometry, including the outcome $|\Psi\rangle$, which shall be the closest approximation to the sum that can be achieved within a fixed maximal bond dimension $D$. Once more, we can find $|\Psi\rangle$ by means of the variational GS algorithm, which handles projections on superpositions of TNs efficiently: After setting up the projector $|\Psi'\rangle\langle\Psi'|$ defined by the sum of TNs $|\Psi'\rangle = \sum_i |\Psi^{(i)}\rangle$ the algorithm for compression can be run as outlined before.

Note that alternatively, one could also first construct the exact sum as a network of bond dimension $D' = \sum_i D^{(i)}$ as follows: Set each resulting tensor $\mathcal{T}'^{[q]}$ to zero and subsequently inscribe the respective tensors $\mathcal{T}^{(i)[q]}$ from TNs $|\Psi^{(i)}\rangle$, $i = 1,\ldots,k$, as subtensors (Sec. 3.5.1) into disjunct index ranges $[1 + \sum_{j=1}^{i-1} D_\eta^{(j)}, \sum_{j=1}^{i} D_\eta^{(j)}]$ for each virtual link $\eta$. The resulting network has to be compressed to the final bond dimension $D$ — a task that requires computational resources scaling polynomially with the bond dimension $D'$, analogous to computing TN scalar products. Unless one resorts to specific data structures for the block-diagonal structure in the contractions of the effective projectors, a drastic compression from dimension $D'$ is therefore less efficient compared to the direct variational compression of a sum of multiple states with smaller bond dimensions each.

**Orthogonalization of Tensor Networks.**   As an example for the addition of TNs, one might want to orthonormalize a TN state $|\Psi_2\rangle$ with respect to another, $|\Psi_1\rangle$. This can be achieved (up to a normalization) by subtracting their common overlap as in $|\Psi_\perp\rangle := |\Psi_2\rangle - |\Psi_1\rangle\langle\Psi_1|\Psi_2\rangle$. After evaluating the involved scalar product, we can employ the variational addition to obtain $|\Psi_\perp\rangle$. Note however that the compression involved in maintaining a fixed bond dimension $D$ usually prevents this approach from delivering an exact orthogonal solution. However, it returns the highest possible fidelity to the exact solution $|\Psi_\perp\rangle$ at the expense of strict orthogonality. Of course, such a result can nevertheless be useful, and one can directly extend this procedure to more than two states, e.g. in a (modified) Gram–Schmidt process.

Interestingly, the specific task of orthogonalization can be addressed more directly with variational compression, similar to targeting excited states: For this purpose, we include in the effective Hamiltonian of Eq. (88) the projectors onto all states that we wish to orthonormalize against. These additional projectors have to be multiplied by a positive energy penalty $\varepsilon_p$. With $\varepsilon_p \to \infty$, orthonormality can be implemented as a hard constraint during compression, and the optimizer's eigenproblem takes the form of a simple orthogonalization against the effective projectors. Conversely to the additive approach, we now expect full orthonormality at the expense of fidelity to the exact solution.

## 5.2   A Practical Application: Binary Tree Tensor Networks

In this section, we discuss a specific loop-free tensor network and see it in action: the binary tree tensor network (bTTN). This network is built on top of a one-dimensional lattice of physical links, but unlike MPS this type of tensor network has the practical advantage of treating open-boundary and periodic-boundary one-dimensional systems on equal footing [182]. It can thus efficiently address periodic systems without the limitations typical of periodic-DMRG [196]. The bTTN is composed entirely of tensors with three links. As we will see, this guarantees moderate scaling of computational cost with the bond dimension $D$. The bTTN architecture with built-in Abelian symmetry handling that we use (as introduced in Sec. 3) is illustrated in Fig. 47. The essential ingredients for achieving the ground state of a many-body Hamiltonian using a loop-free TN ansatz have been outlined previously in Refs. [74, 181, 182] and in Sec. 5.1.2. Here, we summarize some practical technical details that are important when implementing a bTTN ground state algorithm:

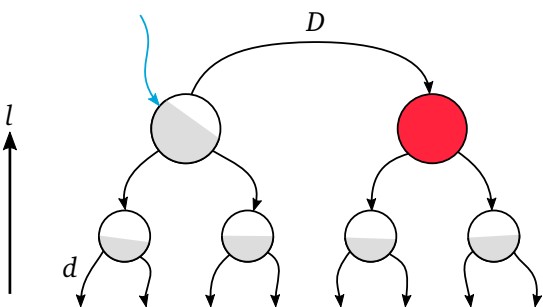

Figure 47: A binary TTN: The black dots at the bottom represent the physical sites of local dimension $d$. Each link in the $l$-th layer (counting from below towards the top) has dimension $\min(d^{2^l}, D)$, where $D$ is a constant that we will refer to as the *bond dimension*. The dangling blue link at the top left tensor has dimension one and carries the global symmetry sector of this symmetric TN state. Its position in the network can be arbitrarily chosen. All tensors are gauged towards a center node (red).

- Choice of decomposition algorithm for central gauging: In order to transform the bTTN into a central gauge, tensor decompositions of the type shown in Fig. 48(a) have to be performed repeatedly (see Sec. 4.2.2). Both the QR decomposition and the SVD are suited to carry out this task. However, as we show in Fig. 48(b), the QR decomposition can be performed in roughly two thirds of the time of the SVD, while exhibiting the same scaling $\mathcal{O}(D^{3.8})$ in the bond dimension (at least in the range $D < 500$ commonly accessible to bTTN simulations). Therefore, it is advisable to implement any gauging routine in a bTTN by means of QR decompositions, and employ SVDs solely for compression purposes.

- Benchmark of diagonalization methods: As mentioned in Sec. 5.1.4, the eigensolver is a component of the GS algorithm which consumes a significant amount of runtime. The performance of this routine therefore crucially influences the whole algorithm runtime and should be first looked at when optimizing the algorithm speed. In Tab. 1 we list three publicly available diagonalization methods, and compare their performances in Fig. 49. Our benchmarks suggest that ARPACK [202] (an implementation of the Arnoldi method [159]) is the most suitable eigensolver in the context of a bTTN ground state algorithm. On a side note, we remark that the setting of a sensible initialization guess can further reduce runtimes of iterative eigensolvers. For the eigenvalue problems occurring in the GS algorithm presented in Sec. 5.1.2, an obvious choice for such an initialization guess is the previous state of the optimization center tensor, before optimization. Especially during the final sweeps, when the tensor elements are already close to their optimal configuration, this measure can significantly reduce the number of necessary operator applications $N_{\mathrm{app}}$. Conversely, one could reduce $N_{\mathrm{app}}$ in the early sweeps of the algorithm by selecting looser starting threshold values for convergence (not included in Fig. 49).

- Computational cost of a bTTN GS algorithm: We observe that the total number of tensors $Q$ scales linearly with the system size $N$, which is the same scaling as in an MPS:

$$Q = \sum_{l=1}^{\log_2(N)-1} 2^l = \frac{2^{\log_2 N} - 1}{2-1} - 1 = N - 2. \tag{89}$$

When using a single-tensor update scheme one deals exclusively with three-link tensors, while for a double-tensor update scheme intermediate four-link tensors occur. Hence,

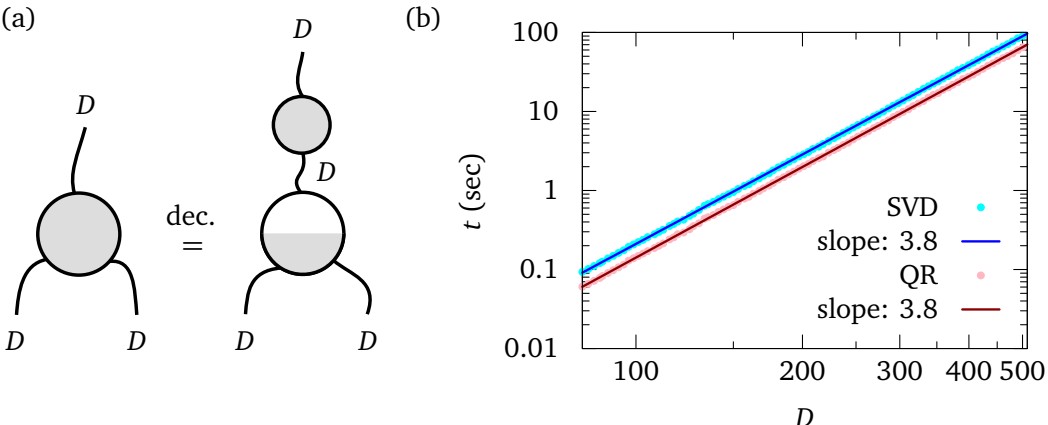

Figure 48: (a) Tensor decomposition arising when transforming a bTTN into central gauge. (b) Corresponding computational performances for QR decomposition (LAPACK routine *zgeqrf*) and SVD (LAPACK routine *zgesdd*). The graph shows the computation time, averaged over 50 runs, needed for the decomposition shown in (a) (performed on a tensor with random entries without any symmetries) as a function of the bond dimension $D$.

Table 1: List and characteristics of some numerical packages capable of finding eigenstates of a hermitian operator $\mathcal{H}_{\text{eff}}$

| Method | Remarks |
|---|---|
| LAPACK (*zheev*) [203] | Full diagonalization (delivers all eigenpairs of $\mathcal{H}_{\text{eff}}$). Requires the construction of $\mathcal{H}_{\text{eff}}$ in full matrix form. |
| ARPACK [202] | Implementation of the Arnoldi algorithm [157, 159]. Iterative diagonalization, designed to deliver a few eigenpairs of $\mathcal{H}_{\text{eff}}$. Requires only knowledge of the action $\mathcal{H}_{\text{eff}}\lvert v\rangle$ on a given vector $\lvert v\rangle$. The number of necessary operator applications $N_{\text{app}}$ strongly depends on the size of the Krylov subspace (parameter *ncv* in ARPACK). A consequence of choosing *ncv* too small is a fast growth of $N_{\text{app}}$ with the problem size $\dim(\lvert v\rangle)$ (where we empirically find that "fast" usually means super-logarithmic, see Fig. 49). |
| Jacobi–Davidson (*jdqz*) [204] | Implementation of the Davidson [158] algorithm. Requirements and features similar to Arnoldi. Here $N_{\text{app}}$ crucially depends on the maximum number of matrix-vector multiplications in the linear equation solver (parameter *mxmv* in *jdqz*). We achieve best performance when choosing $mxmv \approx 5$, i.e. at the lower end of the recommended interval $[5..100]$ (especially when only one eigenpair is demanded). |

(a)

(b)

Figure 49: (a) Eigenvalue problem arising in a bTTN single-tensor update scheme. $\lambda$ is an eigenvalue of the hermitian operator $\mathcal{H}_{\mathrm{eff}}$. (b) Comparison of the performances of the diagonalization algorithms listed in Tab. 1. Upper plot: Computation times needed to solve the eigenvalue problem defined in (a), using an effective Hamiltonian with random entries, as a function of the bond dimension $D$ and for different numbers of calculated eigenpairs $n$. Computation times are averaged over 50 runs, with the error bars indicating the standard deviation. Lower plot: Corresponding number of operator applications $N_{\mathrm{app}}$. Note how $N_{\mathrm{app}}$ only increases weakly (not faster than logarithmically) with the problem size (given that the algorithm parameters are properly configured).

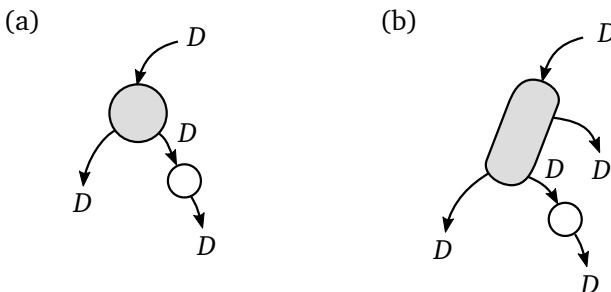

Figure 50: Examples for computationally most expensive contractions in a bTTN GS algorithm (as occurring during the application of the effective Hamiltonian, cf. Fig. 37(c)) in the case of (a) a single-tensor update scheme (contraction cost: $\mathcal{O}(D^4)$), and (b) a double-tensor update scheme (contraction cost: $\mathcal{O}(D^5)$).

we face memory scalings of $\mathcal{O}(ND^3)$ and $\mathcal{O}(ND^4)$, respectively. As for the scaling of the computation time, the most expensive contractions exhibit a cost of $\mathcal{O}(D^4)$ (single-tensor update) and $\mathcal{O}(D^5)$ (double-tensor update), see Fig. 50 for two corresponding example contractions. The SVD in the double-tensor update has a cost bounded by $\mathcal{O}(D^6)$ (see Sec. 5.1.4). Therefore, the algorithm runtime scalings are $\mathcal{O}(ND^4)$ and $\mathcal{O}(ND^6)$, respectively[3]. We stress that for a bTTN all reported scalings are independent of the chosen boundary conditions, and thus apply both for open and periodic boundary conditions.

- Evaluation of local and two-point observables: The expectation value of a local observable $O^{[s]}$, acting on some site $s$ of a bTTN state $|\Psi\rangle$, can be calculated by contracting just three tensors (independent of the system size $N$), if the bTTN is center-gauged towards the tensor $q$ attached to site $s$. This is shown in Fig. 51(a), where we illustrate the expectation value

$$\langle\Psi|O^{[s]}|\Psi\rangle = \sum_{\alpha_s,\beta_s,\alpha_{(s+1)},\alpha_\zeta} \mathcal{T}^{[q]}_{\alpha_s,\alpha_{(s+1)},\alpha_\zeta} O^{[s]}_{\alpha_s,\beta_s} \mathcal{T}^{[q]*}_{\beta_s,\alpha_{(s+1)},\alpha_\zeta}. \tag{90}$$

Two-point correlation observables

$$\langle\Psi|O^{[s]}O'^{[s']}|\Psi\rangle = \sum_{\alpha_s,\beta_s,\alpha_{(s+1)},\alpha_\zeta,\beta_\zeta} \mathcal{T}^{[q]}_{\alpha_s,\alpha_{(s+1)},\alpha_\zeta} O^{[s]}_{\alpha_s,\beta_s} \widetilde{O}'_{\alpha_\zeta,\beta_\zeta} \mathcal{T}^{[q]*}_{\beta_s,\alpha_{(s+1)},\beta_\zeta}, \tag{91}$$

where $\widetilde{O}'$ is the result of a renormalization of the observable $O'^{[s']}$ (obtained from contractions involving all tensors along the path between sites $s$ and $s'$), can be evaluated with a very small number of contractions $\mathcal{O}(\log_2 N)$, owing to the enhanced reachability of two arbitrary sites $s$ and $s'$ due to the hierarchical tree structure (see Fig. 51(b)). This feature not only makes the calculation of observables a computationally cheap task in a bTTN, but also lies at the heart of their predisposition to accurately describe critical systems with algebraically decaying correlations [182].

## 5.3   Binary Tree Tensor Networks – Examples

In this subsection we present bTTN ground state solutions for two different 1D many-body Hamiltonians, both equipped with Abelian symmetries. Namely, we will consider the quantum Ising model in a transverse field (as a typical benchmark model with $\mathbb{Z}_2$ parity symmetry and an easy, well-known analytic solution [8]), and a Bose–Hubbard ring with a rotating barrier [205]

---

[3]Note that the SVD in the single-tensor update with subspace expansion has a cost of $\mathcal{O}(D^4)$ (provided that $d_{\text{pad}}$ is sufficiently small) and therefore does not increase the overall runtime scaling of the algorithm.

(a)

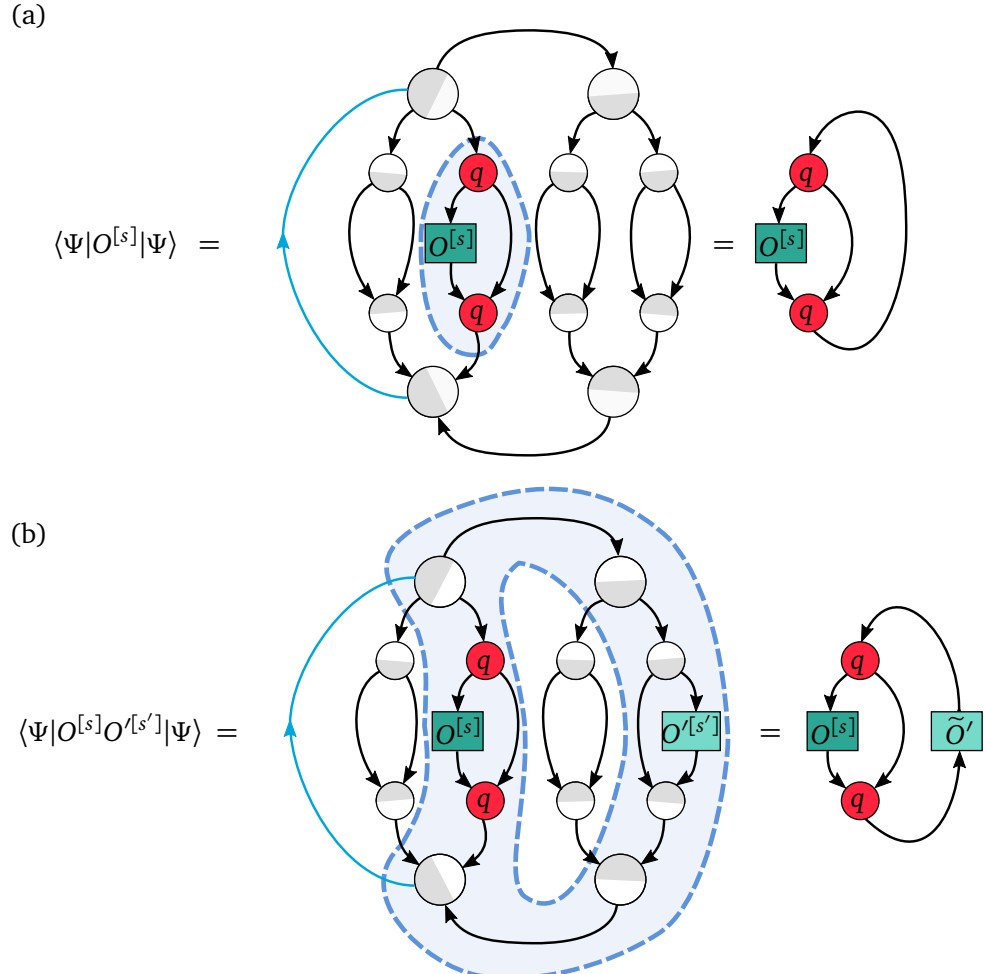

$\langle\Psi|O^{[s]}|\Psi\rangle \ =$

(b)

$\langle\Psi|O^{[s]}O'^{[s']}|\Psi\rangle \ =$

Figure 51: Evaluation of (a) a local and (b) a two-point observable expectation value or a binary TTN state $|\Psi\rangle$. The bra TN $\langle\Psi|$ is drawn as the vertically flipped version of the ket TN $|\Psi\rangle$, indicating hermitian conjugation of all its tensors. If the bTTN is properly gauged, the number of necessary contractions (involved tensors are highlighted) can be reduced to $\mathcal{O}(1)$ and $\mathcal{O}(\log_2 N)$, respectively.

(with U(1) boson number symmetry) as a more complex example. In particular, we provide a field test of the different update schemes presented in Sec. 5.1.3 by comparing both their convergence behaviors and computational resource usage.

**Spin-**$1/2$ **Ising Model.**    We use the following Hamiltonian

$$\mathcal{H} = -\sum_{s=1}^{N}\sigma^{x[s]}\sigma^{x[s+1]} + \lambda\sum_{s=1}^{N}\sigma^{z[s]}, \tag{92}$$

where $\sigma^{x,z[s]}$ are Pauli matrices acting on site $s$. We employ periodic boundary conditions ($N+1 \equiv 1$) and fix the transverse field strength $|\lambda| = 1$, so that the model becomes critical in the thermodynamic limit. At the critical point, the ground-state energy has a simple closed-form solution [8, 206]

$$E(N) = -\frac{2}{N}\left[\sin\left(\frac{\pi}{2N}\right)\right]^{-1}. \tag{93}$$

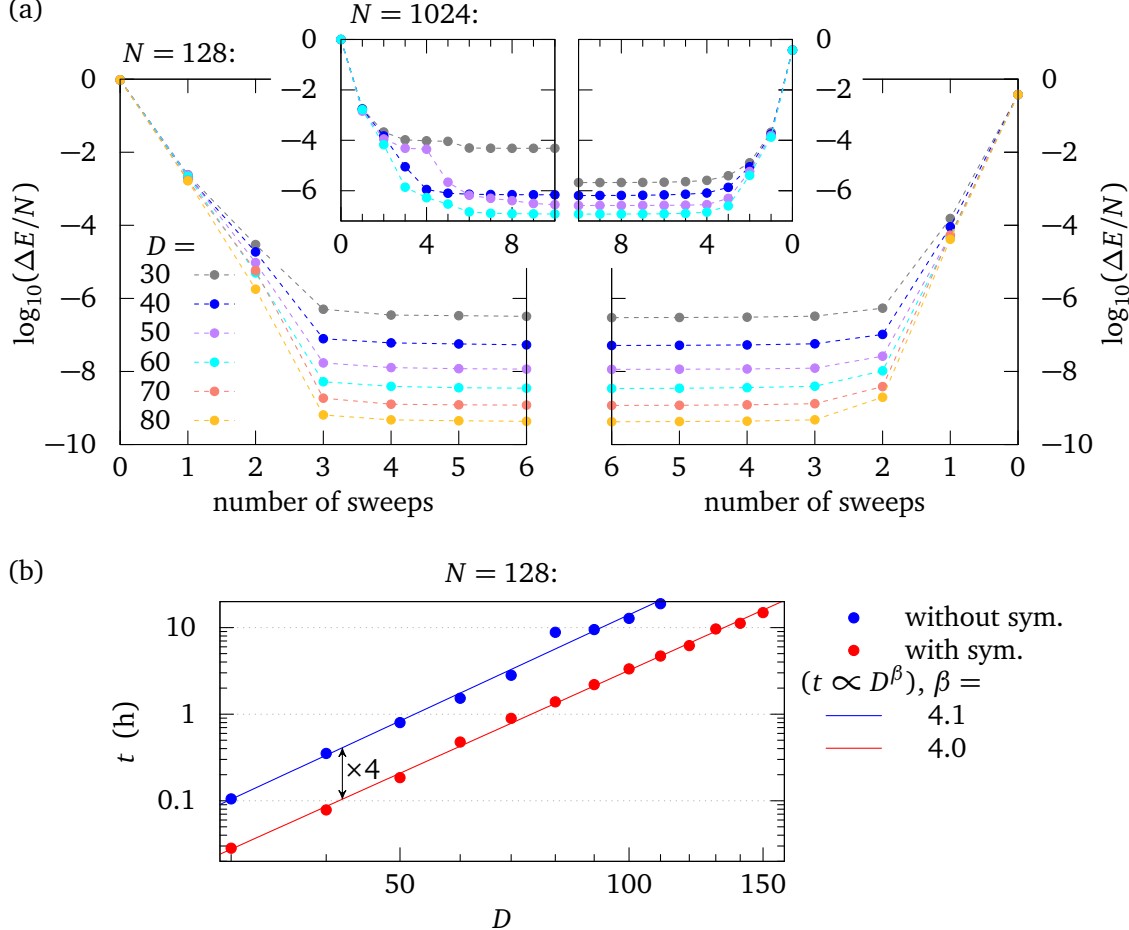

Figure 52: (a) Convergence to the ground state energy of the 1D critical Ising model in a bTTN using a single-tensor update without any inbuilt symmetries (left), compared to a single-tensor update with inbuilt $\mathbb{Z}_2$ symmetry and fixed $\bar{\partial}_e = \bar{\partial}_o = D/2$ (right). Plotted is the error of the ground state energy per site $\Delta E/N$ as a function of the number of optimization sweeps for different bond dimensions $D$ and different system sizes $N$. (b) Runtime comparison of the two algorithms from (a), demonstrating the faster performance of the algorithm with symmetries.

The Hamiltonian defined in Eq. (92) has a global pointwise parity (i.e. group $\mathbb{Z}_2$) symmetry, dividing the state space into two sectors, one with *even* and one with *odd* parity, corresponding to the two representations of $\mathbb{Z}_2$ (for details, see Sec. 3.1). The parity is therefore a good quantum number. Specifically, the Hamiltonian commutes with the symmetry group elements $U(g) \in \{\mathbb{1}^{\otimes N}, \bigotimes_s \sigma^{z[s]}\}$.

For $|\lambda| < 1$ (ferromagnetic phase) the ground state of $\mathcal{H}$ exhibits symmetry breaking, i.e. in the thermodynamic limit there are two degenerate ground states (one in the even sector and one in the odd sector), while for $|\lambda| > 1$ (paramagnetic phase) the thermodynamic ground state is unique and even. For finite system sizes with an even number of sites, the lowest energy state is always in the even sector (i.e. the trivial irrep of $\mathbb{Z}_2$); therefore we will target this global sector in our bTTN simulations. Since there are only two possible symmetry sectors $\ell \in \{e = 0, o = 1\}$, any link representation is defined by two degeneracy subspace dimensions $\bar{\partial}_e$

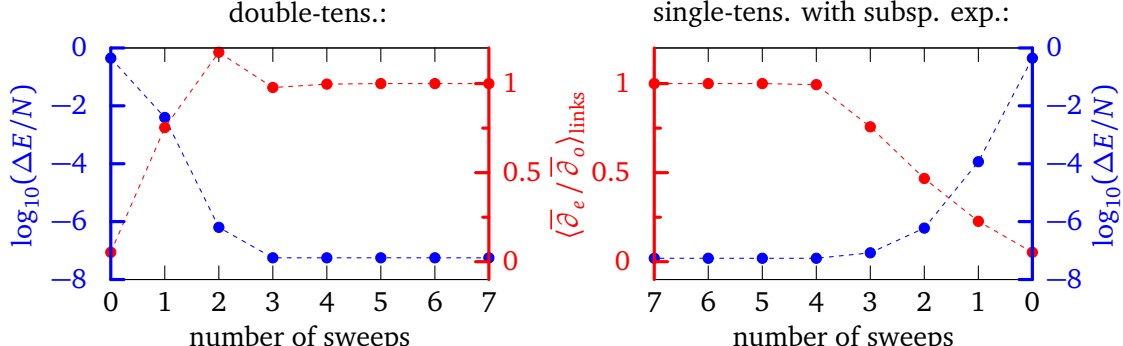

Figure 53: Comparison of link representation convergence for the double-tensor update (left) and the single-tensor update with subspace expansion (right). Plotted is the error of the ground state energy per site $\Delta E/N$ (blue) and the over all virtual links averaged ratio of degeneracy dimensions $\langle \bar{\partial}_e / \bar{\partial}_o \rangle_{\text{links}}$ (red) as a function of the sweep number. For both plots a system size of $N = 128$ and a bond dimension of $D = 40$ have been used.

and $\bar{\partial}_o$. In this sense, it is easy to perform a convergence analysis for virtual link representations during the algorithm runtime. While the ability of a bTTN to accurately describe the ground state of the critical 1D Ising model has been discussed in detail in Ref. [182], here we focus on a comparison between the different algorithm types (with/without inbuilt $\mathbb{Z}_2$ symmetry handling, single-tensor simple update/single-tensor with subspace expansion update/double-tensor update). In Fig. 52(a) we show the convergence to the ground state energy as a function of the number of optimization sweeps. Since the size of the search space is reduced when exploiting symmetries, the convergence behavior towards the ground state is improved in this case. This is most clearly seen for large system sizes, e.g. $N = 1024$. In Fig. 53 we compare the two update schemes: Single-tensor update with subspace expansion and double-tensor update, as described in Sec. 5.1.3. We benchmark their capability to converge to the optimal link representations, which are characterized by $\bar{\partial}_e = \bar{\partial}_o = D/2$ for all links. For the sake of this benchmark, we initialize all virtual links with strongly unbalanced degeneracy dimensions $\bar{\partial}_e \ll \bar{\partial}_o$ and check whether the ratio $\bar{\partial}_e / \bar{\partial}_o$ converges to one as a function of the number of sweeps. Although the double-tensor update converges with slightly fewer sweeps, it is evident from Fig. 53 that both update schemes eventually manage to reach the optimal link representations.

A careful consideration of the theoretically possible runtime reduction resulting from the presence of two symmetry sectors with a flat distribution of degeneracy dimensions (see Sec. 3.5.3) yields a factor of 1/4 for the specific case of a bTTN single-tensor update ground state algorithm. Moreover, the memory consumption is expected to be halved in this case (because half the sectors of each tensor do not couple and therefore do not need to be stored). In practice, we obtained resource savings very close to these predicted values (see Fig. 52(b)), demonstrating that symmetric tensor operations can be implemented with very little bookkeeping overhead.

**Bose–Hubbard Ring With a Rotating Barrier.** A graphical illustration of the system is shown in Fig. 54. In the rotating frame the Hamiltonian can be written as [205]

$$\mathcal{H} = -t \sum_{s=1}^{N} \left( e^{-2\pi i \Omega/N} b^{\dagger[s]} b^{[s+1]} + \text{h.c.} \right) + \frac{U}{2} \sum_{s=1}^{N} n^{[s]} \left( n^{[s]} - 1 \right) + \beta\, n^{[s_b]}, \tag{94}$$

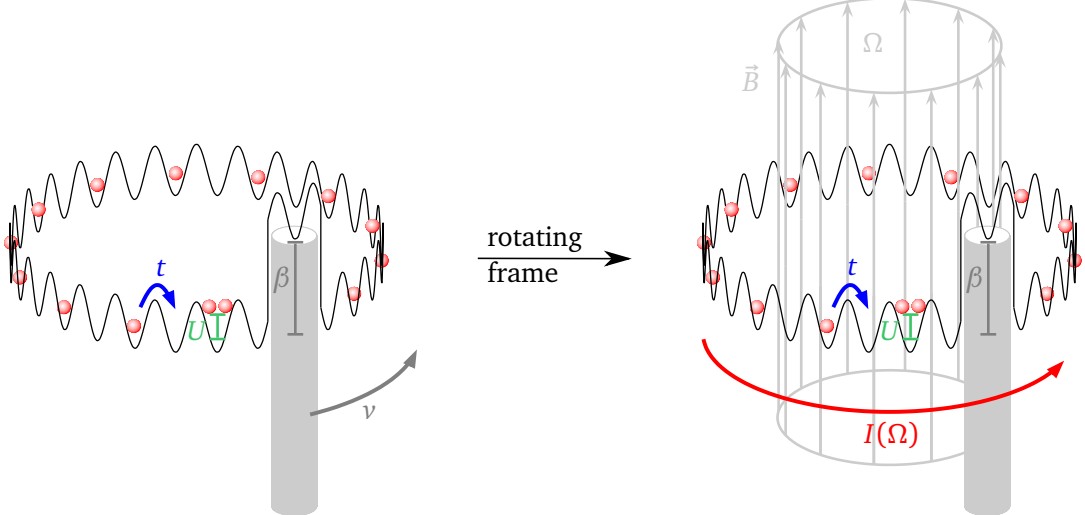

Figure 54: Sketch of a Bose–Hubbard ring with a barrier that rotates with a constant velocity $v$. In the rotating frame the barrier is at rest, but the bosons experience an artificial gauge field $\vec{B}$ giving rise to a flux $\Omega$ through the ring, which leads to a boson current $I(\Omega) \propto \partial E / \partial \Omega$.

where the barrier of height $\beta$ is now at a fixed lattice site $s_b$ and the (artificial) gauge flux $\Omega$ is proportional to the barrier's rotation velocity, $t$ is the hopping amplitude and $U$ the on-site repulsion, while $b^{\dagger[s]}$, $b^{[s]}$, and $n^{[s]}$ are bosonic creation, annihilation, and number operators, respectively. The Hamiltonian defined in Eq. (94) has a total boson number conservation symmetry, which is a global pointwise symmetry with group U(1) (see Sec. 3.1 for details). Specifically, $\mathcal{H}$ commutes with the total boson number operator $n_{\text{tot}} = \sum_s n^{[s]}$, and is thus invariant under $V(\varphi) = e^{i \varphi \, n_{\text{tot}}} = \bigotimes_s e^{i \varphi \, n^{[s]}}$. Hence, the total boson number $N_b$ (with $n_{\text{tot}} |\Psi\rangle = N_b |\Psi\rangle$) is a good quantum number which we can enforce exactly by targeting a specific boson filling in a bTTN algorithm with inbuilt U(1) symmetry handling. Since the U(1) group entails many irreps — potentially all quantum numbers $\ell = 0, \ldots, N_b$ could be present on some links — there is no obvious way *a priori* to assign the sector degeneracies $\bar{\partial}_\ell$ ($\ell = 0, 1, \ldots$) at each virtual link. Therefore, we necessarily need to select the $\bar{\partial}_\ell$ via a variational procedure, as explained in Sec. 5.1.3. We compare again the performances of the two different algorithm types: Double-tensor update strategy against single-tensor update with subspace expansion strategy. The results are summarized in Fig. 55, leading to the same conclusions already obtained from the benchmarks on the Ising model: Both algorithms manage to converge to the optimal link representations, with the double-tensor update requiring a slightly smaller number of sweeps. Nevertheless, the overall computational cost of the double-tensor update is significantly higher (both in terms of runtime and memory consumption), as expected (see Sec. 5.2). We verified that the obtained link representations are indeed optimal by comparing the converged ground state energies to the ones resulting from a bTTN simulation of the same model without inbuilt symmetries. (Without symmetries we have no longer a way of fixing $N_b$ exactly. Nevertheless a desired global filling can be achieved on average by working in a grand canonical ensemble, i.e. by adding an extra term $\mu \sum_s n^{[s]}$ to the Hamiltonian and tuning the chemical potential $\mu$.)

A numerical comparison of the runtime behaviors of the different algorithm types is shown in Fig. 55(c). It appears that the double-tensor update algorithm with inbuilt U(1) performs roughly $\sim 4$ times faster than the single-tensor update algorithm without symmetries, throughout the numerically accessible regime of $D$ (albeit exhibiting a slightly steeper power-scaling). Moreover, the single tensor update with subspace expansion reduces the runtime by another

Figure 55: Benchmark of different bTTN algorithm types for a Bose–Hubbard ring with a rotating barrier ($N = 128$, $N_b = 19$, $\beta/t = 1$, $U/t = 10$, $\Omega = 0$, local bosonic state spaces truncated to $d = 5$). (a) Comparison of ground state energy convergence between double-tensor update (left) and single-tensor update with subspace expansion (right). As reference for the calculation of the error $\Delta E$ the energy from a simulation with bond dimension $D = 400$ has been used. (b) Degeneracy dimensions convergence at the top link of the bTTN, shown for the same two algorithms as in (a). The solid lines are fitted Gaussians, the gray dashed line in the right plot is the final configuration from the left, for ease of comparison. (c) Algorithm runtimes as a function of the bond dimension. The cyan line corresponds to a single-tensor update with fixed degeneracy dimensions $\bar{\partial}_\ell$ (equipped with the converged $\bar{\partial}_\ell$ obtained from the single-tensor update with subspace expansion) and demonstrates that the subspace expansion (purple line) adds a comparatively small overhead.

considerable factor ($\sim 3$ times faster for $D = 50$), on top of having a favorable scaling in $D$, demonstrating the huge potential speed-up accessible by performing TN algorithms with symmetric tensors, especially when the underlying symmetry exhibits a large number of sectors.

# 6 Conclusions

In this anthology we reviewed some cutting-edge computational methods, based on tensor network states, for simulating low-dimensional many-body quantum systems at finite size, approached from the numerical user's perspective. After introducing the concepts and abstract objects which are the building blocks for every tensor network algorithm, we described in detail how these objects are meant to be embedded in a computer. We introduced which manipulations, including (multi-)linear algebra routines, are useful for simulation purposes, stressing the interfaces we suggest for the computation in order to be efficient in terms of time and memory usage. We showed how the same data structures and routines have to be adapted in order to encode pointwise Abelian symmetries, stressing how the conserved quantities help the computation by sensitively reducing the effective variational spaces involved. We reviewed in detail the geometrical and algebraic properties of loop-free tensor networks, focusing on the advantages presented by various gauges and canonical forms. Then, we gave a full description of an algorithm for capturing the properties of ground states and low excited states of Hamiltonians based on loop-free tensor networks: We presented several optimization strategies, and discussed their respective advantages and scaling, while showing practical numerical comparisons on their precision and performance. Altogether, these components form a useful and versatile toolkit which can deal with many low-entanglement scenarios for one-dimensional quantum systems, encompassing the most common simulations in finite-size realizations.

## 6.1 Outlook and Perspectives

Tensor network methods are still far from having reached their ultimate potential. Despite intrinsic difficulties, several steps have been made in the last decade in the direction of simulating two-dimensional quantum systems: 2D quantum physics is inherently fascinating for the non-trivial topological properties of the phases of matter, and consequent exotic quasi-particle statistics (anyons), equilibrium properties still ruled by quantum fluctuations and entanglement (far from mean field description), but it is more computationally challenging. In tensor network language, the challenge translates into the fact that it is difficult to design a tensor network geometry which is both efficiently contractible and capable of capturing the 2D area-law of entanglement [61, 128]. Devising powerful, versatile TN methods able to deal with this scenario is without doubt one of the most relevant challenges of the present decade [62, 195, 207, 208]. Regardless, we stress that recently the same loop-free tensor network architectures that we focused on in this work were employed to study topological properties of interacting two-dimensional models (with toric and open boundary conditions), and showed simulation capabilities well-beyond exact diagonalization [209]. Moreover, loop-free TNs seem to be a promising platform also for out-of-equilibrium simulations, a task likely achievable by adapting the time-dependent variational principle for matrix product states [51, 210, 211], or similar strategies [119, 122], to the more flexible loop-free network geometries, possibly to include finite temperatures [57, 120, 121] and open-system dynamics [212].

## Acknowledgments

The authors gratefully acknowledge support from the EU via the projects SIQS, RYSQ and UQUAM, the German Research Foundation (DFG) via TWITTER, the SFB/TRR21 and OSCAR, the Baden-Württemberg Stiftung via the Eliteprogramm for Postdocs, the Baden-Württemberg Ministry of Research via an LGFG scholarship, the Carl-Zeiss-Stiftung via the Nachwuchs-förderprogramm, and the Studienstiftung des deutschen Volkes via Promotionsförderung. SM gratefully acknowledges the DFG support via a Heisenberg fellowship. Computational resources granted by the bwUniCluster project and by the MOGON cluster of the JGU (made available by the CSM and AHRP) are acknowledged.

# Appendix

## A  Anti-Unitary Symmetries and Time-Reversal

In this section we report a short note on the usage of anti-unitary symmetries for tensor network architectures, and briefly discuss related challenges and implications. To this end we expand the formalism of symmetric tensor networks to include anti-unitary representations of symmetry groups. We remark that anti-unitary symmetries are extremely relevant from a physical point of view since they include, for instance, time-reversal symmetry. As we review here, this characteristic allows in some cases to reduce the Hamiltonian problem from complex to real linear algebra, thus generally providing a computational simplification. Here, we will follow a formulation analogous to Sec. 3.

**Anti-unitary Symmetries.**    A complementary class of invariance $[T, H] = 0$ entails the *anti-unitary* symmetries [213] of the form $T = UK$ where $U$ is unitary, and $K$ shall denote complex conjugation of expansion coefficients in a canonical product-basis $|S\rangle$ of the lattice. Such symmetry arises for instance as a consequence of conventional time-reversal invariance, implying $T^2 = \pm 1$. In the case $T^2 = +1$ (as observed for integer spin or in certain spin-models where spin-degrees of freedom are not explicitly present, e.g. by means of exchange-interactions), $H$ becomes *real* in the sense that it takes the form of a real matrix when written in an invariant basis $T|R\rangle = |R\rangle$. The latter is however easily constructed starting from superpositions in the form $|S\rangle + T|S\rangle$ without having to diagonalize $H$ explicitly. Specifically we consider

$$\langle R|HR'\rangle = \langle TR|THR'\rangle^* = \langle R|HR'\rangle^* \tag{95}$$

by means of anti-unitarity, $T$-invariance of $|R\rangle$ and $|R'\rangle$, and commutation with the Hamiltonian [214].

In the context of lattice models suitable for TNs, our interest lies on cases where we can transform into the basis $|R\rangle$ with a *local* unitary transformation, i.e. one that acts separately on the physical sites. The simplest example is a Hamiltonian that is already real in $|S\rangle$ to begin with, e.g. that is invariant under $T = K$. As a consequence, $H$ becomes real symmetric and all of its eigenstates have real expansions, too. Then, in TN algorithms, by means of a (generalized) gauge transformation, acting locally on links (see Sec. 4.2.1), we can encode certain cases of anti-unitary symmetry by simply restricting tensors to real elements. When realized, we call this the "real gauge" of a TN (see Sec. 4.2.5).

## B  Graphical Notation Legend

Fig. 56 contains the graphical notation legend, reporting the pictorial symbols that we use throughout the manuscript to identify the various TN components (links, tensors, etc.).

Figure 56: Graphical notation legend. Pictorial symbols for links, tensors, their symmetric counterparts, and special properties (such as isometry) are listed.

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
