# Peer review of "The Tensor Networks Anthology: Simulation techniques for many-body quantum lattice systems"

_SciPost Physics Lecture Notes, doi:SciPost Phys. Lect. Notes 8 (2019)_

## Round 1 · Referee Report · Anonymous · 2018-4-12

Strengths
1- Enormous amount of work
2- and supporting graphics
Weaknesses
1- Biased review and bad citation policy.
2- Several key contributions are not mentioned.
3- The average reader would struggle to disentangle the main concepts from the technical details of the implementation.
4- Lack of comparison with existing methods.
Report
I have read the paper and have a mixed feeling, since it clearly contains a huge amount of work. However I find it very difficult in the present form to recommend it for publication. It looks to me at this stage more like an internal note, that misses some of the most important ideas, and fills several pages with irrelevant (to the general reader maybe important for a technical note inside the group) technical details.
I will try to give a constructive feedback in the case the authors want to try to reformulate the paper.
Based on the introduction (pag 5) the anthology should serve both as a review of known tensor network methods and as a basis for teaching a technical introductory course on Tensor Networks, both are very ambitious goals.
Let's start with the review part. A good review should indeed not only be the showcase of the proficiency of the authors on a given technique or subject but also a well equilibrated overview of the relevant results in the field supported by the appropriate references.
I feel that this last aspect is completely missing in the present form. The citation policy seems to me a bit random and clearly overstating the importance of the authors contributions . Key contributions are relegated to mere mentions after papers that build on their results.
A notable example is the fact that the original Vidal paper that mentioned the relation between entanglement and matrix product states is cited as references 46 after a later generalization of Vidal with Orus and a completely marginal paper by Vidal himself.
The foundational paper Physical review letters 93 (4), 040502 is not even cited, and the paper about the thermodynamic limit Physical review letters 98 (7), 070201 and the mixed state simulations, Physical review letters 93 (20), 207205.
Furthemore there are important omissions on the key out-of equilibrium contributions such as t-dmrg (White Feiguin, ...) and the approaches using MPS to compute directly structure factors (a good list of references can be found in the introduction of arxiv:1711.09207 for example), and on theoretical results obtained with tensor networks.
Another example is the mentioning of lattice gauge theories where the authors cite Wilson, the book by Creutz and their own PRX paper. I don’t doubt about the relevance of the paper but if I had to pick up 4 references on lattice gauge theories I am not sure I would list that paper.
Similarly when mentioning the importance of the many body problem in high energy they cite among a couple of references their own review on the topic. Any high-energy physicist would disagree on this choice. The fact that the QCD vacuum is a complex many body system is well known (and actually is at the origin of the invention of lattice QCD by Wilson).
The current self-citation policy makes of the current review a very biased piece of work.
I actually found few of the key papers on tensor network cited in an appropriate way in the anthology. There is an entire section on loop-free tensor networks that includes tree tensor network. All the relevant contribution in those field seem to come from the authors. The enormous developments of tree tensor network in studying relevant quantum many body systems are not even mentioned.
So would I suggest the reading of the anthology to young researcher starting in the field?
Possibly not in its current state. In the abstract of the anthology for example mention and innovative one site update for loop free tensor networks. I failed to understand what is the novelty compared to the traditional approaches presented originally in for example in Ref 60 but this fact is not mentioned or discussed.
This brings me to the section about Abelian symmetries. Here again, the presentation follows closely the results presented in a series of papers in the Vidal group. Despite this fact these papers are only marginally cited at pag 31 of the manuscript. Ref 138 is the basic reference that introduces the ideas presented in this anthology, and Ref 134 specializes it to the case of Abelian symmetries and the presentation here is a reformulation of that of Ref 134. This fact should be explicitly stated.
Furthermore the fact that symmetries and tensor networks have been used in order to characterize the phases of many body quantum systems is not even mentioned (see Polmann, Cirac, Verstraete etc etc).
There are fundamental theorems about the fact that acting with a symmetry operator on a state is equivalent to a gauge transformation on the elementary tensors of the network that are completely ignored.
Even saying that the anthology wants to focus on applications and hands on approaches,
the use of symmetries in numerical tensor networks is based on the main idea that rather than working on normal Hilbert spaces one wants to work on graded Hilbert spaces. I was very surprised that this fact (that is at the basis of the full construction) is not even mentioned. The discussion at Pag. 32 starts with mentioning Abelian symmetric links, without mentioning the structure of the vector spaces they act on. I find it very confusing, another example where the implementation get confused with the basic ideas.
Here I now get to the second part, using the Anthology as a set of lecture notes.
From my point of view, good lecture notes should consist in a good review and in addition a set of technical pieces that should be used as example to allow the student to work on more complicated material independently.
Neither good reviews nor good lectures notes can substitute the assimilation stage of someone that needs to learn the material. In a way the assimilation process cannot be transmitted but is something that one has to perform individually.
I have the impression that the authors have made an attempt to transmit their own assimilation process of the known tensor network material to the readership.
The anthology contains a lot of technical details that are from my point of view non-necessary and rather detrimental to the reader.
Why should the reader decide to implement symmetries in the way the authors have implemented them?
After all there are several open source implementation of symmetric tensor network libraries (think of Uni10, iTensor, TNT etc etc), so if someone is interested in writing his own implementation he/she will
1) need to read the original papers
2) understand the concepts
3) decide how to implement them in his/her own code or alternatively go for an open-source implementation
The present Anthology is somehow in between. It tries to explain the main ideas through an implementation, but it does not provide the code. So I would hardly recommend it in the current form to anyone starting on tensor networks, they would get a quite distorted picture, they would have an hard time to disentangle the basic ideas from the details of the implmentation.
Furthermore, by using this anthology in a course people would get a very biased picture focused on the Authors contributions (in order to justify the relevance of the many-body problem in the introduction they cite their own review for example).
Summarising, I think that in the present form the anthology misses its objectives to be a good review or a set of lecture notes. It is highly biased on the authors contributions ignoring the most relevant paper in the field and not mentioning explicitly what are the basic concepts and where the concepts that are being explained have been introduced originally. The fact that the authors have invested many hours of work in writing such an anthology does not justify the lack of accuracy in their citation policy nor their lack of comparison with existing methods/works.
Requested changes
1- Mention explicitly the original contributions, sort the references on similar topics by date and relevance. Add the key contributions to each of the field addressed.
2- Mention explicitly in the text what was obtained in each of the key references and explain what the current reformulation adds.
3- Compare the optimization methods described with those already available, discuss the differences (if any) and advantages.

---

## Round 2 · Referee Report · Anonymous (Referee 2) · 2019-2-15

Report

I have appreciated the effort made by the authors to improve overall their citation policy.

I am concerned about the editorial policies of SciPost. I tend to see this paper more suited to the Lecture Notes, rather than to the review, but I guess this is a subtle difference.

The snapshot the Review should give about future directions, seems to me very biased here once more towards the authors contributions.

Symmetric Loop free tensor networks have been already implemented and used in the past for systems ranging from Fermionc systems to Gauge theories and conformal field theories. So claiming that they will provide the next breakthrough seems to me exagerated.

In particular I reiterate my concerns about the claims of novelty of the algorithm presented for loop free tensor networks. The numerical results obtained in section 5.2 are analogous to those presented already many times in the literature. There is no comparison with those and no clear advantage that would justify the claims of novelty about the algorithm described.

The subspace expansion is also common practice in tensor network simulations since one usually starts the simulation for larger bond dimension starting from the results obtained with smaller bond dimension.

For these reason I would feel more comfortable if the paper was transferred to SciPost Lecture Notes.
There I would have no further issues about it anymore and I would be in favor of publication.

On the other hand, if novelty and "pointers towards future opportunities" are the criteria that I should judge, I would really indeed like to see a detailed comparison of the results obtained with the methods that the authors claim different from those already known and a clear discussion on why their algorithms should allow to solve problems that the traditional ones have not solved.

Requested changes

Either publish as it is in Lecture Notes or give better "pointers towards future opportunities" that were not there roughly 10 years ago when these algorithms and techniques were first proposed.

  • validity: good
  • significance: ok
  • originality: ok
  • clarity: good
  • formatting: excellent
  • grammar: excellent

Author:  Pietro Silvi  on 2019-02-21  [id 448]

(in reply to Report 1 on 2019-02-15)
Category:
answer to question

We thank the Referee for her/his positive feedback, and agree with her/his suggestion to transfer the submission to SciPost Lecture Notes.

Originally, we believed our submission to be fitting both to 'Physics' and to 'Lecture Notes' sections. We understand now that the format we embraced is more suitable to 'Lecture Notes'. Thus we accept the change, as proposed by the editor.

---

## Round 2 · Author Response

We thank the editor for handling our submission and the referee for her/his thoughtful remarks on the manuscript. We are happy to find that no mistakes or inaccuracies have been reported -- aside from citation policy, where the referee has identified room for improvement. Beyond this, we found few constructive remarks and conversely many general but vague criticisms which lack details on what should be addressed. However, as reported hereafter, we made our best efforts to take them in consideration. We think that the changes we implemented greatly improve the quality of our presentation, and thus we resubmit the paper to SciPost.

We believe a main criticism springs from a misunderstanding on what are the goals of the anthology. We have written it as a self-contained but selective technical introduction to numerical tensor network methods. We do not aim to give an encompassing review on the topic, but instead focus on the basic techniques and selected advanced concepts, and we demonstrate how these basic techniques can be used to build a fully-fledged network algorithm.

In the following, we reply to the points raised by the referee in detail and report the changes we made accordingly (we use page-, citation-, and figure numbers from the resubmitted manuscript):

(1) General aim and conceptual critique

(1.1) Referee: [...] misses some of the most important ideas, and fills several pages with irrelevant (to the general reader maybe important for a technical note inside the group) technical details. Referee: The anthology contains a lot of technical details that are from my point of view non-necessary and rather detrimental to the reader. Referee: [...] not mentioning explicitly what are the basic concepts [...]

Reply: We clearly are not exhaustive in ideas and concepts in tensor networks (for instance we specialize in Sections 4 and 5 to loop-free networks), but completeness is not the intended goal of our anthology, as we stated for instance in the introduction ("We focus mostly on those tensor network techniques which have been previously published by some of the authors.", "[..] it does not contain an extended review of physical results or a pedagogical introduction to the field of tensor networks in general. For those two aspects, we refer the reader to the excellent reviews in Refs. [19, 39]."). Following the referee's concerns, we have additionally revised the abstract to make our aim more clear. We are confident that we properly introduce the basic concepts needed for our presentation, and we believe that the technical details given provide practical value to the reader, setting this work apart from existing literature.

(1.2) Referee: [...] anyone starting on tensor networks, they would get a quite distorted picture, they would have an hard time to disentangle the basic ideas from the details of the implementation.

Reply: For starters on tensor networks, we explicitly recommend the excellent reviews [19,39] early on in the introduction. Our work is furthermore not meant for readers who want to use tensor networks as an exclusively analytical tool, as we emphasize with the abstract ("We present a compendium of numerical simulation techniques [...]"). Our focus lies on a presentation accessible from an implementation point of view, and we separate basic ideas from implementation details and algorithms: We first briefly motivate and introduce every topic, before turning to the more technical content. We actually believe that bringing together basic concepts and implementation in a single framework is a feature that distinguishes our work, and serves as a bridge for readers intending to implement or use tensor network techniques as a numerical tool in their research.

(1.3) Referee: From my point of view, good lecture notes should consist in a good review and in addition a set of technical pieces that should be used as example to allow the student to work on more complicated material independently. Referee: Based on the introduction (pag 5) the anthology should serve both as a review of known tensor network methods and as a basis for teaching a technical introductory course on Tensor Networks, both are very ambitious goals.

Reply: We believe that our ambitions are different from what the referee is expecting from this anthology: As said before, we did not attempt to write an extensive, let alone complete, review of tensor network methods. We are convinced, also based on the experience of some of the authors, that the anthology can be of great value to graduate- or PhD students, or generally to anyone who is not yet fully acquainted with the modern framework of numerical tensor network methods. We target an audience that aims to either implement or use those methods (e.g. with existing libraries) as building blocks in their high-level tensor network algorithms. We also envision it as an accompanying material in a computational quantum mechanics course for instance, or in study projects. Furthermore, we believe it can be beneficial to scientists coming from different backgrounds that seek to use the power of computational TN methods for their own field of research, given that the areas of applications of TN methods are becoming more and more diverse.

(1.4) Referee: The present Anthology is somehow in between. It tries to explain the main ideas through an implementation, but it does not provide the code.

Reply: We agree, we see it as the referee -- our work stands in between. Actually, we firmly believe that this specific format is a strong point of our manuscript, as it makes it complementary to the existing literature on TN: We indeed approach the topic from a numerical user's or programmer's point of view, but avoid getting lost in implementation- or language-specific details, and we do not want to provide a mere documentation as has already been done in various forms, as the referee has pointed out.

(1.5) Referee: Why should the reader decide to implement symmetries in the way the authors have implemented them? After all there are several open source implementation of symmetric tensor network libraries (think of Uni10, iTensor, TNT etc etc), so if someone is interested in writing his own implementation he/she will 1) need to read the original papers 2) understand the concepts 3) decide how to implement them in his/her own code or alternatively go for an open-source implementation

Reply: We fully agree with the referee that there is no need to follow our outlined implementation in full depth, and that there are many ready-to-use implementations out there building on similar concepts. We are aware of those implementations and refer the reader to them. Indeed, the basic concepts here are all closely related, based on block-decompositions in tensors and quantum numbers tied to (virtual) link representations. Our introduction thus outlines the concepts behind existing implementations, while the reader can also implement the proposed objects from our manuscript in a programming framework of her or his liking. Our suggested implementation has proven to be efficient, fast, and a natural extension of non-symmetric tensors with arbitrary numbers of links. Furthermore, the proposed symmetric tensor- and link objects serve as an exemplary reference implementation that we draw upon in the rest of the manuscript (e.g. in the description of symmetric tree-TN). We believe that our manuscript provides a useful and coherent reformulation of the original concepts introduced in the diverse literature, in that - it organizes the topics in a modular form (Fig. 1), both from a pedagogical and implementational perspective; - it can be used as a complementary, self-consistent source, which provides shortcuts to the relevant literature and presents the material in a consistent and unified formulation; - it assembles practical tips and proven recipes in a single document, so the reader can easily combine them; - it provides the essential overview of use-cases, runtimes, optimization techniques and possible extensions that one should be aware of before designing an algorithm or implementation. We found that these aspects make coding and developing numerical TN algorithms more efficient, and a manuscript covering all of them was missing from the literature. Specifically on (Abelian) symmetries, the principles are clear, but the issue we try to address is how to make them work in practice: We have learned from several students and researchers, asking us how to actually implement symmetries in tensor networks, that these questions deserve more attention in the literature.

(1.6) Referee: In the abstract of the anthology for example mention and innovative one site update for loop free tensor networks. I failed to understand what is the novelty compared to the traditional approaches presented originally in for example in Ref 60 but this fact is not mentioned or discussed. Referee: [...] lack of comparison with existing methods/works.

Reply: In light of the referee's comment, we have changed the sentences on the innovative one-site update in the abstract and introduction and added a paragraph to briefly introduce (and discuss the differences to) existing methods of perturbative subspace expansion of Refs. [143,201] at the end of Sec. 5.1.3 (p. 84). While the presented single-tensor update with subspace expansion is a) innovative, b) can be implemented straightforwardly, and c) demonstrates good performance and usefulness in Sec. 5.3, the main reason why we report it in the anthology is that it demonstrates the interplay of basic techniques (e.g. symmetric link operations and sub-tensors), and how they can be combined to overcome additional challenges arising in (Abelian) symmetric tensor networks.

(2) Concerning the referee's comments on citations

(2.1) Referee: A good review should indeed not only be the showcase of the proficiency of the authors on a given technique or subject but also a well equilibrated overview of the relevant results in the field supported by the appropriate references. I feel that this last aspect is completely missing in the present form.

Reply: We reiterate our statement from above on the selective focus of our work. Under this premise, we do not agree that an "overview [...] supported by the appropriate references" is "completely missing". For instance, we included an introduction to the main developments in, and a brief section on historical origins of, tensor network methods in Sec. 1, accompanied by the appropriate references. Thanks to the referee's suggestions, we have now critically revised our references therein and expanded the list.

(2.2) Referee: The citation policy seems to me a bit random and clearly overstating the importance of the authors contributions. Referee: The current self-citation policy makes of the current review a very biased piece of work. Referee: [...] by using this anthology in a course people would get a very biased picture focused on the Authors contributions (in order to justify the relevance of the many-body problem in the introduction they cite their own review for example). Referee: [...] it is highly biased on the authors contributions ignoring the most relevant paper in the field

Reply: We acknowledge that the selection of topics follows the authors fields of work -- a choice that necessarily must ignore many research directions in this broad field. We further agree with the referee that reference lists had been kept too short. We improved this aspect by thoroughly expanding on the references. While formerly self-citations made up 18 among 170 references (10.6 %), our resubmission counts 21 self-citations among 214 references (9.8 %), a ratio which we do not deem excessive. Without knowing what work the referee has in mind, we cannot agree with her/his concern on "ignoring the most relevant paper in the field", as we think we have included most - if not all - of them.

(2.3) Referee: Key contributions are relegated to mere mentions after papers that build on their results. Referee: I actually found few of the key papers on tensor network cited in an appropriate way in the anthology. Referee: not mentioning explicitly [...] where the concepts that are being explained have been introduced originally Referee: [...] lack of accuracy in their citation policy

Reply: We thank the referee, indeed we were able to improve our citation policy based on her/his remarks: We added and regrouped citations throughout the manuscript to improve the citation style. In general, we keep literature review brief and concentrated to the introductory paragraphs of each section. Citations are provided in the usual form, without aiming at an extensive or historic review that does not fit our intention to provide a practical hands-on guide with the required brevity. However, we do not agree with the statement that "few of the key papers on tensor network" were "cited in an appropriate way.", because we do name the generally accepted key papers in the introductory parts. Where the referee made specific remarks, we address them below.

(2.4) We address the specific remarks of the referee on citations in the following:

(2.4a) Referee: A notable example is the fact that the original Vidal paper that mentioned the relation between entanglement and matrix product states is cited as references 46 after a later generalization of Vidal with Orus and a completely marginal paper by Vidal himself. The foundational paper Physical review letters 93 (4), 040502 is not even cited, and the paper about the thermodynamic limit Physical review letters 98 (7), 070201 and the mixed state simulations, Physical review letters 93 (20), 207205.

Reply: Some of the foundational papers are clearly included and highlighted in the cited reviews, and some of them are also explicitly cited by us later on when some specific aspects have been addressed. This is the reason why some foundational PRL's appear after the reviews, and we are convinced that there is no fundamental problem with this citation policy. However, following the referee's remarks, we cite Ref. [26] (former Ref. 46) now early on in the introduction, and we reworked our citations of fundamental works on area laws in the same context. We agree that PRL 93 (4), 040502 is an important work, derived from the seminal paper [26] (former Ref. 46), which warrants a separate citation and is now added as Ref. [48]. The work PRL 98 (7), 070201 about the thermodynamic limit was formerly cited in Sec. 4 in the context of gauging, and we have now cited it more appropriately in the introduction, too. The work PRL 93 (20), 207205 about mixed state simulations was formerly cited in Sec. 1.5, we now also cite it in Sec. 1.2 together with its preceding letter PRL 93 (20), 207204.

(2.4b) Referee: Furthemore there are important omissions on the key out-of equilibrium contributions such as t-dmrg (White Feiguin, ...) and the approaches using MPS to compute directly structure factors (a good list of references can be found in the introduction of arxiv:1711.09207 for example), and on theoretical results obtained with tensor networks.

Reply: The referee correctly points out that there is a huge body of literature on tensor networks. In the context of TEBD we completed our references by adding the suggested Ref. [49] for the t-DMRG. Generally, we included a subset of topics and references that we deemed to be important enough to appear in the document, but others are treated marginally, and to avoid a disproportionate reference list we omitted them. The use of tensor networks for obtaining analytical/theoretical results is one such example. Again, we think that this is a reasonable citation policy and that authors have the right to make their own choices at this level of detail.

(2.4c) Referee: [..] when mentioning the importance of the many body problem in high energy they cite among a couple of references their own review on the topic. Any high-energy physicist would disagree on this choice. The fact that the QCD vacuum is a complex many body system is well known (and actually is at the origin of the invention of lattice QCD by Wilson). Referee: Another example is the mentioning of lattice gauge theories where the authors cite Wilson, the book by Creutz and their own PRX paper. I don’t doubt about the relevance of the paper but if I had to pick up 4 references on lattice gauge theories I am not sure I would list that paper.

Reply: We have substantially expanded the reference list in order to reflect the vivid research in this field and updated Refs. [5,6]

(2.5) Referee: There is an entire section on loop-free tensor networks that includes tree tensor network. All the relevant contribution in those field seem to come from the authors. Referee: The enormous developments of tree tensor network in studying relevant quantum many body systems are not even mentioned.

Reply: Our choice of references, which included contributions from Refs. [70,181], has been substantially expanded here.

(2.6) Referee: [...] the fact that symmetries and tensor networks have been used in order to characterize the phases of many body quantum systems is not even mentioned (see Polmann, Cirac, Verstraete etc).

Reply: We thank the referee for pointing us to this important application, which utilizes a closely related construction of tensor networks based on symmetry properties. Our limited scope does not allow for a detailed discussion of those admittedly intriguing aspects. For completeness, we now refer the reader to the comprehensive Ref. [179] on the subject (p. 28) and references therein.

(3) Concerning the referee's comments on the section about Abelian symmetries:

(3.1) Referee: [...] the presentation follows closely the results presented in a series of papers in the Vidal group. Despite this fact these papers are only marginally cited at pag 31 of the manuscript. Ref 138 is the basic reference that introduces the ideas presented in this anthology, and Ref 134 specializes it to the case of Abelian symmetries and the presentation here is a reformulation of that of Ref 134. This fact should be explicitly stated.

Reply: Our basic reference for the conception and presentation is indeed [135], and we have added a sentence in Sec. 3.2 (p. 28) to clarify this fact: "Our presentation primarily draws upon Ref. [135], which specializes the underlying construction introduced in Ref. [170] to the case of an Abelian group." However, we also intentionally deviate, and omit concepts from, the presentation in e.g. [135,170] as the referee notes later. Omissions include for instance graded vector spaces, tensors as linear maps, fusion trees, etc., which we deem not relevant in our documentation, aiming at a complete yet simple and practical introduction to Abelian symmetric tensor networks and their implementation. For instance, we base our definition of tensors and links on index sets and quantum numbers, which are entirely sufficient for our purposes and typically closer to practical implementations, as opposed to (graded) vector spaces. For this reason we do not follow that presentation. We also decompose Abelian symmetric tensors of any number of links without introducing additional concepts such as explicitly encoded fusion trees in tensors.

(3.2) Referee: [...] the use of symmetries in numerical tensor networks is based on the main idea that rather than working on normal Hilbert spaces one wants to work on graded Hilbert spaces. I was very surprised that this fact (that is at the basis of the full construction) is not even mentioned. The discussion at Pag. 32 starts with mentioning Abelian symmetric links, without mentioning the structure of the vector spaces they act on. I find it very confusing, another example where the implementation get confused with the basic ideas.

Reply: Expanding on our reply in (3.1), throughout the anthology, we treat tensors as "arrays" of elements over index spaces (consistent with the definition on p. 13 in Sec. 2.1). We do not treat tensors as linear maps over (dual) vector spaces. Consistently, symmetric links are introduced as a direct extension of the index sets in ordinary tensor networks. Instead of graded spaces, we introduce directed links in Sec. 3.3.1 (p. 28f), with the link direction being a property of the symmetric tensor object. In Sec. 3.3.2 (p. 29) and also Sec. 3.3.4 (p. 33) we changed the wording in order to clarify this important concept. Based on these definitions, the full construction of symmetric tensor networks follows in Sec. 3.4 (p. 33f), and Fig. 16. We believe that our presentation does not lack, nor confuse, these basic ideas and instead brings all basic ideas together in a simple, self-consistent presentation.

(3.3) Referee: There are fundamental theorems about the fact that acting with a symmetry operator on a state is equivalent to a gauge transformation on the elementary tensors of the network that are completely ignored.

Reply: We cover the relation between the action of a global symmetry operation and gauge transformations briefly in Sec. 3.4 (p.33f) and Fig. 16, based on Eq. (30) and its context (p. 29). We have now explicitly cited Ref. [170] in Sec. 3.4.

Below we respond on how we implemented the changes requested by the referee:

1 - We have improved our citations throughout the manuscript, with specifics detailed in (2). To the best of our knowledge, the original papers are cited. We argue that the brevity of our limited review does not allow for, and neither require, an explicit citation style with references sorted by date and relevance. However, we have carefully regrouped and reordered citations accordingly, insofar as supported by the SciPost bibliography style.

2 - With the changes made in the abstract and our replies in (1), we state that the intention of the manuscript is to provide a practical guide to TN methods accessible from an implementation perspective. Indeed the manuscript provides a reformulation of the ideas and concepts from our references, which we cite in the introductory paragraphs of each technical section. To name an example, in Sec. 5.2 we open with "The essential ingredients for achieving the ground state of a many-body Hamiltonian using a loop-free TN ansatz have been outlined previously in Refs. [74, 181, 182] and in Sec. 5.1.2". We also added clarifications, e.g. as detailed in (3.1).

3 - See our reply to (1.6)

---

## Round 2 · List of Changes

- Thorough revision and improvements in sentence structures, wording and notation, grammar, symbols and indices, throughout the document
- Revised and significantly expanded citations throughout the document
- Added sentence to explicitly clarify that we draw upon Refs [135,170] for Abelian symmetries
- Revised the abstract to make our aim more clear and removed sentence on single-tensor update.
- Added paragraph on perturbative subspace expansion methods, and comparison to the methods presented
- More concise presentation of the symmetric tensor decomposition (Sec. 3.5.2, p. 49) with corresponding labels added in Fig. 23
- Added suggestion for increasing efficiency and robustness of the compression of a tensor network (Sec. 5.1.6, p. 87)
- Equation numbering

---

## Editorial Decision

published